# A redefined InDel taxonomy provides insights into mutational signatures

Gene Ching Chiek Koh [1,2,3,5], Arjun Scott Nanda [1,2,5], Giuseppe Rinaldi [1,2], Soraya Boushaki[1,2], Andrea Degasperi [1,2], Cherif Badja [1,2], Andrew Marcel Pregnall [1], Salome Jingchen Zhao[1,2], Lucia Chmelova[1,2], Daniella Black[1,2], Laura Heskin [1,2], João Dias [1,2], Jamie Young[1,2], Yasin Memari[1,2], Scott Shooter[1,2], Jan Czarnecki [1,2], Matthew Arthur Brown [4], Helen Ruth Davies [1,2], Xueqing Zou[1,2] & Serena Nik-Zainal [1,2] ✉

Despite their deleterious effects, small insertions and deletions (InDels) have received far less attention than substitutions. Here we generated isogenic CRISPR-edited human cellular models of postreplicative repair dysfunction (PRRd), including individual and combined gene edits of DNA mismatch repair (MMR) and replicative polymerases (Pol ε and Pol δ). Unique, diverse InDel mutational footprints were revealed. However, the prevailing InDel classification framework was unable to discriminate these InDel signatures from background mutagenesis and from each other. To address this, we developed an alternative InDel classification system that considers flanking sequences and informative motifs (for example, longer homopolymers), enabling unambiguous InDel classification into 89 subtypes. Through focused characterization of seven tumor types from the 100,000 Genomes Project, we uncovered 37 InDel signatures; 27 were new. In addition to unveiling previously hidden biological insights, we also developed PRRDetect—a highly specific classifier of PRRd status in tumors, with potential implications for immunotherapies.

Small insertions and deletions (InDels; <100 bp) represent the second most prevalent form of genetic variation after substitutions[1]. InDel mutagenesis is substantial and nonrandom, reflecting underlying mutational processes, be they benign by-products of normal physiology or malign consequences of exogenous exposures and/or endogenous dysfunctions.

While studies of mutational processes over the last decade have focused primarily on substitutions[2,3], recent advancements in InDel detection and annotation[4–6] have led to the identification of 18 small InDel signatures (IDS) in human cancers[7]. These signatures were defined using an 83-channel classification system (that is, 83 InDel subtypes, herein referred to as COSMIC-83), founded upon characteristics

such as the InDel size, nucleotides affected, lengths of flanking mononucleotide/polynucleotide repeat and sequence homology at the InDel junctions. Subsequent re-analysis of the same dataset using a revised algorithm reported nine additional de novo IDS[8].

Accurate InDel characterization is crucial for biological and clinical purposes. For instance, a high proportion of microhomology-mediated deletions is a key predictor of clinically actionable homologous recombination deficiency, carrying the greatest weight in homologous recombination deficiency detection and classification algorithms[9,10]. Additionally, the detection of microsatellite instability (MSI) in mismatch repair (MMR)-deficient tumors relies on measuring genome-wide InDel mutagenesis and/or analyzing panels of

[1]Department of Genomic Medicine, School of Clinical Medicine, University of Cambridge, Cambridge, UK. [2]Early Cancer Institute, Department of Oncology, University of Cambridge, Cambridge, UK. [3]School of Medical and Life Sciences, Sunway University, Sunway City, Malaysia. [4]Genomics England, Queen Mary University of London, Dawson Hall, Charterhouse Square, London, UK. [5]These authors contributed equally: Gene Ching Chiek Koh, Arjun Scott Nanda. ✉e-mail: sn206@cam.ac.uk

mononucleotide/dinucleotide repeats[11–13]. Recent studies have also highlighted how 2–5 bp short deletions and 2–4 bp duplications are distinct readouts of TOP1 activity (amplified in *RNaseH2*-null cells)[14] and TOP2A dysfunction[15], respectively.

Given the increasing importance of InDel mutagenesis in tumor classification and prediction of therapeutic sensitivities[16–19], we established a 'ground truth' set of experimental IDS focused on 'postreplicative repair deficiency' (PRRd), which encompasses defects in MMR and replicative polymerase proofreading—biological abnormalities that often display exquisite sensitivity to immune checkpoint inhibition (ICI)[20–22]. We uncovered inherent limitations in the prevailing InDel classification schema (COSMIC-83) that hamper its ability to distinguish biologically distinct signatures. To address this, we explored whether incorporating sequences 5′ and 3′ to an InDel—features crucial for substitution classification—could contribute additional understanding and resolution power. Additionally, sequence motifs known to increase mutational vulnerability and their genome-wide prevalence were factored into our proposition. Our approach unambiguously classifies every InDel into a specific subcategory. Here we demonstrate that our alternative InDel taxonomy uncovers new etiologies of InDel mutagenesis, offering mechanistic insights and potential clinical added value.

## Results

### Diversity of InDel patterns in PRRd

We generated a 'ground truth' set of isogenic cellular models by introducing CRISPR edits to key PRRd-associated genes in an hTERT-immortalized RPE1 (*TP53*[−/−]) cell line[23]. We created four single MMR gene knockouts (*ΔMLH1*, *ΔMSH2*, *ΔMSH3* and *ΔSETD2* (ref. [24])), two knock-in missense mutants of polymerase Pol ε (POLE exonuclease mutant p.P286R and p.L424V), two mutants of Pol δ (POLD1 exonuclease mutant p.S478N and polymerase mutant p.R689W) and three double mutants with combined polymerase proofreading mutation and MMRd (*POLD1*[S478N/+]*ΔMLH1*, *POLD1*[S478N/+]*ΔMSH2* and *POLE*[P286R]*ΔMSH2*; Supplementary Tables 1 and 2). Successfully edited clones were propagated in culture for approximately 45–50 days to permit mutation accumulation. Subsequently, two to five daughter subclones were isolated per genotype for whole-genome sequencing (WGS) and mutational signature analyses (Fig. 1a).

Except for *ΔSETD2*, we observed elevated InDel burdens in all gene edits compared to an unedited control (background) (Fig. 1b and Supplementary Table 3). The mutation burden was approximately twofold higher in *ΔMSH3* and *POLD1*[R689W], tenfold in *POLD1*[S478N], *POLE*[L424V] and *POLE*[P286R], 55-fold in *ΔMSH2* and *ΔMLH1* and particularly significant in combined gene edits—around 200-fold in *POLD1*[S478N/+]*ΔMLH1* and *POLE*[P286R]*ΔMSH2*, and 300-fold in *POLD1*[S478N/+]*ΔMSH2*.

All lines except *ΔSETD2* showed variations in their COSMIC-83 InDel signature profiles compared to control (Fig. 1c and Supplementary Table 4). We noted discriminative characteristics between gene edits (Fig. 1d and Extended Data Fig. 1). Dominant 1 bp T deletions at homopolymers of 6 bp or more (poly-T$_{6+}$) were observed for *ΔMLH1*, *ΔMSH2* and *ΔMSH3*, while *POLD1*[S478N] and *POLE*[P286R] showed exclusive 1 bp T insertions at poly-T$_{5+}$. *POLD1*[R689W], *POLE*[L424V] and all three combined polymerase/MMRd edits predominantly exhibited 1 bp T insertions at long homopolymers, although not exclusively, with variations of 1 bp T deletions between different genotypes. Together, these experiments revealed unique, diverse InDel signatures among different PRRd mutants. Remarkably, mutations within the same gene but affecting different functional protein domains manifested signature variations (that is, POLD1 exonuclease p.S478N versus polymerase p.R689W).

We also examined the substitution profiles of all gene edits (Extended Data Fig. 2 and Supplementary Table 5). Intriguingly, MMRd lines showed lower substitution-to-InDel ratios compared to control, while polymerase-dysfunction (Pol-dys) lines exhibited markedly increased ratios (Extended Data Fig. 2h). This suggests that genome instability is predominantly driven by an excess of InDel

mutagenesis in MMRd, whereas substitution mutagenesis plays a more significant role in polymerase proofreading dysfunction. Furthermore, mutational asymmetry analyses revealed enrichment of both substitutions and InDels on the leading strand for *POLE* mutants while *POLD1* mutants exhibited lagging strand bias, specifically T insertions at homopolymeric tracts of 5–7 nts (Supplementary Fig. 1 and Supplementary Table 6). This is in keeping with the hypothesized preferential activity of Pol ε and Pol δ in leading and lagging strand synthesis, respectively[25,26], suggesting that *POLE*/*POLD1* mutants tend to accumulate 1 bp A insertions on the nascent strand while replicating through 5–7 nts poly-T-tracts[27,28]. This lends support to the proposition that polymerase ε and δ are more proficient at detecting incorrectly paired bases at template adenines[27].

Nevertheless, while the diversity of experimental InDel profiles was appreciable among PRRd genotypes, it was difficult to disambiguate gene-edit signatures from background mutagenesis. Clustering analyses and direct comparisons of gene edit and control InDel profiles showed extremely high similarity (cosine similarity > 0.9; Extended Data Fig. 1a,b,d). Discrimination between MMRd and Pol-dys signatures was also limited (Extended Data Fig. 1e). Unsupervised clustering using cosine distance revealed mainly two groups of signatures—deletion-driven MMRd signatures and insertion-driven polymerase mutant signatures (Extended Data Fig. 1f). We thus investigated the sufficiency of COSMIC-83 taxonomy, given that signal variation among the ten gene edits was primarily observed in two channels—1 bp T insertions at poly-T$_{5+}$ and 1 bp T deletions at poly-T$_{6+}$.

### Limitations of current InDel taxonomy

We compared experimental gene-edit InDel signatures with COSMIC IDS[7]. InDel signatures of *ΔMSH2* and *ΔMLH1* showed no similarity to the purported MMRd-associated ID7 (Fig. 1f). Instead, *ΔMSH2* and *ΔMLH1* signatures most resembled ID1 and ID2, ascribed to normal replication errors associated with nascent and template strand slippage, respectively (Fig. 1d–g).

The COSMIC-83 taxonomy aggregates 1 bp InDels at homopolymers >5 bp into single channels (that is, T$_{6+}$ for deletions and T$_{5+}$ for insertions, respectively; Fig. 1e and Extended Data Fig. 3a–d). Yet, the probability of microInDel formation increasing with the length of simple nucleotide repeats in MMRd[29] is a recognized hallmark of MSI[30]. We surmised that the conflation of discriminatory signals within longer homopolymers into single 'insertion at T$_{5+}$' or 'deletion at T$_{6+}$' channel likely reduces the separative capacity for signature extraction. Hence, MMRd signatures cannot be distinguished from signatures of normal replication errors. This contrasts with corresponding PRRd-associated substitution signatures, which manifest as distinct and diverse patterns amongst MMRd and/or Pol-dys cancers[3,7,31] (Extended Data Fig. 2b–g).

Notably, ID7 lacks signal within reputedly the most informative homopolymer channel (>5 bp). Instead, signals are only present in channels associated with ID1 and ID2 (Fig. 1e), resulting in systematic misattribution of all MMRd gene-edit signatures to ID1 and ID2 (Fig. 1f,g). Moreover, InDel signatures of *POLE*, *POLD1* mutants and all combined polymerase/MMRd edits were indistinguishable from ID1 using COSMIC-83 taxonomy and sometimes indistinguishable from each other (Extended Data Fig. 1e). The signature of polymerase mutant *POLD1*[R689W] did not resemble any reported signatures. Because InDel mutagenesis of gene edits occurred predominantly at longer homopolymers and was erroneously assigned to ID1 and/or ID2 (Fig. 1g), we explored whether expanding on the long homopolymer channels and modifying the information presented in individual InDel channels could improve the resolution to disentangle the ostensibly alike but distinct biological signatures without compromising the power for signature extraction.

### A new framework for classifying InDels

As with substitutions, incorporating surrounding sequence characteristics may enhance the discriminatory capacity of InDel catalogs for

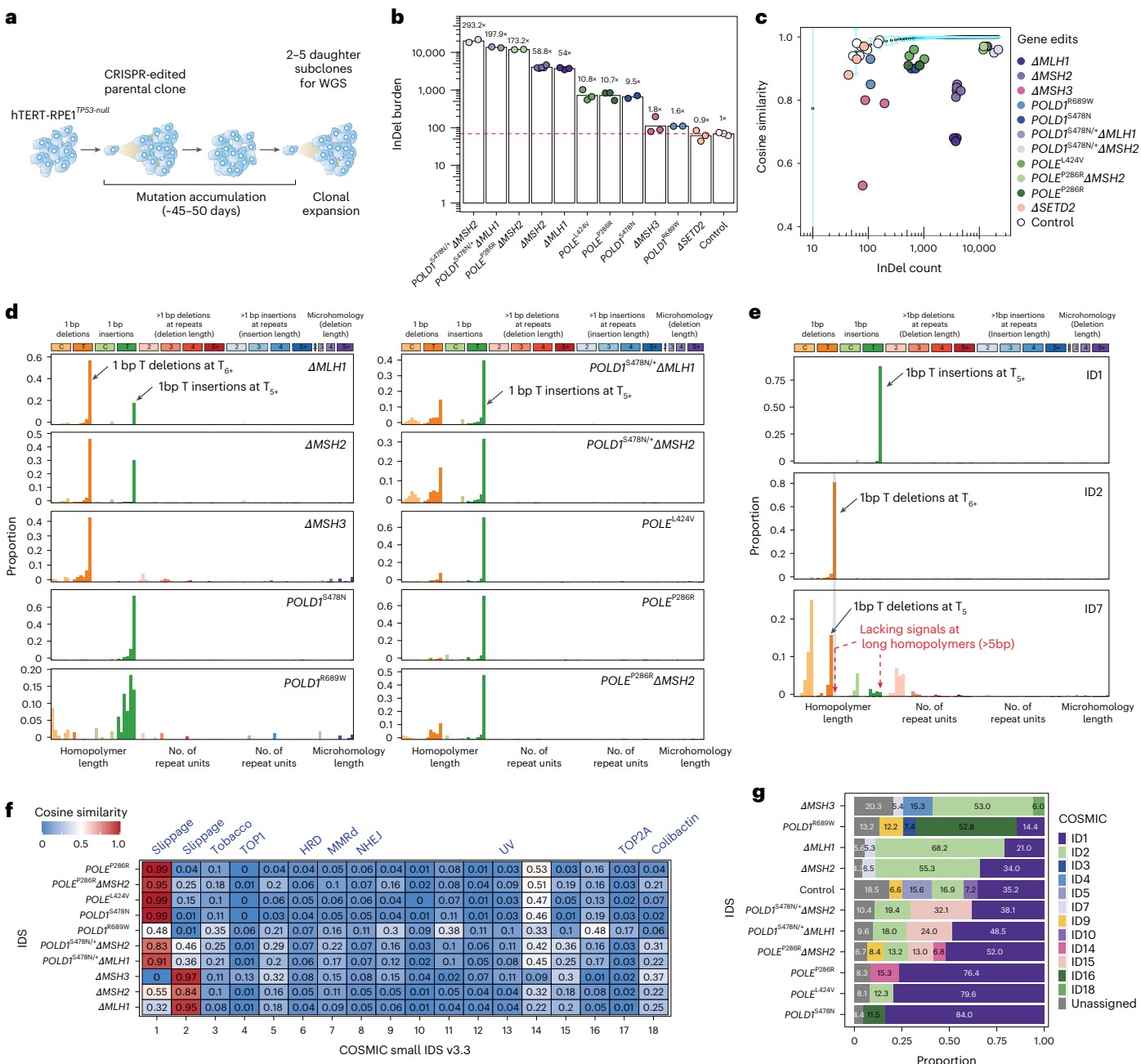

**Fig. 1 | Isogenic PRRd human cell lines exhibit distinct InDel patterns.**
**a**, Mutation accumulation experiment in *TP53*-null hTERT-immortalized retinal pigment epithelial cell (hTERT-RPE1$^{TP53-null}$, herewith referred to as the background control). **b**, InDel burden and average InDel fold increase of CRISPR gene edits (n = 2–5 subclones per genotype; Supplementary Tables 1–3). Red dashed line represents the mean InDel burden of control subclones. The *y* axis shows the InDel burden in log scale. **c**, Distinguishing COSMIC-83 InDel profiles of edited subclones from background control. Light blue error bars depict the mean ±3 s.d. of cosine similarities between n = 100 bootstrapped InDel profiles of

unedited controls and the background profile (Extended Data Fig. 1d) aggregated from n = 7 unedited subclones. The *x* axis shows the InDel count in log scale. **d**, COSMIC-83 InDel mutational signatures associated with gene edits following background subtraction (Supplementary Table 4). **e**, Key features of COSMIC ID1, ID2 and ID7 (v.3.3). **f**, Heatmap of cosine similarities between gene-edit IDS and COSMIC IDS (v.3.3). Known, proposed etiologies are annotated above the heatmap (blue). **g**, Decomposed solution of gene-edit InDel signatures in **d** into COSMIC IDS (v.3.3).

signature analyses[32]. We first classified InDels according to whether they were insertions, deletions or complex InDels (simultaneous insertions and deletions; Fig. 2a). Within insertions and deletions, InDels were subclassified by motif size (1 bp versus ≥2 bp). For 1 bp InDels, we considered the nucleotide content (C/G versus A/T motifs), the 5′ and 3′ flanking bases and the length of homopolymeric tracts. For InDels ≥2 bp, we identified the maximally repetitive motif within the InDel and accounted for its repeat length in the 3′ sequence (Supplementary

Note 1). For deletions occurring with microhomology at the InDel junction, we considered the deletion motif length (*L*) and the microhomology length (*M*). This comprehensive taxonomy yielded 476 nonoverlapping InDel subcategories (channels; Supplementary Table 7 and Supplementary Note).

We examined whether all 476 channels were informative. By analyzing the InDel distribution across all channels in 18,522 tumors covering most cancer types from the International Cancer Genome

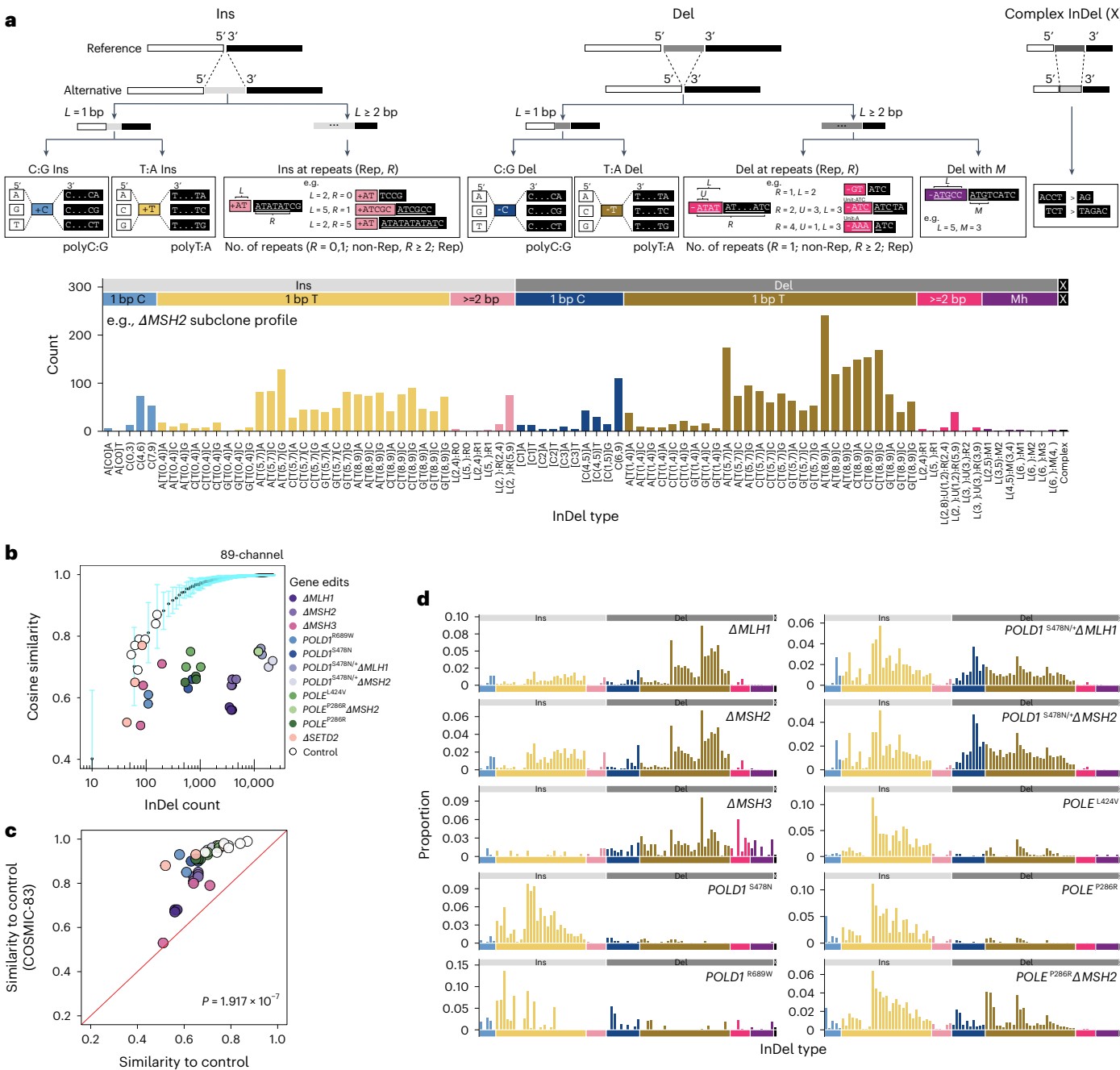

**Fig. 2 | Redefined InDel taxonomy improves discriminatory power and reveals differential InDel patterns associated with PRR gene edits. a**, Proposed InDel classification schema and an example 89-channel InDel profile of an *ΔMSH2* subclone. **b**, Distinguishing 89-channel InDel profiles of edited subclones from background control. Light blue error bars depict the mean ± 3 s.d. of cosine similarities between *n* = 100 bootstrapped InDel profiles of unedited controls and the background profile (Extended Data Fig. 5b) aggregated from *n* = 7 unedited subclones. The *x* axis shows the InDel count in log scale. **c**, Cosine similarities of edited subclones and bootstrapped controls in COSMIC-83 InDel profiles against 89-channel InDel profiles. Two-tailed Wilcoxon signed-rank test, *P* = 1.917 × 10⁻⁷. **d**, The 89-channel InDel mutational signatures associated with PRRd gene edits following background subtraction (Supplementary Table 4; https://signal.mutationalsignatures.com/explore/main/experimental/experiments?study=7). Ins, insertion; Del, deletion.

Consortium (ICGC)/The Cancer Genome Atlas (TCGA)[33], Hartwig[34] and the Genomics England (GEL) 100,000 Genomes Project[35] (Extended Data Fig. 3e), we identified noninformative channels (that is, channels with no signal) and consolidated those with low signal to reduce the total number of InDel channels to 89 (Fig. 2a, Extended Data Fig. 3f and Supplementary Table 8). Overall, compared to COSMIC-83, the 89-channel taxonomy expands upon channels that had most of the signals, here 1 bp A/T InDels, into a larger array of channels, and condenses longer InDels and/or genome motifs infrequent in the

genome (where signals were scant or nonexistent) into fewer InDel subcategories (Extended Data Fig. 4). Although the final numbers are not vastly different between the two classification systems, our data-driven approach, incorporating sequence contexts and enhancing signal distribution of mononucleotide/polynucleotide repeat tracts into additional channels, provides alternative information to the mutational signature extraction and assignment process, potentially increasing the likelihood of detecting new biologically meaningful signatures.

To test this, we applied the new 89-channel InDel taxonomy to our ground truth gene-edit dataset (Supplementary Table 4). Cosine similarities between experimental InDel profiles and control were much lower with the 89-channel format than with COSMIC-83 (Fig. 2b,c and Extended Data Fig. 5a,b), indicating that the new classification improved separation of gene edits from the background (mean cosine similarity, 0.68 ± 0.08 for 89-channel versus 0.89 ± 0.11 for COSMIC-83; two-tailed Wilcoxon signed-rank test, $P = 1.917 \times 10^{-7}$). We subsequently determined signatures associated with each gene edit using the 89-channel format. The resultant signatures showed more evenly distributed signals across the entire 89-channel profile (Figs. 1d, 2d and Extended Data Fig. 5c). Gene-edit signatures were also more readily discernible from one another (mean signature pairwise cosine similarity, 0.57 ± 0.25 for 89-channel versus 0.64 ± 0.3 for COSMIC-83; two-tailed Wilcoxon signed-rank test, $P = 1.483 \times 10^{-5}$; Extended Data Fig. 5d,e). Notably, InDel signatures of combined MMRd/polymerase mutants were not simply the sum of the individual mutational processes, likely reflecting the biological interactions of Pol ε and Pol δ with MMR in suppressing InDel formation during the replication of repetitive DNA.

Interestingly, we noted that while MMRd deletions are particularly amplified at longer homopolymers (8–9 bp > 5–7 bp > 0–4 bp), polymerase mutants displayed strikingly different distribution of excess insertion mutagenesis at shorter homopolymers (5–7 bp > 8–9 bp > 0–4 bp; Fig. 2d). The higher InDel rates in shorter homopolymers conferred by defective proofreading of Pol ε and Pol δ likely reflect the distance over which they interact with duplex DNA upstream of the polymerase active site[29]. Indeed, crystal structures of Pol ε and Pol δ have shown numerous contacts made within 5–7 bp of the polymerase active sites with duplex DNA[36,37], with experimental model reinforcing this optimal distance[29], explaining how proofreading may offer reduced protection against InDels outside of this 'footprint' (that is, unpaired bases further upstream of the active site[38]; longer runs where MMR plays a more crucial role[29]). These unique insights were only appreciable due to the new 89-channel format, offering enhanced capturing of biological variation.

To compare the discriminatory capacity of both classification systems, we also performed de novo signature extraction on our ground truth experimental dataset ($n = 37$; Extended Data Fig. 6a). With COSMIC-83, only two de novo signatures were extracted—one dominated by T insertions at poly-T$_{5+}$ (ID83A) and the other by T deletions at poly-T$_{6+}$ (ID83B; Extended Data Fig. 6b). In contrast, the 89-channel format yielded four signatures, matching our expectation of a predominantly deletion-driven MMRd signature (InD89B), a predominantly insertion-driven polymerase signature (InD89D) and two distinct signatures with differing proportions of InDels (InD89A and InD89C), likely reflecting the combined polymerase/MMRd phenotypes (Extended Data Fig. 6c).

Finally, to determine whether this observed relationship between channel information content and signature extraction extended to other datasets and workflows, we applied three different algorithms[3,8,39] to an unrelated cohort of 52 colorectal WGS from ICGC[33] (Extended Data Fig. 6d). All three algorithms failed to discern all available signatures using COSMIC-83, reaching a discrimination limit of five, yielding sparse signatures with signal density highly concentrated in two channels (Extended Data Fig. 6d,e,g). Contrarily, the 89-channel format consistently enabled the detection of more de novo signatures across all algorithms used (Extended Data Fig. 6f,h). The extracted signatures also displayed signals across more channels, highlighting the superior performance of the 89-channel classification over COSMIC-83 in uncovering additional, true mutational processes.

### New InDel signatures (InDs) in seven cancer types

To explore the impact of our new InDel taxonomy on signature discovery beyond PRRd phenotypes in human cancers, we analyzed seven tumor types ($n = 4,775$) known to display clinically relevant high tumor mutational burden (TMB) due to a range of abnormalities (for example, MMRd, environmental ultraviolet (UV) radiation, APOBEC-related mutagenesis)—bladder ($n = 347$), brain (CNS, $n = 392$), colorectal ($n = 2,146$), endometrial ($n = 695$), lung ($n = 958$), stomach ($n = 181$) and skin ($n = 56$) cancers from the GEL 100,000 Genomes Project[35] (Fig. 3a).

We performed mutational signature analysis per tumor type as previously described[3] (Fig. 3a, Extended Data Fig. 7, and Supplementary Tables 9–11; Methods). We identified 37 consensus InDel signatures, referred to as InDS (to distinguish from COSMIC IDS; Fig. 3b). Ten signatures shared characteristics mappable to known IDS (InD1, InD2a, InD3a/InD3b, InD4a, InD6, InD8, InD9a, InD13 and InD18)[7]. The remaining 27 were new.

Exogenous exposures underlie five InDS. InD3a and InD3b often co-occurred in lung cancers with tobacco exposure. InD3a/InD3b clustered with experimental signatures induced by benzo(a)pyrene and its metabolite benzo(a)pyrene diol epoxide (Extended Data Figs. 8 and 9), supporting the notion that they represent modulated versions of tobacco-related DNA damage. InD13, characterized by T deletions at TT dinucleotides, is linked to UV damage, and InD18, found exclusively in colorectal samples, is due to colibactin exposure[40]. InD32 was identified in samples with prior exposure to platinum and was associated with a new platinum-associated signature, SBS112 (ref. 3).

Twenty InDS had probable endogenous origins (Extended Data Fig. 9). Several have been described, including InD1 and InD2a, errors associated with nascent and template strand slippage during normal DNA replication, respectively[7]. InD1 and InD2a were seen universally across all tumor types except CNS and skin cancers, which showed a tissue-specific variant, InD2b (Fig. 3a). InD4a is attributed to TOP1 transcription-associated mutagenesis[14]. InD6, marked by microhomology-mediated deletions, is associated with deficiency in HR repair[7]. InD8, which had deletions with little to no microhomology at deletion junctions, likely reflects the footprint of nonhomologous end-joining activity and/or radiotherapy[41].

InD9a, correlated with SBS2 and SBS13 hypermutation, featured 1 bp C deletions at T$\underline{C}$T and T$\underline{C}$A (mutated base underlined), identical to mutable motifs characteristic of SBS2/SBS13, particularly at short poly-T tracts. It was presumptively induced by APOBEC (Extended Data Fig. 8c), corroborated by experimental evidence from an APOBEC overexpression DT40 model[42]. We proposed a mutagenesis mechanism wherein following C-to-U deamination at T$\underline{C}$T by APOBEC, uracil removal by UNG leaves an uninformative abasic site. Template strand slippage can then occur at this short repetitive T tract, leading to a C deletion (Extended Data Fig. 8d). For reasons currently unclear, we also found similar C-deletion-dominated InD9b/InD9c, which, although resembled InD9a, lacked the predilection for a preceding T, and was possibly caused by an alternative mechanism.

Interestingly, we extracted eight gene-specific MMRd and Pol-dys InDS. MMRd-InD7 contrasts with COSMIC ID7. InD7 is characterized by the expected excess of 1 bp and 2 bp deletions, particularly at longer mononucleotide/dinucleotide repeat tracts. InD7 clustered with experimental signatures of ΔMLH1, ΔMSH2 and ΔMSH6 (Extended Data Fig. 9). We also identified InD19 (due to PMS2 deficiency), InD14 (associated with POLD1 exonuclease mutations), InD15 (associated with POLE exonuclease mutations), InD16a and 16b (resulting from concurrent loss of POLE proofreading and MMR), InD21 (associated with combined POLD1 proofreading defect and MMRd) and InD20, which we found through experimental investigations, was due to MMRd occurring on a POLE dysfunction background.

The remaining 12 signatures were of uncertain etiology. Five were probably artifacts—InD27 and InD28 often co-occurred, incurring thousands of InDels, and were related to SBS57, potentially an amplification or a sequencing artifact[7]. InD28m was likely a mixed signature of InD28 and InD4, remaining to be resolved with larger cohorts. InD5 and InD10 were ubiquitous and possibly artifacts.

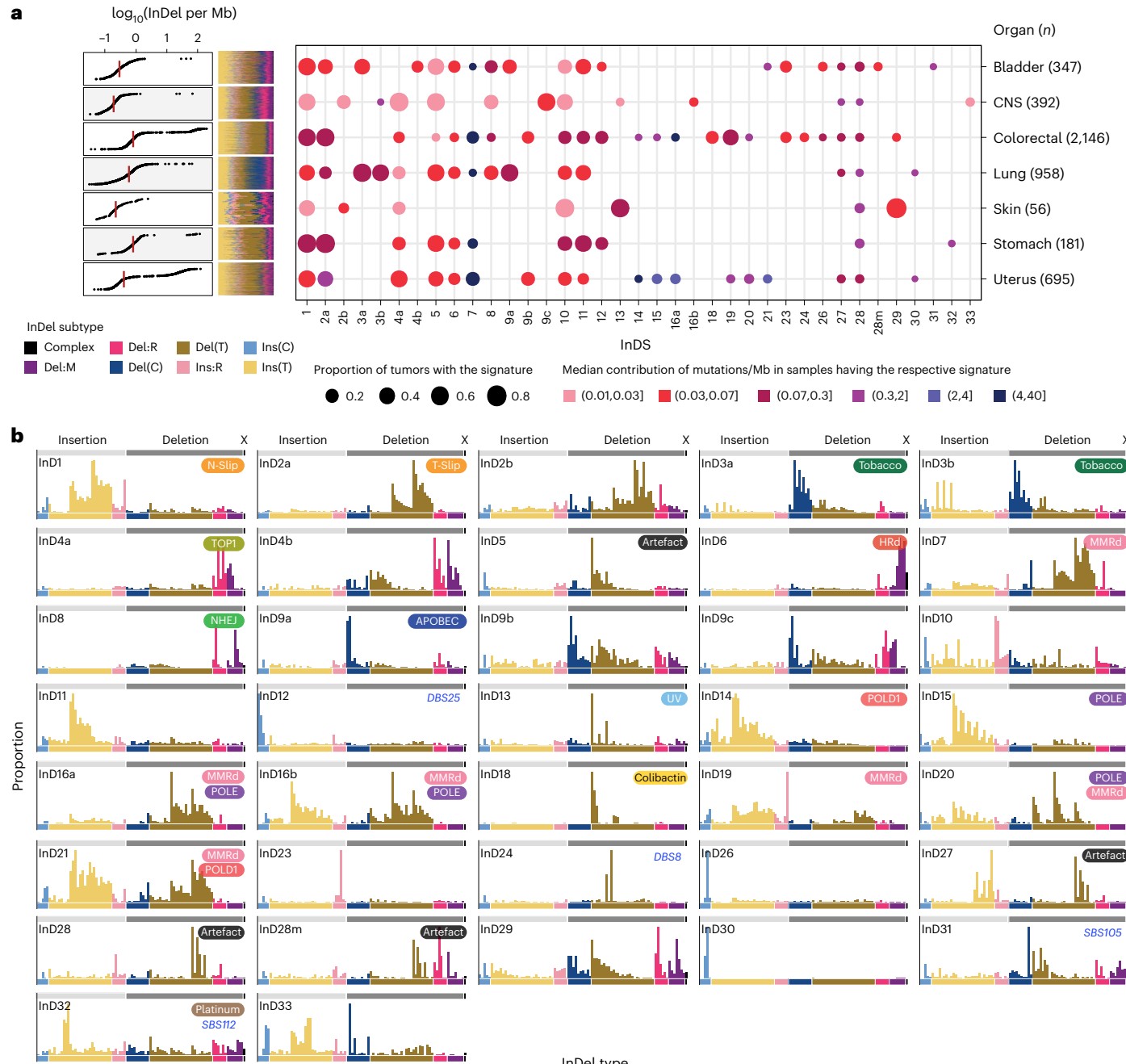

**Fig. 3 | De novo signature extraction using redefined InDel taxonomy uncovers 37 InDS in seven cancer types in the GEL cohort. a**, InDel burden across seven cancer types (*n* = 4,775; left) and the number of mutations contributed by each InD to the GEL tumors. The size of each dot represents the proportion of samples of each tumor type that shows the mutational signature. The color of each dot represents the median mutation burden (per Mb) of the signature in samples that show the signature. **b**, Profiles of 37 consensus InDel mutational signatures (InDS) extracted and curated from seven GEL cancer cohorts (Supplementary Table 10; https://signal.mutationalsignatures.com/explore/main/cancer/signatures?mutationType=3&study=7). Putative etiologies are provided in the top-left squircles. N-Slip, nascent strand slippage; T-Slip, template strand slippage; NHEJ, nonhomologous end joining.

---

While C insertions dominated both InD26 and InD30 at poly-C tracts followed by a 3′A, InD30 C insertions induced thousands of insertions at homopolymers CCC and CCCC, whereas InD26 C insertions mainly occurred at longer CCCCC and were not associated with hypermutation.

Three InDS (InD31, InD24 and InD12) showed striking correlations with signatures of other classes. InD31 displayed distinct C deletions at short homopolymers (<5 bp) followed by 3′G and T deletions at short homopolymers (<5 bp) followed by 3′A. It was only reported in samples with novel rare SBS105 (ref. 3), and often co-occurred with InD8.

InD24 deletions peaked prominently at GTA and GTG and were strongly correlated with DBS8, which shows double substitutions at the same motifs (TGTG > TAGG/TTGG). InD12 exhibited C deletions between dinucleotides AA and AT and was associated with DBS25 featuring a tall peak at TT dinucleotide. Despite clear co-occurrence, the causes for these signatures remain cryptic.

InD4b and InD29 shared features with InD4a and InD8, respectively. Whether they represented tissue-specific variants, were mixed or caused by different mechanisms requires further investigation. InD11 appeared related to InD1 and might be an oversplit signature frequently

enriched in high InDel burden samples, such as those with MMRd and Pol-dys. Seen in bladder and colorectal cancers, InD23 showed a striking pattern of longer insertions (≥5 bp) at nonrepeats. These insertions were almost exclusively tandemly duplicated from immediate neighboring sequences. InD33 was most prominent in one CNS tumor treated with temozolomide; however, its etiology remains unknown.

In summary, 5 InDS were of likely exogenous origins (InD3a, InD3b, InD13, InD18 and InD32), 20 were endogenous (InD1, InD2a, InD2b, InD4a, InD4b, InD6, InD7, InD8, InD9a, InD9b, InD9c, InD11, InD14, InD15, InD16a, InD16b, InD19, InD20, InD21 and InD29) and 12 had uncertain sources (InD5, InD10, InD12, InD23, InD24, InD26, InD27, InD28, InD28m, InD30, InD31 and InD33).

### A signature-based classifier of PRR dysfunction

PRRd subtypes, typified by MSI, are clinically actionable with potential selective sensitivity to immunotherapies[20–22]. Current methods of detecting PRRd mainly rely on immunohistochemistry (IHC) staining of MMR proteins (but not for polymerase mutants) and/or PCR-based assays to determine MSI at selected genomic loci. These assays are not sensitive or robust enough, especially in nonepithelial tissues[16]. Using insights from this study, we therefore explored constructing a classifier for tumor PRRd stratification, reporting MMRd, Pol-dys and mixed MMRd/Pol-dys as distinct classes versus PRR proficiency.

We used 571 GEL cancers assigned as MMRd ($n = 214$), Pol-dys ($n = 36$), mixed MMRd/Pol-dys ($n = 41$) or PRR-proficient (controls, $n = 280$) based on confirmed causal genotypes and allelic status, and/or supporting IHC staining (Fig. 4a and Supplementary Table 12). Samples treated as controls had neither MMRd and/or Pol-dys confirmed through the lack of driver mutations in key MMR genes (that is, *MLH1*, *MSH2*, *MSH6*, and *PMS2*), *POLE*, *POLD1*, and displayed no evidence of MSI associated with these abnormalities[43]. We trained multiple multinomial elastic net regression models applying 7:3 partitioning iteratively across the dataset. Through exploring all possible features/models (Supplementary Table 13), we identified exposures of SBS and InDS associated with MMRd, Pol-dys and mixed MMRd/Pol-dys, as well as the ratio of total InDels to substitutions as the most predictive features (Fig. 4b and Supplementary Table 14; Methods). The final model, termed PRRDetect (postreplicative repair detect), was retrained on the entire dataset ($n = 571$). Then, in an independent validation cohort of 504 ICGC breast cancers[44,45] and 847 GEL cancers, for which the true labels of PRRd were known, PRRDetect achieved an AUROC (area under the ROC curve) of 1 and an AUPRC (precision–recall curve) of 0.99 at distinguishing PRR-dysfunctional from PRR-proficient samples, performing superiorly to other MSI/MMRd detection tools, including MSIseq[43], MMRDetect[46] and TMB—an approved biomarker for immunotherapies[20,21,47–49] (Fig. 4c,d and Supplementary Tables 12, 15 and 16).

Next, to survey the prevalence of PRRd across alternative cancer cohorts, we applied PRRDetect on seven cancer types commonly enriched with hypermutator samples from ICGC[33] and Hartwig[34] (Fig. 4c,e,f, Extended Data Fig. 10a,b and Supplementary Table 17). PRRDetect predicted 3.7% (50/1,335) samples as PRR-dysfunctional, correctly identifying all Pol-dys, MMRd/Pol-dys samples and missing two subclonal MMRd samples (based on available published driver information for PRRd status). MSIseq missed 6 of 43 PRRDetect-predicted MMRd, 2 of 4 mixed MMRd/Pol-dys cases while displaying poor concordance for detecting pure Pol-dys cases (that is, missed all 7 cases). Unsurprisingly, PRRDetect captured all MMRDetect-positive cases. However, MMRDetect failed to identify all PRRd cases as it was not designed to detect Pol-dys/mixed phenotypes and missed seven MMRd samples. Crucially, we noted that many PRRDetect-positive cases did not have an associated driver mutation identified (33/50). This is clinically significant. Of 50 PRRDetect-positive cases, 39 were MMRd (only 8 had an associated driver mutation), 7 were Pol-dys (all had driver mutations in polymerase proofreading domains) and 4 were predicted as mixed MMRd/Pol-dys (2 had *POLE* exonuclease mutations and none had MMR drivers). If PRRDetect predictions were all true and sequencing approaches focused exclusively on identifying driver events associated with these deficiencies were used, a significant proportion of cases (66%) could be missed.

Given that PRRd cancers often present with high TMB, and TMB is used as a biomarker for immunotherapies, we explored the limits of TMB-based patient stratification. With an FDA-approved TMB cutoff of 10 mutations per Mb[49], just over a tenth of 459 cases classified as TMB-high (50/459, 10.9%) had predicted PRR dysfunction (Fig. 4f, Extended Data Fig. 10b and Supplementary Table 17). The majority of other cases (353/459, 76.9%) had high TMB from tobacco, UV and APOBEC exposure; 56 (12.2%) were due to alternative causes. Thus, across independent cancer cohorts where MMRd and Pol-dys are known to occur at higher frequencies, ~89% of the samples classified as TMB-high may not have the intrinsic biological underpinnings associated with response to immunotherapies, with implications for the use of TMB as a selective biomarker for ICI[50,51].

We asked whether this trend extended to the larger GEL cohort ($n = 4,775$). Among the 1,371 TMB-high cases, nearly half (677, 49.4%) were predicted as having MMRd and/or Pol-dys (Fig. 4g), of which, only ~50% of them had an identified driver. The remaining 564 (41.1%) had high TMB due to alternative mutagenic exposures; 130 (9.5%) were due to other undetermined causes. Furthermore, beyond revealing PRR dysfunction in typical tumor types such as colorectal cancers (19%, 400/2,146) and uterine cancers (37%, 255/695), PRRDetect predicted PRRd in a small but notable proportion of stomach (11/181, 6%), bladder (3/347, 1%), CNS (3/392, 1%) and lung cancers (8/958, 1%; Extended Data Fig. 10c and Supplementary Table 12). This reinforces two important clinical points—first, PRRd is not restricted to colorectal and uterine cancers despite being more prevalent in these tumor types; second, WGS can serve as a tumor-agnostic assay uncovering PRRd and any other actionable biological abnormalities in the future.

## Discussion

The ability to distill biologically relevant signatures is heavily influenced by how mutations are represented or classified, more so than the underlying algorithms used for signature extraction. Here we showed that a classification schema that aggregates potentially discriminatory signals into only a few channels and/or does not take surrounding sequence context into account is limited in its ability to discern biologically insightful InDel patterns, irrespective of the extraction algorithms used. Consequently, some of the currently reported InDel signatures may correspond to multiple mutational processes, affecting the specificity of their assignments. To overcome this limitation, we proposed an alternative InDel taxonomy that incorporates flanking sequence context and distributes signals across a broader set of channels, offering increased discriminative capability without sacrificing power for signature extraction. Using this framework, we captured the distinct MSI phenotypes and true biological diversity of PRRd-associated InDel patterns, evident in both isogenic cellular models and patient tumors. Indeed, these InDel signatures mirror the diverse PRRd-associated SBS signatures in cancers[3,7,31].

Furthermore, we deciphered 37 consensus InDS from seven cancer types. We confirmed ten previously described IDS[7,14,42], including those associated with tobacco use, UV exposure and APOBEC activity and reported eight new InDS of MMRd and polymerase proofreading dysfunction. While we have offered putative causes and associations for several new signatures, our current understanding of InDel mutagenesis remains incomplete. Future studies incorporating more cancer types and/or larger sample cohorts will help uncover additional signatures and illuminate new etiologies. The possibility of adapting the taxonomy in the future, to include features currently not explorable due to the limitations incurred by technological error rates of calling InDels using short-read WGS (that is, at longer simple repeats), could also be revealing.

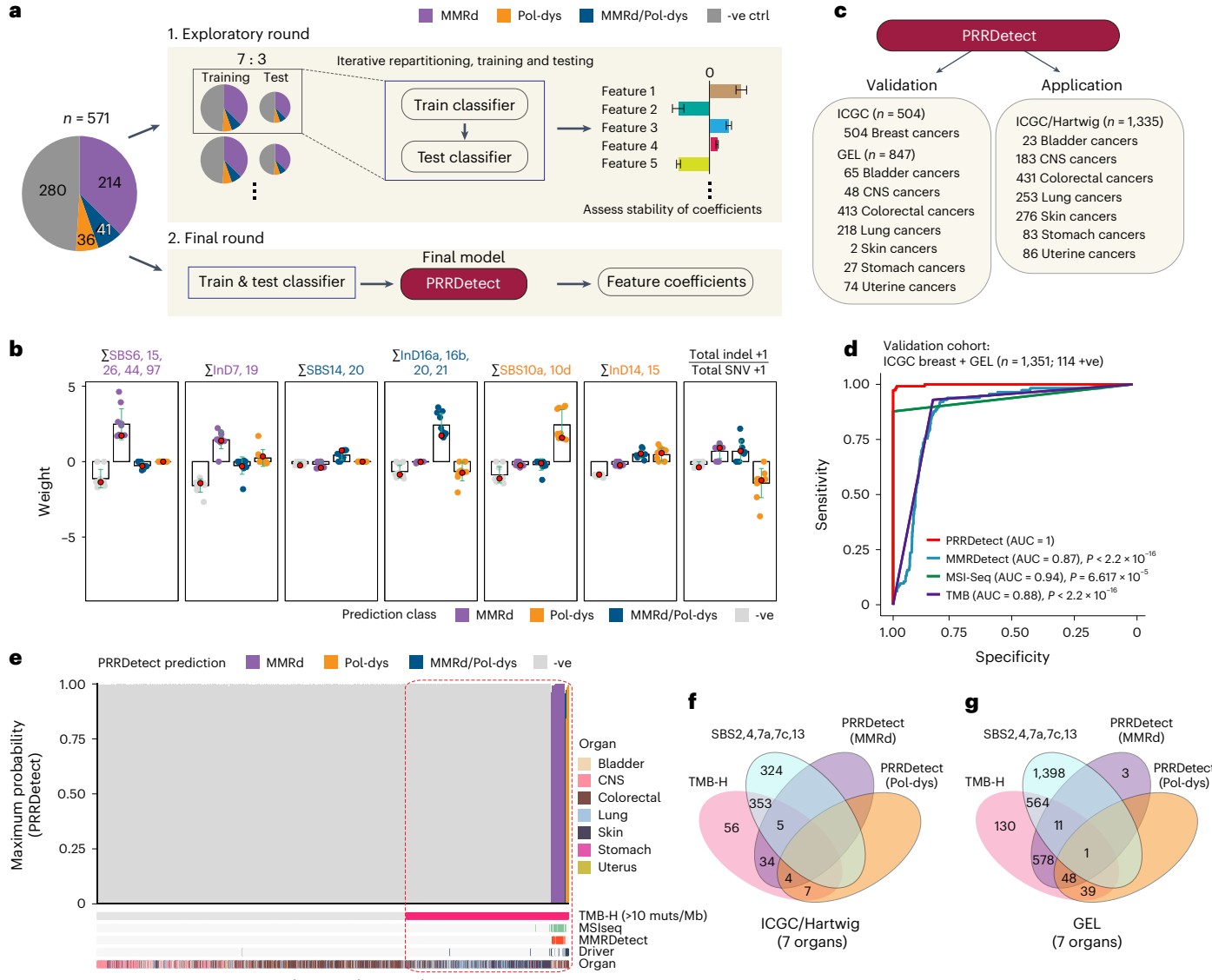

**Fig. 4 | PRRDetect improves the detection of tumors with PRR dysfunction.**
**a**, Simplified workflow of the development of PRRDetect classifier. (1) Initial exploratory training using 571 ground truth samples. (2) Final retraining to produce the PRRDetect classifier. -ve ctrl, negative control. **b**, Distribution of coefficients across seven genomic features contributing to the final PRRDetect classifier. Green error bars depict the mean ± s.d. from ten replicates of training in cross-validation. Red dots indicate the final coefficients chosen for each class prediction (Supplementary Table 14). **c**, Validation and application of PRRDetect on independent cancer cohorts. **d**, ROC curves demonstrating the superior performances of PRRDetect on independent cancer cohort ($n = 1,351$) against alternative biomarker strategies. $P$ values were calculated using two-sided nonparametric test based on the bootstrap distribution (10,000) of the difference in AUCs[53]. MMRDetect, $P < 2.2 \times 10^{-16}$; MSIseq, $P = 6.617 \times 10^{-15}$;

TMB, $P < 2.2 \times 10^{-16}$. **e**, PRRDetect results of $n = 1,335$ ICGC and Hartwig cancers, ordered from the lowest to the highest prediction probability across the $x$ axis (left to right) for MMRd (purple), combined MMRd/Pol-dys (blue) and Pol-dys samples (orange). Negative samples were ordered by TMB in increasing order from left to right. Results of MSIseq, MMRDetect, cancer gene driver annotation and cancer tissue origin are labeled at the bottom tracks. Dashed rectangle highlights the extent of false positive overcalling if using TMB > 10 mutations per Mb as a cutoff. **f**, Concordance of calls among TMB-high (>10 mutations per Mb), positive exposure to SBS signatures that impart hypermutation and PRRDetect prediction across $n = 1,335$ ICGC and Hartwig cancers. **g**, Concordance of calls among TMB-high (>10 mutations per Mb), positive exposure to SBS signatures that impart hypermutation and PRRDetect prediction across $n = 4,775$ GEL tumors. muts, mutations.

Our classifier, PRRDetect, is highly sensitive and specific. It utilizes both SBS and InD signatures to stratify tumors by PRRd subtypes and, to our knowledge, is the only tool with this capability. Importantly, we found a lack of concordance between the current biomarkers of MSI/MMRd and the true biological state. Particularly, TMB, despite being FDA approved, is nonspecific[50,51]. This has profound clinical implications as more than 50% of TMB-high (>10 mutations per Mb) cancers arise from biological abnormalities and environmental exposures that have no substantiated biological basis for immunotherapies, potentially impacting patient outcomes. PRRDetect can also detect

samples that have signatures of PRR deficiency but for which no drivers can be detected (nearly 50% of all PRRd cases). A limited sequencing assay would have simply missed a substantial portion of these tumors. Finally, our classifier does not distinguish between MMRd genotypes despite clear differences between, for example, *MLH1*, *MSH2*, *MSH6* and *PMS2*; currently, there is no clinical indication to do so. However, should it become clinically important to distinguish between these genes, it shall be possible to do so.

In summary, our study highlights how mutation classification directly impacts the accuracy of signature analysis. Our decision to

leverage the surrounding sequence context for classifying InDels stems from mechanistic work demonstrating the relationship between InDel formation and flanking 3′ and 5′ sequences[32,52]. Nevertheless, optimal classification remains an active research area. Alternative schema could unveil additional mutational processes in the future. Unraveling the landscape of InDel mutagenesis through the refined framework described here will hopefully translate into meaningful benefits for cancer patients.

## Online content

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

## Methods

The experiments described herein did not require approval from a specific ethics board.

### Cell lines

All cell line models generated and used in this study are provided in Supplementary Table 1. All cells were grown in DMEM/F12 medium (Gibco/Thermo Fisher Scientific) supplemented with 10% FBS, at 37 °C and 5% $CO_2$ in a humidified incubator. The original hTERT-RPE1 $\Delta TP53$ cell was generated from a previous study[23]. To generate the remaining isogenic CRISPR-edited cell lines, 200k RPE1 $\Delta TP53$ cells per each edit were electroporated with preformed ribonucleoprotein complex (RNP with final concentrations of 120 pmol gRNAs and 100 pmol Alt-R Cas9) in supplemented SE buffer, using nucleocuvette and AMAXA 4D-Nucleofector (Lonza) on program EH-158 according to the manufacturer's instructions. Following electroporation, cells were replated into fully supplemented DMEM/F12 medium to recover for 48 h. For knock-ins, homology-directed repair (HDR) donor oligos were supplied with the RNP for electroporation, and cells were replated for recovery in medium spiked in with final concentrations of 2 μM M3814 (Selleckchem) and 0.5 μM Alt-R HDR Enhancer (Integrated DNA Technologies) for the first 24 h. All cells were then cultured for additional 2 to 4 days to allow for gene editing and eventually subjected to limiting dilution on 96-w plates to isolate single-cell clones. Targeted clones were screened for successful frameshift edits for knockouts or incorporation of intended missense mutations for knock-ins via Sanger sequencing. All gRNAs, sequencing primers and genotypes of cell lines generated in this study are summarized in Supplementary Tables 1 and 2.

### Mutation accumulation experiment and sequencing

Double mutants were harder to establish but had similar doubling time to single mutants. Edited clones were cultured for 45–55 days (~40 to 50 doublings) to allow for mutation accumulation before a second round of single-cell limiting dilution was performed to isolate two to three daughter subclones per edit genotype, providing a bottleneck to capture mutations that had occurred since the isolation of the initial edited parental clones.

Genomic DNA was isolated from all samples using Quick-DNA Miniprep Plus Kit (ZymoResearch) following the manufacturer's protocol. WGS libraries were prepared and sequenced (150 bp, paired end, 25×) on the Illumina NovaSeq 6000 platform by Novogene.

### Somatic variant calling

Short reads were aligned to GRCh38/hg38 using BWA-MEM 0.7.17-r1188. Substitutions and InDels in genetic reference clones and daughter subclones were called as previously described[46]. Postprocessing filters were applied to improve the specificity of mutation-calling. Specifically, for single-nucleotide variant (SNV) calls by CaVEMan[54] (v.1.13.15), we used CLPM == 0 and ASMD ≥ 140. To reduce false positive calls by Pindel[55] (v.3.2.0), we used QUAL ≥ 250 and REP < 10. Cell clones with an average variant allele frequency of <0.4 were designated as polyclonal and excluded from all subsequent quantitative analyses (that is, estimation of mutation burden). A variant allele frequency filter of 0.2 was applied to substitutions and InDels. De novo substitutions and InDels in subclones were obtained by subtracting from respective parental clones whenever available, or by removing mutations shared among subclones. De novo mutation counts are provided in Supplementary Table 3. Rearrangements were not analyzed as they were too few to be informative.

### Mutational signature analysis of experimental samples

The derivation of gene edit-associated mutational signatures with cosine similarity, bootstrapping and background subtraction was performed using published framework (https://github.com/xqzou/COMSIG_KO)[46]. In short, we first (1) determined the background mutational signature in background control by aggregating the unedited and untreated subclone mutational profiles, (2) determined the difference between the mutational profiles of the edited clones and background mutation profile and if an edit generates a signature, we (3) removed background mutation profile from the mutation profile of the edited subclone (Supplementary Tables 4 and 5).

We also used Uniform Manifold Approximation and Projection (UMAP)[56] to cluster the InDel profiles. Experimentally derived signatures were compared to published reference signatures[3,7] using signature.tools.lib (v.2.4.4) from https://rdrr.io/github/Nik-Zainal-Group/signature.tools.lib/.

### Replicative strand asymmetry

Replicative strand bias analysis was performed for 1 bp InDels only. InDels were mapped to leading or lagging strand using Repli-seq data (MCF-7) of the ENCODE project[57]. All 1 bp InDels were aligned with pyrimidines as the mutated base, following the convention in the field. This facilitated the identification of the InDel's corresponding strand. IntersectBed[58] in BEDTools (v.2.26.0-114-g4c407ce) was utilized to identify mutations overlapping specific genomic features. To assess a specific mutational signature, the 'expected' ratio of InDels between lagging and leading strands was calculated according to the distribution of the repeats in these regions. The 'observed' ratio of InDels among different strands was determined by mapping InDels to genomic coordinates of all leading/lagging regions. The asymmetry between different strands was quantified by calculating the odds ratio of InDels occurring on one strand (for example, leading) versus the other strand (for example, lagging). $P$ values were computed using binomial test or $X^2$ test and corrected for multiple testing using the Benjamini–Hochberg method.

### De novo extraction of InDel signatures from hTERT-RPE1 samples

De novo signature extraction and decomposition of mutational signatures were performed using SigProfilerExtractor[39] (v.1.1.18), along with SigProfilerMatrixGenerator[59] (v.1.2.4). The recommended default settings (including 500 NMF replicates) were applied (https://github.com/AlexandrovLab/SigProfilerExtractor). Signatures were also extracted using Indel.signature.tools[3] (v.2.4.4) with default settings (20 bootstraps, 500 repeats per bootstrap, matched clustering). The number of InDel signatures selected per channel set was determined by maximal drop in average silhouette width of clustered signatures. For extraction using MuSiCal[8] (https://github.com/parklab/MuSiCal, v.1.0.0), the default hyperparameters included random initialization (init hyperparameter), the minimum volume nonnegative matrix factorization (MVNMF) algorithm (method hyperparameter), 20 MVNMF replicate runs (n_replicates hyperparameter) and between 10,000 and 1,000,000 MVNMF iterations per replicate run (min_iter and max_iter hyperparameters, respectively).

### GEL cohort

WGS data (v.8) for colorectal ($n = 2,146$), endometrial ($n = 695$), stomach ($n = 181$), bladder ($n = 347$), brain (CNS, $n = 392$), lung ($n = 958$) and skin cancers ($n = 56$) were considered in this study, subset from the curated 12,222 sample cohort previously described[3]. Indels were called with Strelka[60] (v.2.4.7) using somatic calling mode.

### InDel segmentation

We exploited the relationship between InDels and their associated 3′ sequence context to determine, for each InDel, the minimal InDel prefix that is maximally repetitive in the 3′ sequence context and within the InDel itself. In brief, for each prefix of the InDel, we identified the maximally repetitive substring in the associated suffix and 3′ context. This applies to all InDel variants that are left-aligned and parsimonious. We termed these partitions of the InDel and 3′ context 'segmentations'

and selected the segmentation that contains the smallest prefix that has maximum repetition in the 3′ sequence context and the InDel. Segmentation produced multiple values quantifying the repetitiveness of the prefix sequence (termed, 'unit'). These values, along with the sequence context, may be used to group InDels into biologically relevant, nonoverlapping InDel subcategories or 'channels'. See Supplementary Note for details.

## InDel channel construction

Using the segmentation values for each InDel, we constructed a set of 476 nonoverlapping InDel channels. By surveying the InDel frequency distribution across all 476 channels in 17,253 tumors covering most cancer types from ICGC/TCGA, Hartwig and the GEL 100,000 Genomes Project, we discarded channels with no signal and consolidated those with low signal to reduce the total number of InDel channels to 89. The final 89-channel set was used for the experimental gene-edit dataset and the cancer WGS from the GEL 100,000 Genome Project considered in this study. Generally, channels may be grouped into six broad divisions—1 bp insertions, 1 bp deletions, ≥2 bp insertions, ≥2 bp deletions, ≥2 bp deletions with evidence of microhomology and complex InDels. Channels were constructed such that each InDel could only be unambiguously assigned to a single channel. A complete description of each channel and exemplar InDels are included in Supplementary Table 8. A description of the reasoning behind channel construction is presented in Supplementary Note.

## InDel signature extraction of GEL cohort

Our approach to signature extraction was motivated by previous study[3]. We observed that hypermutator samples strongly influenced signature extraction using the standard β-divergence NMF model. We sought to filter out hypermutator samples from our initial extraction. For each tissue, we first removed samples with a total InDel burden <100, and then clustered sample profiles according to their cosine similarity (cosine distance, 1 − cosine similarity) using hierarchical clustering with complete linkage. Clusters of similar samples were determined by thresholding the resulting dendrogram such that the average silhouette width was maximized, and within-cluster variation was minimized.

To determine cluster-specific hypermutators, for each cluster, we fit the total burden per sample using a two-component Gaussian mixture model (mixtools[61]) compared to a one-component mixture model, using the Bayesian information criteria for model selection. Hypermutators were defined as the union of samples with a total burden more than third quartile + 1.5 × IQR, where quartiles were calculated over the total dataset, and samples with a greater than 50% probability of being generated by the higher burden Gaussian distribution. Normal and hypermutator clusters were then manually reviewed per tissue, and only normal clusters were used in primary extraction.

Signature extraction was performed per tissue as described, with an increased number of bootstraps (40) and repetitions per bootstrap (1,000), to increase final solution stability.

## Signature refitting and reliably determining excess variation

We sought to determine whether excess variation indicative of rare or unextracted signatures was present in sample catalogs using a published framework[3]. In refitting tissue-specific catalogs, hypermutator samples included, with signatures extracted from nonhypermutator samples using FitMS[3], we observed profiles with lower InDel counts displayed higher degrees of error (as measured by total residual normalized by sample burden). Generally, error decreased logarithmically as InDel burden increased. Therefore, using a single threshold on fitting error to define samples with excess variation excluded a large proportion of samples.

To more accurately calibrate our expectation of fitting error, and therefore, our threshold for detecting excess variation, we performed a parametric bootstrapping procedure to generate a sample-specific expected error distribution. For each sample catalog, we constructed a multinomial distribution using the per-channel density from the normalized reconstructed profile produced by FitMS. Using this distribution, we simulated 10,000 sample profiles with a total burden equal to the sample burden, fit these profiles with FitMS and calculated the resulting error distribution.

Comparing the experimentally derived error distribution to the resulting null distribution allowed us to estimate an empirical P value. This procedure was repeated for all samples in the GEL cohort, and P values were corrected for multiple testing. To control the false discovery rate at 5%, samples with an adjusted P value less than 0.05 were selected for further analysis to determine rare signatures.

## Rare signature extraction and refinement

For each tissue, samples with excess variation were clustered using the residual signal after subtracting out FitMS reconstructed profiles. Hierarchical clustering with average linkage and Euclidean distance resulted in multiple clusters per tissue. A secondary round of signature extraction per cluster was performed using the primary tissue-specific signature set to initialize the signature matrix. An additional one to five rare signatures were determined per cluster, and the number of rare signatures was determined as the minimal value of n, such that rare signatures were not found to perfectly recapitulate cluster members or match common signatures. All extracted rare signatures across a tissue were subject to manual curation to identify recurrent patterns, and the rare signature exemplars that displayed minimal mixing with common signatures were selected. Using this consensus set of common and rare signatures per tissue, all samples in each catalog were refitted using FitMS to determine per-sample signature exposures.

## Deriving consensus IDS

We clustered InDel signatures from seven GEL cancer cohorts (111 tissue-specific InDS) using hierarchical clustering with average linkage and cosine distance and derived a set of consensus signatures following a published framework[3]. The associated tissue-specific signature to reference signature conversion matrix (Supplementary Tables 9–11) was then used to transform tissue-specific signature exposures to consensus signature (final InD) exposures for downstream analyses.

## Annotating InD reference signatures using gene-edit and mutagen exposure signatures

Variants in samples generated in previous studies[46,52] were reclassified and analyzed using our refined InDel classification scheme. Experimental signatures were obtained via background subtraction and determined using a bootstrapping and cosine similarity-based framework previously described[46]. Consensus InDel signatures and all experimentally derived signatures were clustered using hierarchical clustering and cosine distance.

## PRRDetect model

We trained PRRDetect, a multinomial elastic net regression model, on a subset of 571 GEL cancers confidently assigned as MMRd (n = 214), Pol-dys (n = 36), mixed MMRd/Pol-dys (n = 41) or PRR-proficient (negative controls, n = 280) based on manual curation of relevant driver mutations and/or supporting immunohistochemistry where possible. Samples with neither Pol-dys and/or MMRd were confirmed to lack driver mutations in key MMR genes (that is, *MLH1*, *MSH2*, *MSH6* and *PMS2*), *POLE*, *POLD1*, and displayed no evidence of MSI associated with these abnormalities[43].

To create our classifier, we explored a range of feature combinations as model inputs, including (1) summed exposures of SBS and InDel signatures related to PRR deficiency (MMRd, MMRd/Poly-dys,

Poly-dys); (2) feature set (1) combined with TMB; (3) feature set (1) combined with total InDel/SNV ratio; (4) summed exposure of SBS signatures related to PRRd; (5) summed exposure of InD signatures related to PRRd. For each feature set, we constructed the model using either proportion (that is, normalized signature exposure) or the absolute values of the features (that is, raw mutation count contributed by each signature). In total, ten model structures were attempted (five sets of features × two normalizations; Supplementary Table 13).

For all models, the feature values were first $\log_2$ transformed, then $z$-score normalized using the formula $x' = \frac{x-\mu}{\sigma}$. We used the implementation of multinomial elastic net regression (glmnet) in caret (https://topepo.github.io/caret/). In each training iteration, we first partitioned the cohort into 70% for training and 30% for testing, retaining relative proportions of MMRd, Pol-dys, mixed and negative categories across the training and test datasets. A ten-repeat tenfold cross-validation strategy was adopted within the 70% training group. A grid search approach was used to determine the best combination of two hyperparameters (that is, $\alpha$, which acts as a balancing factor between a lasso and a ridge penalty; and $\lambda$, which defines the strength of the penalty), aiming to minimize the log loss. The model was then tested on the held-out 30% test set. We performed the 70:30 repartitioning ten times to assess the stability/robustness of the model (Fig. 4a). We then retrained the model on the entire cohort of 571 samples using a repeated tenfold cross-validation strategy as in the first round to obtain the final feature coefficients for individual models.

To identify the best-performing model, seven measures were defined—training accuracy of the final model, median accuracy on the test sets, s.d. of $\alpha$, s.d. of $\lambda$, median coefficient s.d., median multiclass area under the curve (AUC) on the test set and training multiclass AUC of the final model (Supplementary Table 13). Eventually, PRRDetect was selected as the one having the following input variables: (1) summed exposures to MMRd-associated SBS 6, 15, 26, 44, 97; (2) summed exposures to Pol-dys-associated SBS 10a, 10d; (3) summed exposures to combined MMRd/Pol-dys SBS 14, 20; (4) summed exposures to MMRd-associated InD7, InD19; (5) summed exposures to Pol-dys InD14, InD15; (6) summed exposures to combined MMRd/Pol-dys InD16a, InD16b, InD20, InD21; and (7) total InDel to total SNV ratio, with proportional normalization of the first six features (Supplementary Table 14). The final PRRDetect was retrained using the abovementioned features on the entire curated cohort of 571 samples.

The model outputs a categorical distribution across the four PRRd subclasses (that is, MMRd, Pol-dys, MMRd/Pol-dys or Neg). A sample is PRR-proficient if the probability for the 'negative' class is >0.5; otherwise, the positive class (MMRd, MMRd/Pol-dys or Pol-dys) with the highest probability is assigned.

### PRRDetect in additional cohorts

We validated PRRDetect in an independent ICGC breast cohort ($n = 504$)[44,45] and a subset of held-out samples from GEL ($n = 847$), for which the true PRRd labels were established based on immunohistochemistry staining of four MMR proteins (PMS2, MLH1, MSH2 and MSH6) and driver mutations. The final cohort consists of 1,351 samples, for which we also computed the MSIseq and MMRDetect prediction results. The ROC curves and their relative AUC values were calculated using R package 'pROC'[53]. $P$ values were calculated using the roc.test() function with bootstrap method with 10,000 resamples.

To survey the prevalence of PRRd in other cancer cohorts, we applied PRRDetect to two additional datasets not included in InD signature extraction and PRRDetect training—the ICGC/TGCA pan-cancer dataset[33] and Hartwig Medical Foundation metastatic cancer cohort[34] ($n = 1,335$), focusing on seven cancer types commonly enriched with samples with high InDel burdens. InDels from individual samples in these cohorts were processed to 89-channel profiles as was done for GEL cohort samples. For these datasets, we used published driver annotations as PRRd labels.

### Statistics and reproducibility

All comparisons were between biologically independent samples. No statistical method was used to predetermine sample size. No data were excluded from the analyses. The experiments were not randomized. The investigators were not blinded to allocation during experiments and outcome assessment. Further details are provided in the Reporting Summary.

### Reporting summary

Further information on research design is available in the Nature Portfolio Reporting Summary linked to this article.

## Data availability

Raw sequence files from the hTERT-RPE1 mutation accumulation experiment are deposited at the European Genome-Phenome Archive with accession EGAD50000000209. Mutation calls have been deposited at Mendeley (https://doi.org/10.17632/3k2tpx9ssr.2). RPE1 cells can be obtained directly from the authors. The curated data are available for general browsing from our reference mutational signatures website, Signal (https://signal.mutationalsignatures.com). All other data were previously published[46,52].
Primary data from the 100,000 Genomes Project, which are held in a secure research environment, are available to registered users. See https://www.genomicsengland.co.uk/research for further information or contact research-network@genomicsengland.co.uk. ICGC/TCGA WGS data can be obtained from https://dcc.icgc.org/releases/PCAWG. Hartwig metastasis WGS data can be obtained from Hartwig Medical Foundation through standardized procedures and request forms that can be found at https://www.hartwigmedicalfoundation.nl/en/appyling-for-data/.
Mutagen signatures from human induced pluripotent stem cells (iPS)[52] can be accessed via https://data.mendeley.com/datasets/m7r4msjb4c/2. Human iPS knockout signatures can be obtained directly from https://doi.org/10.1038/s43018-021-00200-0 (ref. 46). The results of RPE1 experimental signatures can be browsed at https://signal.mutationalsignatures.com/explore/main/experimental/experiments?study=7. InD signatures of the seven cancer types are accessible at https://signal.mutationalsignatures.com/explore/main/cancer/signatures?mutationType=3&study=7. Source data are provided with this paper.

## Code availability

The R source code of PRRDetect is available via GitHub at https://github.com/Nik-Zainal-Group/PRRDetect and Zenodo at https://doi.org/10.5281/zenodo.14906103 (ref. 62). InDel segmentation and signature classification script can be accessed via GitHub at https://github.com/Nik-Zainal-Group/indelsig.tools.lib and Zenodo at https://doi.org/10.5281/zenodo.14906117 (ref. 63).

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

## Acknowledgements

G.C.C.K. is supported by the Sir Jeffrey Cheah Foundation. The work performed by S.N.-Z.'s laboratory was funded by the Cancer Research UK (CRUK) Advanced Clinician Scientist Award (C60100/A23916), Dr. Josef Steiner Cancer Research Award 2019, Basser Gray Prime Award 2020, CRUK Pioneer Award (C60100/A23433), CRUK Grand Challenge Awards (C60100/A25274 and CGCATF-2021/100013), CRUK Early Detection Project Award (C60100/A27815) and the National Institute of Health Research (NIHR) Research Professorship (NIHR301627). This work was also supported by the NIHR Cambridge Biomedical Research Centre (BRC-1215-20014). The views expressed are those of the author(s) and not necessarily those of the NIHR or the Department of Health and Social Care. This research was made possible through access to data in the National Genomic Research Library, which is managed by Genomics England Limited (a wholly owned company of the Department of Health and Social Care). The National Genomic Research Library holds data provided by patients and collected by the NHS as part of their care and data collected as part of their participation in research. The National Genomic Research Library is funded by the National Institute for Health Research and NHS England. The Wellcome Trust, Cancer Research UK and the Medical Research Council have also funded research infrastructure. The 100,000 Genomes Project uses data provided by patients and collected by the National Health Service as part of their care and support.

## Author contributions

G.C.C.K., A.S.N., X.Z. and S.N.-Z. conceived the project and designed the experiments. G.C.C.K., S.B., J.Y., C.B. and S.J.Z. performed gene editing and mutation accumulation experiments. G.C.C.K., A.S.N., G.R. and X.Z. designed and implemented computational analyses, with input from A.D., A.M.P., L.C., D.B., L.H., J.D., Y.M., S.S., J.C., M.B. and H.R.D. S.N.-Z. supervised the work. Data interpretation and write-up were provided by G.C.C.K., A.S.N., X.Z., G.R. and S.N.-Z., with inputs from all the other authors. All authors had the opportunity to edit the manuscript and approve the final manuscript.

## Competing interests

A.D., A.S.N., G.C.C.K., G.R., H.R.D., X.Z. and S.N.-Z. hold patents or have submitted applications on clinical algorithms of mutational signatures: MMRDetect (PCT/EP2022/057387), HRDetect (PCT/EP2017/060294), clinical use of signatures (PCT/EP2017/060289), rearrangement signature methods (PCT/EP2017/060279), clinical predictor (PCT/EP2017/060298), hotspots for chromosomal rearrangements (PCT/EP2017/060298), InDel signature methods (PCT/EP2024/077959) and PRRDetect (PCT/EP2024/078030). All other authors declare no competing interests.

## Additional information

**Extended data** is available for this paper at https://doi.org/10.1038/s41588-025-02152-y.

**Correspondence and requests for materials** should be addressed to Serena Nik-Zainal.

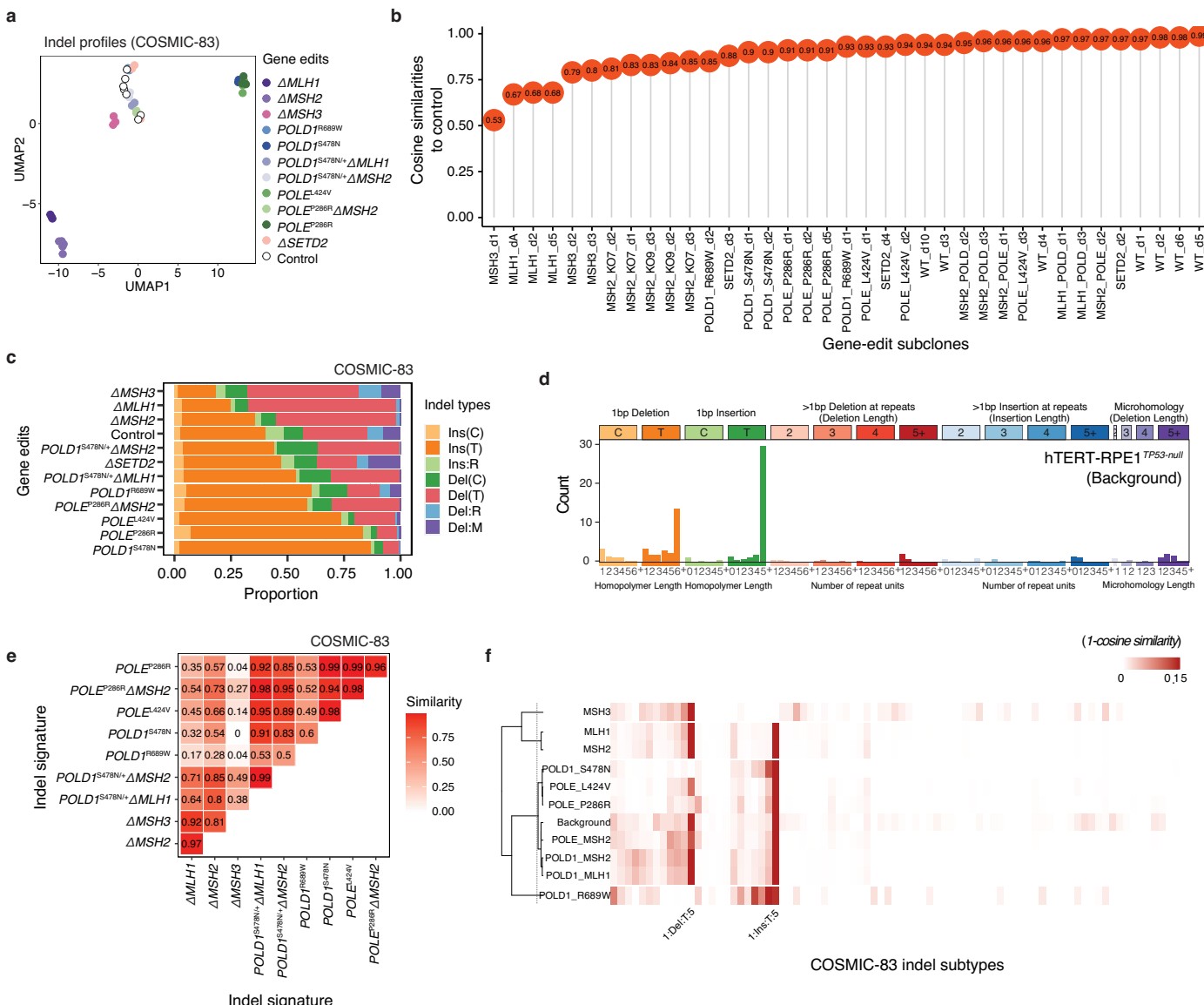

**Extended Data Fig. 1 | COSMIC-83 InDel signatures of PRR deficiency in isogenic human cell line models. a**, UMAP clustering of de novo COSMIC-83 InDel profiles of all gene-edit subclones (*n* = 37). UMAP, uniform manifold approximation and projection for dimension reduction. **b**, Cosine similarities between individual gene-edit subclones and the averaged InDel profile of background controls (**d**). **c**, Relative proportion of seven InDel mutation types of each gene-edit. Ins, insertion; Del. deletion; R, repeat; M, microhomology; C, cytosine; T, thymine. **d**, Aggregated InDel profile of hTERT-RPE1^*TP53-null* background control. **e**, Cosine similarities among experimentally generated COSMIC-83 PRRd InDel signatures (Fig. 1d). **f**, Unsupervised clustering of experimentally generated COSMIC-83 PRRd InDel signatures with cosine distance. Dashed vertical line marks 0.1 cut-off.

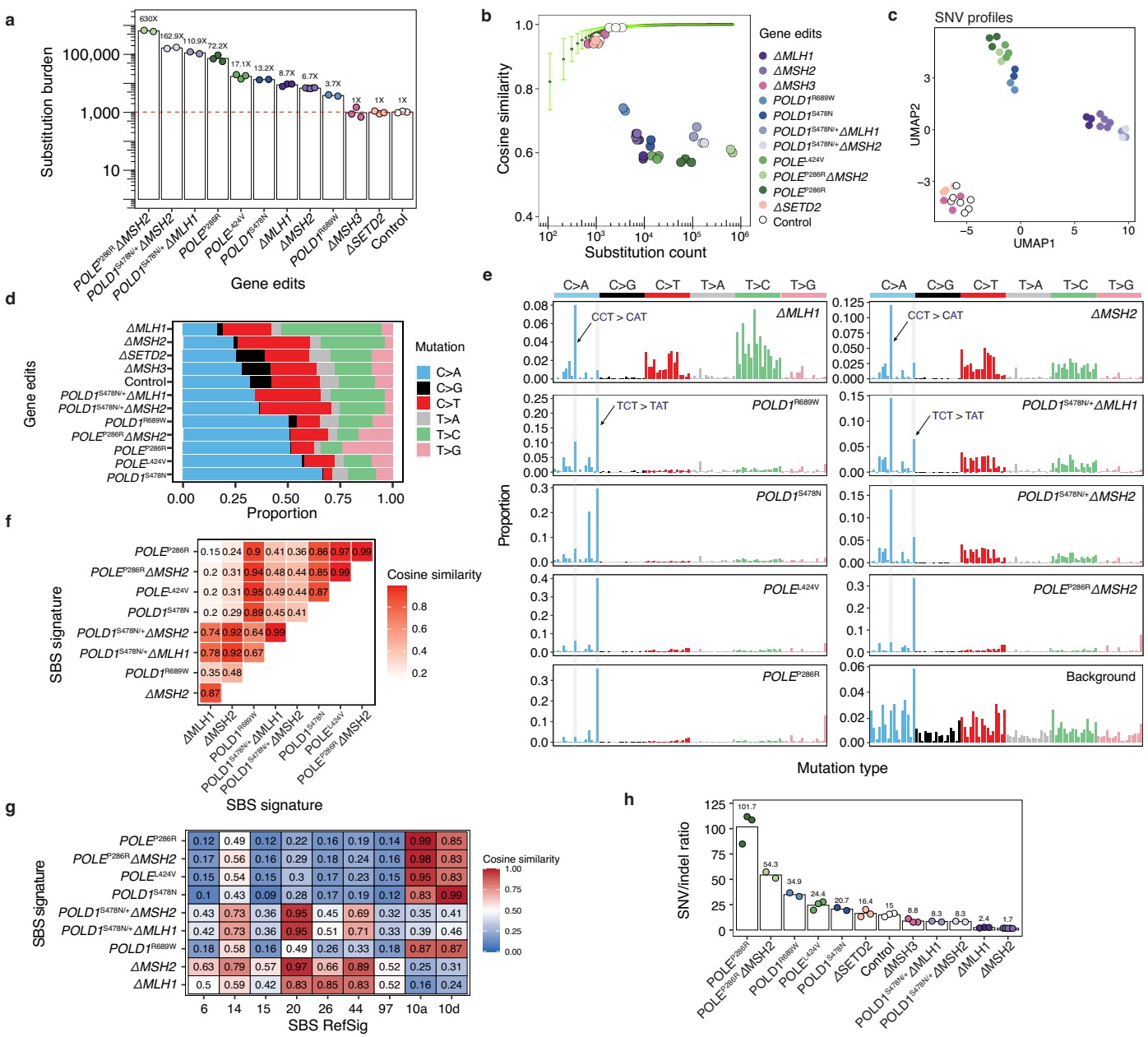

**Extended Data Fig. 2 | Substitution mutational signatures associated with PRR deficiency in isogenic human cell line models. a**, De novo substitution burden, and average substitution fold-increase of CRISPR gene edits and unedited control (*n* = 2–5 subclones per gene edit; Supplementary Table 3). Red dashed line represents the mean substitution burden of control. **b**, Distinguishing substitution profiles of edited subclones from background controls. Light green error bars depict the mean ±3 s.d. of cosine similarities between *n* = 100 bootstrapped substitution profiles of unedited controls and the background profile aggregated from *n* = 7 unedited subclones. The *x* axis shows the substitution count in log scale. **c**, UMAP clustering of de novo substitution profiles of all subclones (*n* = 37). UMAP, uniform manifold approximation and projection for dimension reduction. **d**, Relative proportion of six substitution mutation types of each edit genotype. **e**, Substitution signatures associated with PRRd gene edits following background subtraction. **f**, Cosine similarities among experimentally generated PRRd substitution signatures in **e**. **g**, Cosine similarities between experimentally generated PRRd substitution signatures and cancer-derived SBS reference signatures (RefSig)[3]. **h**, Averaged SNV to InDel ratio of gene edits. Bar represents the averaged ratio for each genotype.

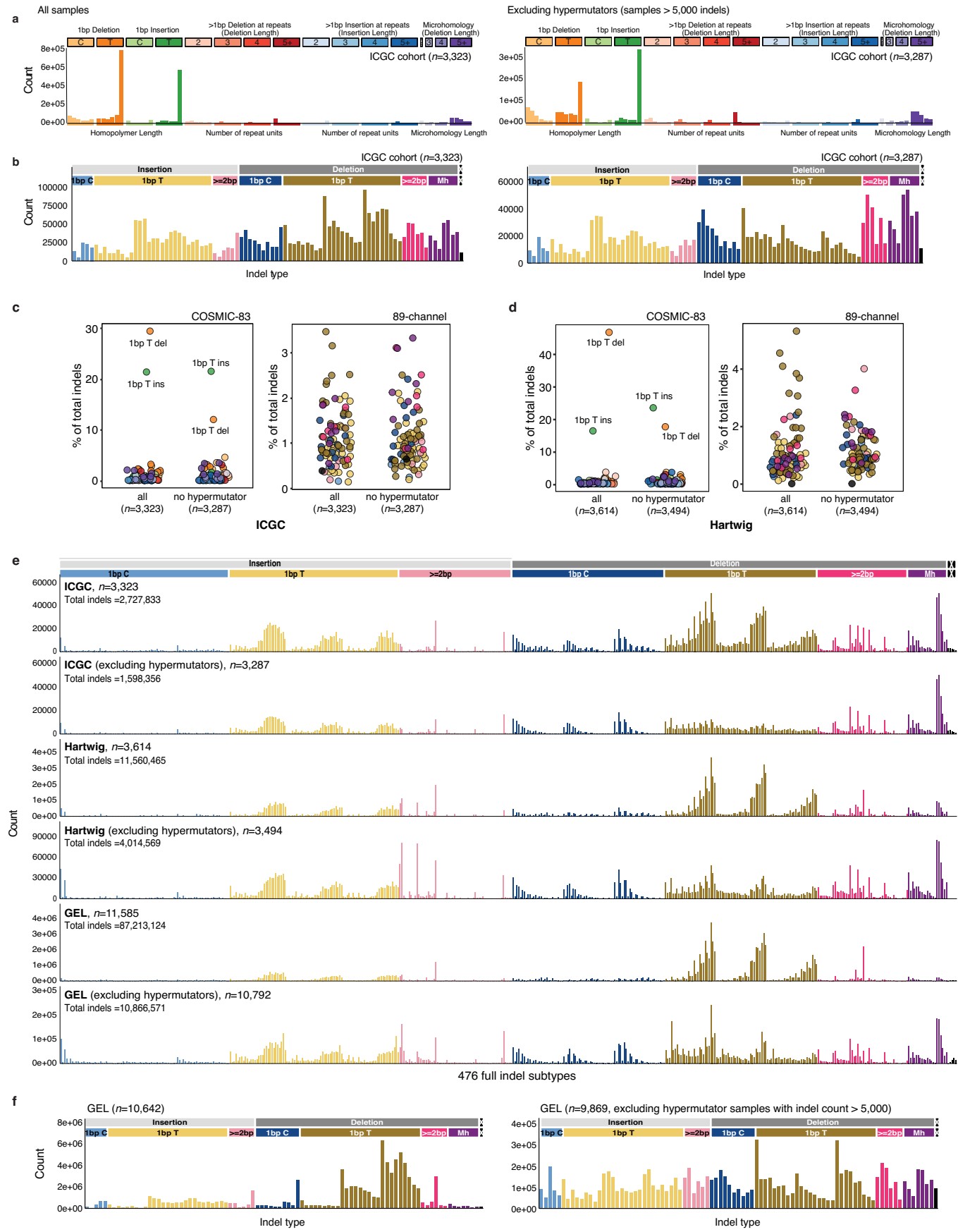

**Extended Data Fig. 3 | See next page for caption.**

**Extended Data Fig. 3 | InDel frequency distribution across pan-cancer datasets and the consolidation of 476 full InDel channels into a redefined 89-channel InDel classification framework. a**, Aggregated COSMIC-83 InDel profile of ICGC samples, with (left, $n$ = 3,323) and without (right, $n$ = 3,287) hypermutator samples. **b**, Aggregated 89-channel InDel profile of ICGC samples, with (left, $n$ = 3,323) and without (right, $n$ = 3,287) hypermutator samples. **c**, Percentage of total InDels per channel for ICGC cohort, with COSMIC-83 (left) or 89-channel (right) format. **d**, Percentage of total InDels per channel for Hartwig cohort, with COSMIC-83 (left) or 89-channel (right) format. **e**, Aggregated 476-full-channel InDel profiles of ICGC, Hartwig and GEL cohorts. **f**, Final, consolidated 89-channel InDel profiles of aggregated samples from GEL cohort, with (left, $n$ = 11,585) and without (right, $n$ = 10,792) hypermutator samples.

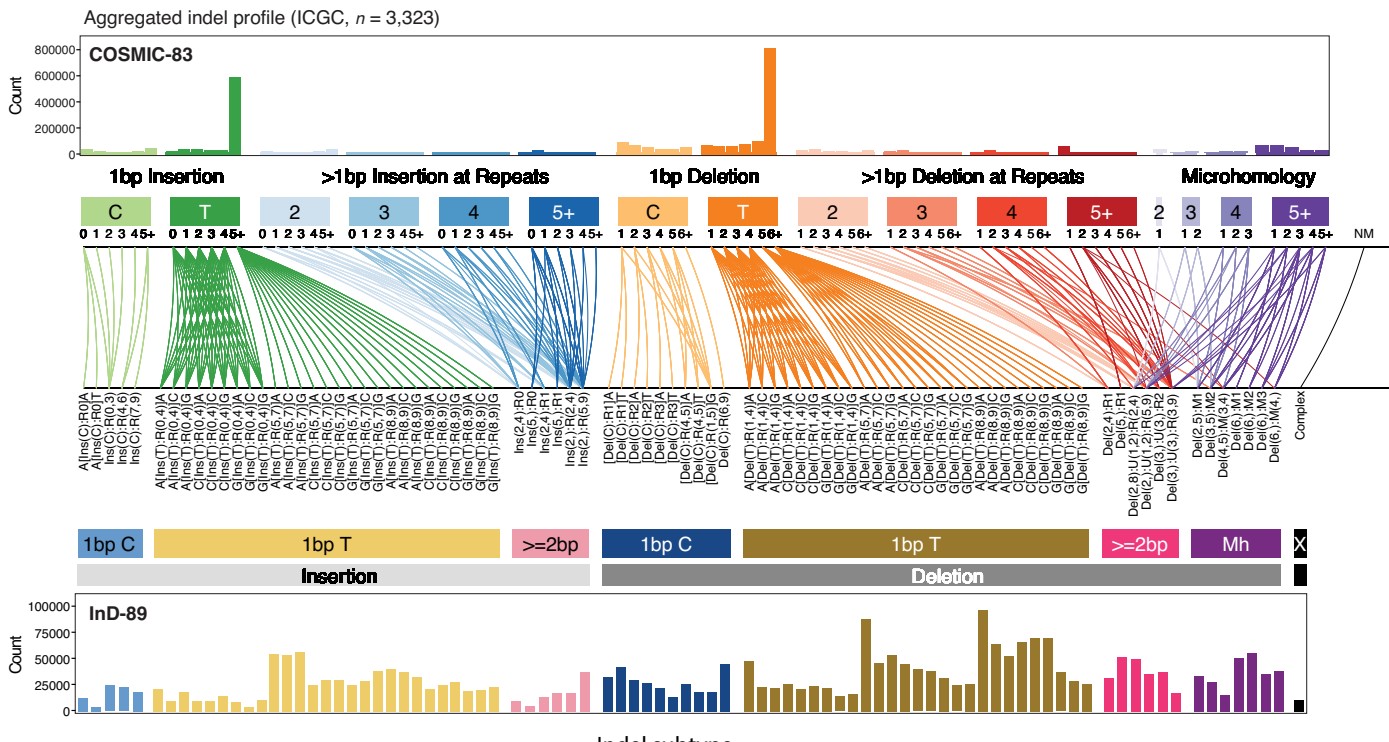

**Extended Data Fig. 4 | Channel mapping between COSMIC-83 and 89-channel InDel classification methods using InDel data from the ICGC cohort (*n* = 3,323).** There is no simple, direct 1-to-1 mapping between the methods. In general, the 89-channel taxonomy expands upon channels that had most of the signals, here 1 bp A/T InDels, into a larger array of channels, and condenses longer InDels and/or genome motifs infrequent in the genome (where signals were scant or nonexistent) into fewer InDel subcategories. NM, not mapped.

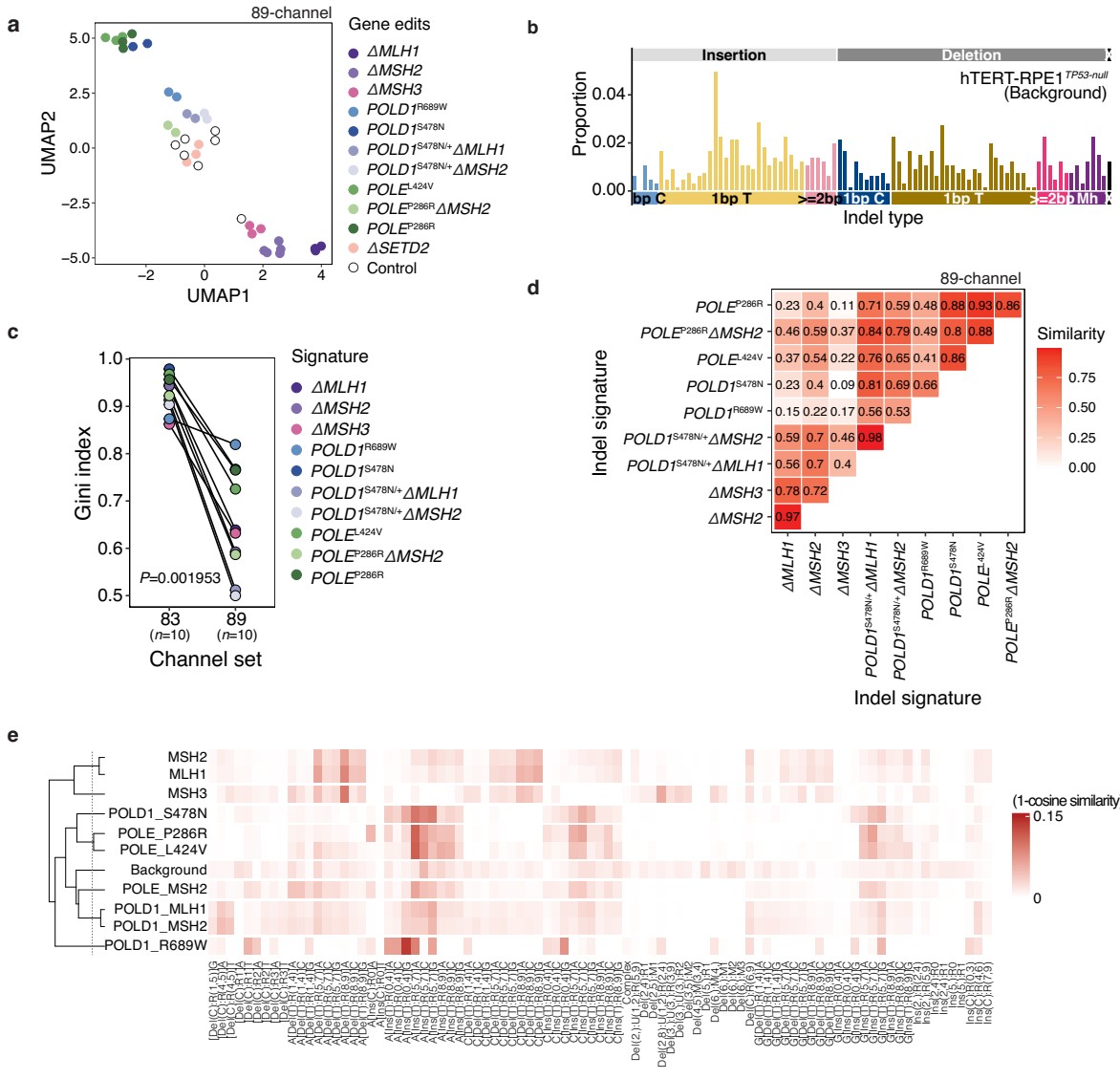

**Extended Data Fig. 5 | The 89-channel InDel signatures of post-replicative repair (PRR)-deficient isogenic human cell line models. a**, UMAP clustering of de novo 89-channel InDel profiles of gene-edit subclones (*n* = 37). UMAP, uniform manifold approximation and projection for dimension reduction. **b**, Aggregated 89-channel background InDel profile of hTERT-RPE1$^{TP53\text{-null}}$ control subclones (*n* = 7). **c**, Gini index of experimentally generated InDel signatures (*n* = 10) in

COSMIC-83 versus 89-channel format. Two-tailed Wilcoxon signed-rank test, *P* = 0.001953. **d**, Cosine similarities among experimentally generated 89-channel PRRd InDel signatures (Fig. 2d). **e**, Unsupervised clustering of experimentally generated 89-channel PRRd InDel signatures using cosine distance. Dashed vertical line marks 0.1 cut-off.

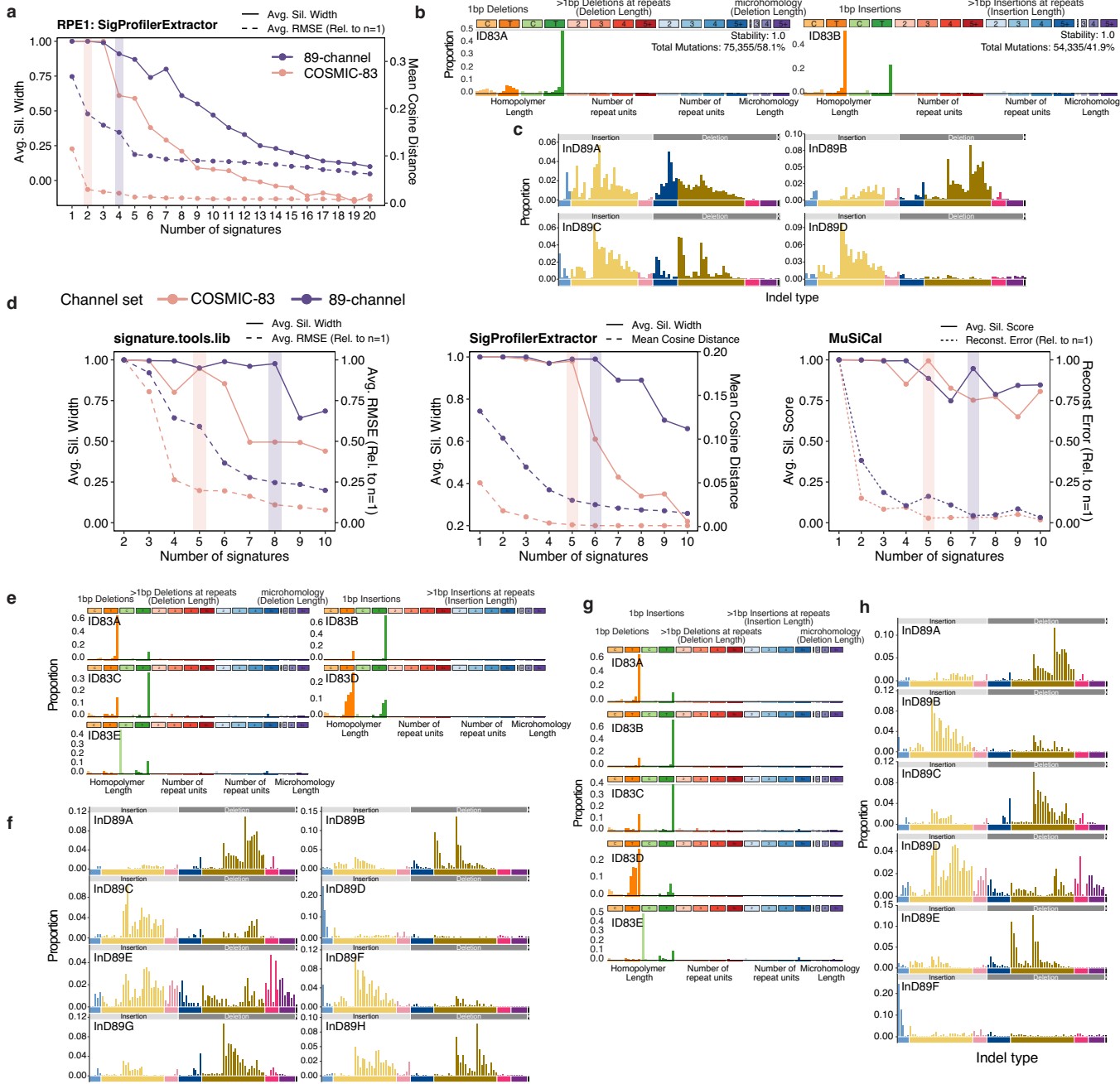

**Extended Data Fig. 6 | De novo signature extraction of COSMIC-83 versus 89-channel InDel catalogs demonstrates differential signature recovery.**
**a**, Signature selection plot of de novo extraction by SigProfilerExtractor[39] of *n* = 37 experimental samples in COSMIC-83 or 89-channel InDel catalogs. Suggested solutions are shaded in orange or purple for COSMIC-83 and 89-channel catalogs, respectively. **b**, Suggested de novo solution (two signatures) by SigProfilerExtractor for COSMIC-83 experimental cohort (*n* = 37). **c**, Suggested de novo solution (four signatures) by SigProfilerExtractor for 89-channel experimental cohort (*n* = 37). **d**, Signature selection plots of de novo extraction by signature.tools.lib[3], SigProfilerExractor[39] and MuSiCal[8] of *n* = 52

ICGC colorectal cancers in COSMIC-83 versus 89-channel InDel catalogs. Suggested solutions are shaded in orange or purple for COSMIC-83 and 89-channel InDel catalogs, respectively. **e**, Selected de novo solution (five signatures) from signature.tools.lib for ICGC colorectal cancer COSMIC-83 catalogs. **f**, Selected de novo solution (eight signatures) from signature.tools.lib for ICGC colorectal cancer 89-channel catalogs. **g**, Suggested de novo solution (five signatures) by SigProfilerExtractor for ICGC colorectal cancer COSMIC-83 catalogs. **h**, Suggested de novo solution (six signatures) by SigProfilerExtractor for ICGC colorectal cancer 89-channel catalogs. Signatures from MuSiCal extraction are not shown.

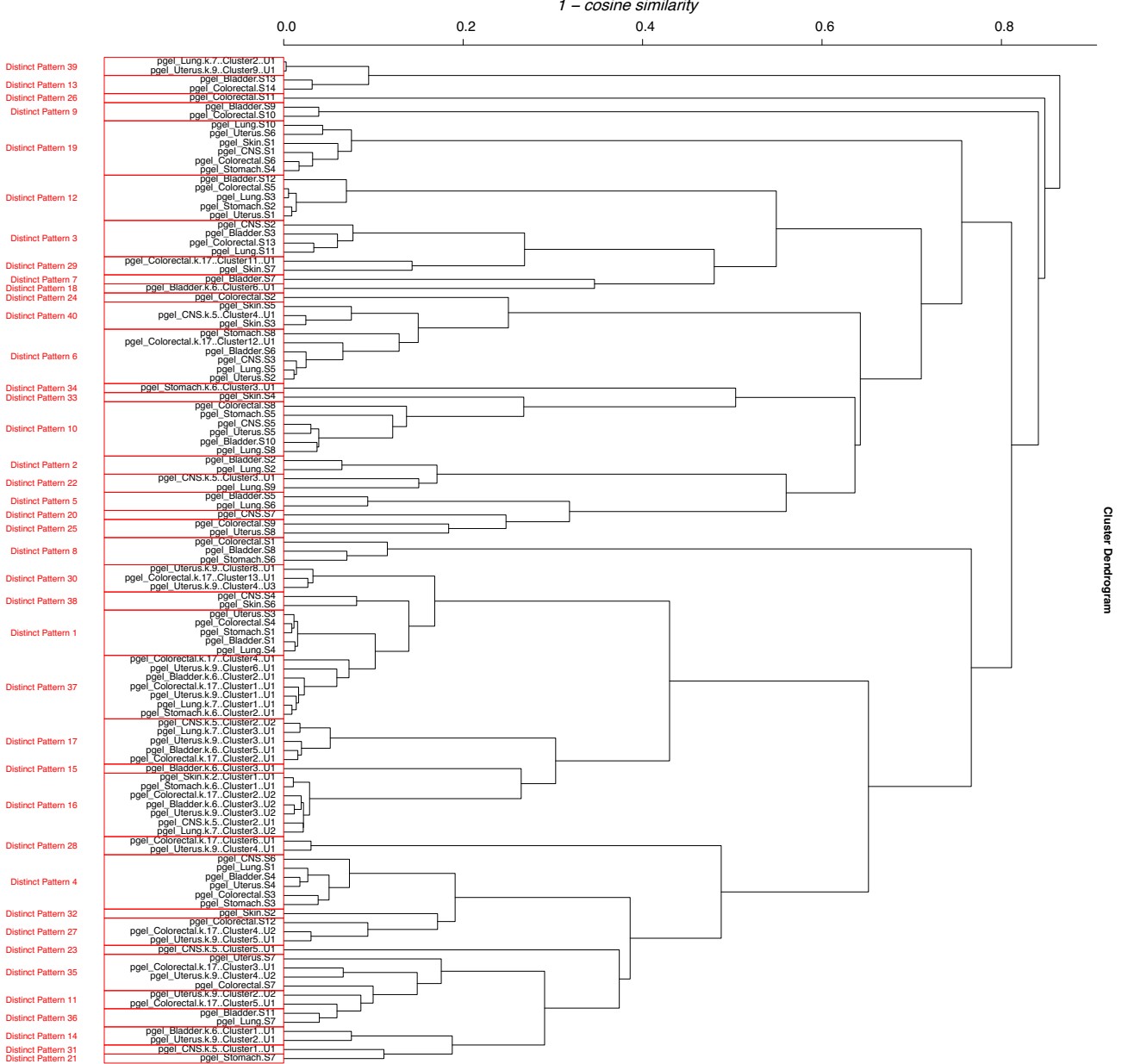

**Extended Data Fig. 7 | Hierarchical clustering and dendrogram partitioning of *n* = 111 tissue-specific signatures into *n* = 40 distinct patterns.** Hierarchical clustering and dendrogram partitioning of *n* = 111 tissue-specific signatures into *n* = 40 distinct patterns using cosine distance (1 − cosine similarity) as distance metric, with a cut-off of 0.15. The 40 distinct patterns were manually reviewed, revised and inspected for mixed patterns to produce the final 37 consensus InDS (Fig. 3 and Supplementary Tables 9–11; Methods).

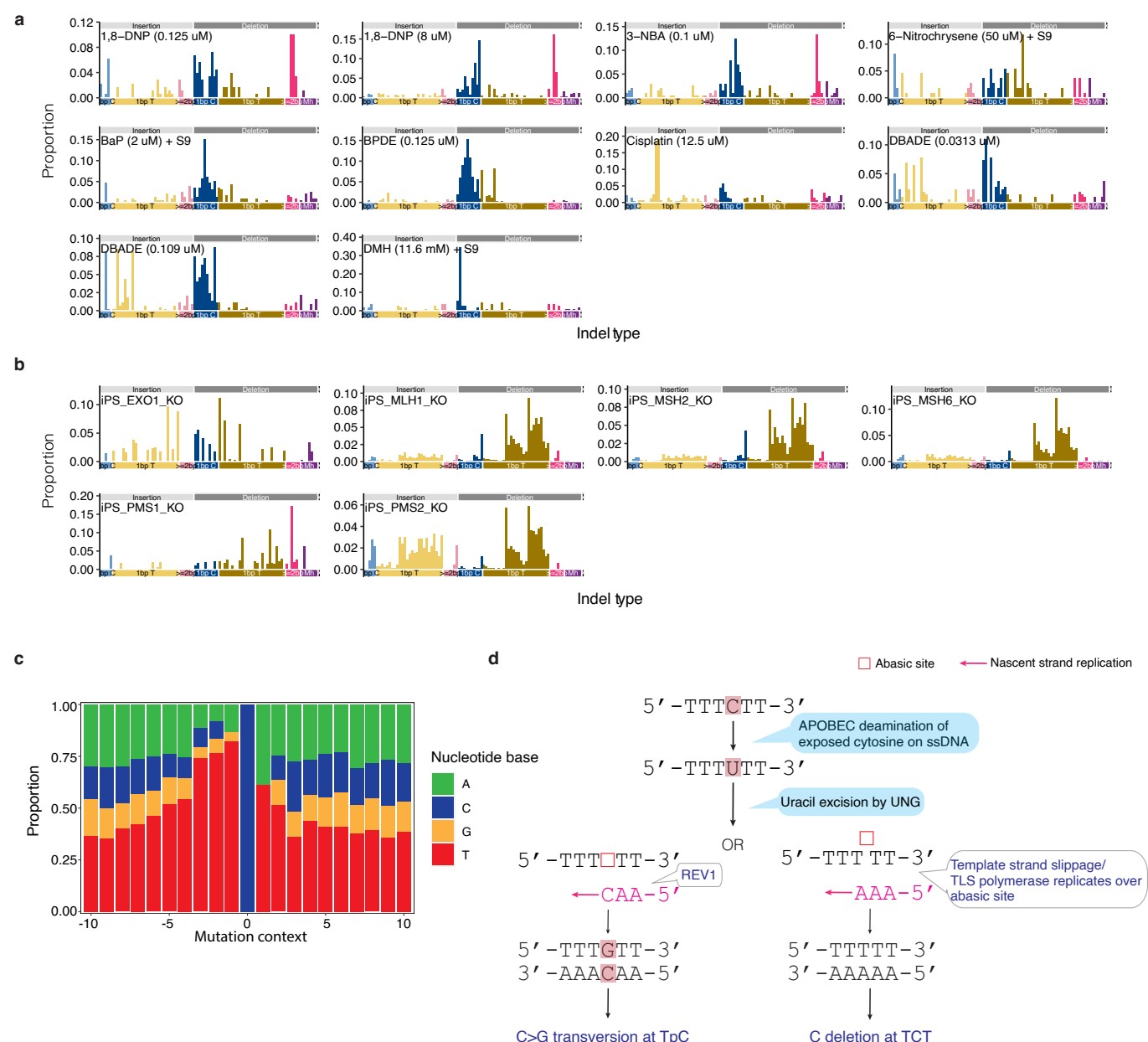

**Extended Data Fig. 8 | InDel signatures of mutagen exposures and DNA repair/replication gene knockouts in human induced pluripotent stem cells, as well as proposed etiology for InD9a. a**, InDel signatures of exposures to environmental mutagens in human induced pluripotent stem cells (hiPSC). Data reanalyzed from previous study[52]. **b**, InDel signatures of DNA repair/replication gene knockouts in human induced pluripotent stem cells. Data reanalyzed from previous study[46]. **c**, Extended sequence context of 1 bp C deletions of InD9a showed TTCT/TTCA enrichment for APOBEC deamination. APOBEC, apolipoprotein B mRNA-editing enzyme, catalytic polypeptide. **d**, Proposed etiology for APOBEC-associated InD9a. APOBEC, apolipoprotein B mRNA-editing enzyme, catalytic polypeptide; TLS, translesion synthesis; UNG, Uracil-N-glycosylase.

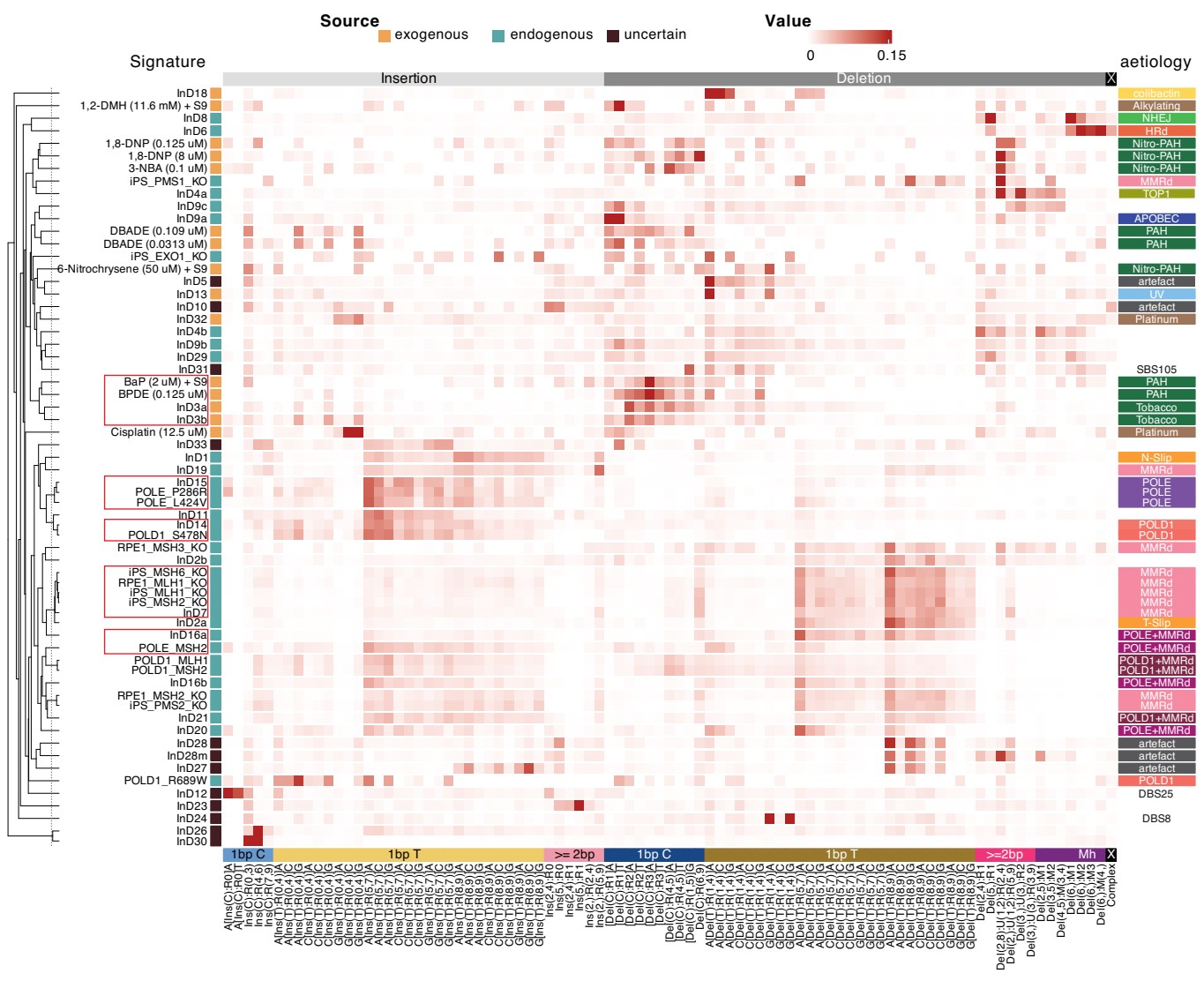

**Extended Data Fig. 9 | Heatmap clustering of *n* = 26 experimental InDel signatures and *n* = 37 cancer-derived InDS.** Heatmap clustering of *n* = 26 experimentally generated signatures and 37 cancer-derived InDS. Experimentally generated signatures include *n* = 10 from the current study, *n* = 10 from the mutagen study[52] (Extended Data Fig. 8a) and *n* = 6 from iPSC knockout study[46]

(Extended Data Fig. 8b). Putative sources (left) and etiologies (right) of the signatures are annotated if known. HRd, homologous recombination deficiency; PAH, polycyclic aromatic hydrocarbons; N-Slip, nascent strand slippage; T-Slip, template strand slippage; MMRd, mismatch repair deficiency.

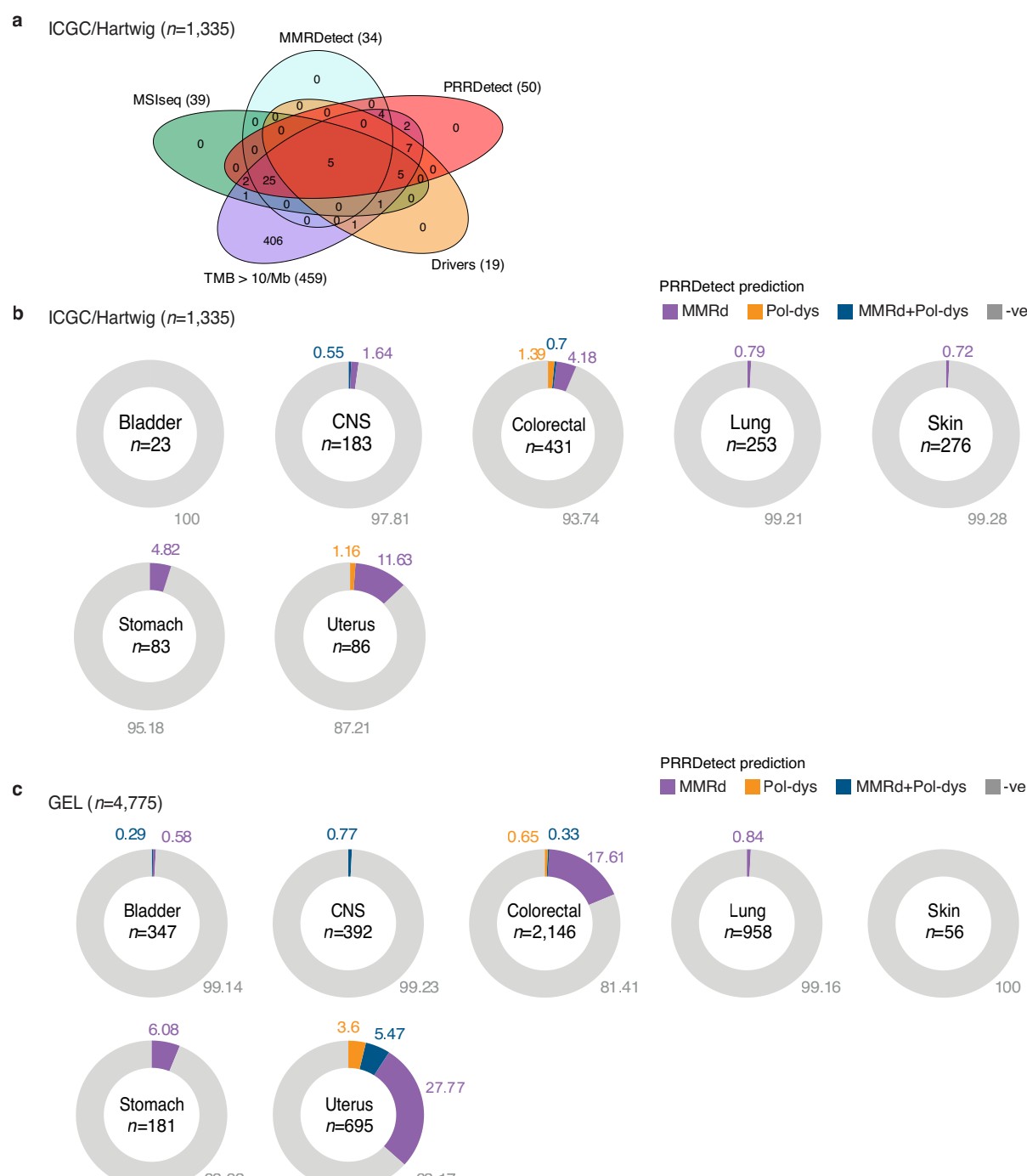

**Extended Data Fig. 10 | Prediction results of PRRDetect on *n* = 1,335 cancers from ICGC/Hartwig selected cohorts and *n* = 4,775 GEL cancers, stratified by tissue types. a**, Venn diagrams showing the concordance and discordance between different predictors in selected ICGC and Hartwig cohort (*n* = 1,335).

**b**, PRRDetect prediction of ICGC and Hartwig seven cancer types (*n* = 1,335; Supplementary Table 17). **c**, PRRDetect prediction of GEL seven cancer types (*n* = 4,775; Supplementary Table 12).

# Reporting Summary

## Statistics

For all statistical analyses, confirm that the following items are present in the figure legend, table legend, main text, or Methods section.

| n/a | Confirmed | |
|---|---|---|
| ☐ | ☒ | The exact sample size (*n*) for each experimental group/condition, given as a discrete number and unit of measurement |
| ☐ | ☒ | A statement on whether measurements were taken from distinct samples or whether the same sample was measured repeatedly |
| ☐ | ☒ | The statistical test(s) used AND whether they are one- or two-sided *Only common tests should be described solely by name; describe more complex techniques in the Methods section.* |
| ☒ | ☐ | A description of all covariates tested |
| ☒ | ☐ | A description of any assumptions or corrections, such as tests of normality and adjustment for multiple comparisons |
| ☐ | ☒ | A full description of the statistical parameters including central tendency (e.g. means) or other basic estimates (e.g. regression coefficient) AND variation (e.g. standard deviation) or associated estimates of uncertainty (e.g. confidence intervals) |
| ☐ | ☒ | For null hypothesis testing, the test statistic (e.g. *F*, *t*, *r*) with confidence intervals, effect sizes, degrees of freedom and *P* value noted *Give P values as exact values whenever suitable.* |
| ☒ | ☐ | For Bayesian analysis, information on the choice of priors and Markov chain Monte Carlo settings |
| ☒ | ☐ | For hierarchical and complex designs, identification of the appropriate level for tests and full reporting of outcomes |
| ☒ | ☐ | Estimates of effect sizes (e.g. Cohen's *d*, Pearson's *r*), indicating how they were calculated |

*Our web collection on statistics for biologists contains articles on many of the points above.*

## Software and code

Policy information about availability of computer code

| Data collection | We performed whole-genome sequencing of all experimental RPE1 samples generated in this study on Illumina Novaseq 6000 platform, generating 150 base pair paired-end reads, aiming for an average genome-wide sequence coverage of 25x. |
|---|---|
| Data analysis | De novo signature extraction and decomposition of mutational signatures was performed using SigProfilerExtractor (v.1.1.18), along with SigProfilerMatrixGenerator (v.1.2.4) available at https://github.com/AlexandrovLab/SigProfilerExtractor. Signatures were also extracted using signature.tools.lib (https://github.com/Nik-Zainal-Group/signature.tools.lib, v.2.4.4) and MuSiCal (https://github.com/parklab/MuSiCal, v.1.0.0) using default parameters. Experimental WGS short read data were were aligned to the human reference genome GRCh38 assembly using "bwa mem 0.7.17-r1188". Quality control and bioinformatic analysis of the WGS data was performed using "CaVEMan v1.13.15" for substitutions, "Pindel v3.2.0" for insertions/deletions. Structural rearrangement counts were low, copy number data were not informative, and hence not analysed. Experimental signature derivation was performed as described in doi: 10.1038/s43018-021-00200-0 and codes can be obtained from https://github.com/xqzou/COMSIG_KO and https://github.com/Nik-Zainal-Group/signature.tools.lib. The source code for clinical classifier, PRRDetect, can be obtained from https://github.com/Nik-Zainal-Group/PRRDetect. Indel segmentation and signature classification script can be accessed via https://github.com/Nik-Zainal-Group/indelsig.tools.lib.<br><br>For replicative strand analysis, intersectBed in BEDtools (v.2.26.0-114-g4c407ce) was utilized to identify mutations overlapping specific genomic features. |

For manuscripts utilizing custom algorithms or software that are central to the research but not yet described in published literature, software must be made available to editors and reviewers. We strongly encourage code deposition in a community repository (e.g. GitHub). See the Nature Portfolio guidelines for submitting code & software for further information.

## Data

Policy information about availability of data

All manuscripts must include a data availability statement. This statement should provide the following information, where applicable:
- Accession codes, unique identifiers, or web links for publicly available datasets
- A description of any restrictions on data availability
- For clinical datasets or third party data, please ensure that the statement adheres to our policy

Genomics England (GEL) cohort data (version 8) can be accessed via https://www.genomicsengland.co.uk/. Indels were called with Strelka v.2.4.7 using somatic calling mode. ICGC and TCGA WGS data were as published in https://doi.org/10.1038/s41586-020-1969-6 and can be obtained from https://dcc.icgc.org/releases/ PCAWG. Hartwig metastasis WGS data can be obtained from Hartwig Medical Foundation through standardized procedures and request forms that can be found at https://www.hartwigmedicalfoundation.nl/en/appylying-for-data/.

Raw sequence files from the hTERT-RPE1 mutation accumulation experiment are deposited at the European Genome-Phenome Archive with accession numbers EGAD50000000209. Mutation call, variant data have been deposited at Mendeley: doi: 10.17632/3k2tpx9ssr.2. The curated data are available for general browsing from our reference mutational signatures website, Signal (https://signal.mutationalsignatures.com). Mutagen exposure data in human induced pluripotent stem cells were published and can be accessed via https://data.mendeley.com/datasets/m7r4msjb4c/2. Human iPS knockout data were obtained directly from doi: 10.1038/s43018-021-00200-0.

Primary data from the 100,000 Genomes Project, which are held in a secure research environment, are available to registered users. See https://www.genomicsengland.co.uk/research for further information or contact research-network@genomicsengland.co.uk. The results of RPE1 experimental signatures can be browsed at https://signal.mutationalsignatures.com/explore/main/experimental/experiments?study=7. InD signatures of the seven cancer types are accessible at https://signal.mutationalsignatures.com/explore/main/cancer/signatures?mutationType=3&study=7. All downstream analyses data and results are provided in the Supplementary Tables accompanying the study.

## Research involving human participants, their data, or biological material

Policy information about studies with human participants or human data. See also policy information about sex, gender (identity/presentation), and sexual orientation and race, ethnicity and racism.

| | |
|---|---|
| Reporting on sex and gender | n/a |
| Reporting on race, ethnicity, or other socially relevant groupings | n/a |
| Population characteristics | n/a |
| Recruitment | n/a |
| Ethics oversight | n/a |

Note that full information on the approval of the study protocol must also be provided in the manuscript.

# Field-specific reporting

Please select the one below that is the best fit for your research. If you are not sure, read the appropriate sections before making your selection.

☒ Life sciences      ☐ Behavioural & social sciences      ☐ Ecological, evolutionary & environmental sciences

For a reference copy of the document with all sections, see nature.com/documents/nr-reporting-summary-flat.pdf

# Life sciences study design

All studies must disclose on these points even when the disclosure is negative.

| | |
|---|---|
| Sample size | From a statistical standpoint, this was an exploratory study, and there were no pre-defined hypothesis tests for which sample-size power calculations would have been appropriate. The sample size was determined by the numbers of genes of interest, or by the number of tumor genomes in relevant cancer types represented by publicly available somatic mutation data, e.g., Sample size of GEL cohort was chosen based on the availability of whole genome sequencing of tumour/normal pairs in the Genomics England research environment. |
| Data exclusions | rom a statistical perspective, this was an exploratory study, and there were no pre-defined hypothesis tests for which pre-defined data exclusion criteria would have been appropriate. Therefore, no data were excluded from by our algorithms. Having said that, we did exclude four control samples for quantitative analyses. This was specified in Supplemental Table 1. The reasons for exclusion were mainly due to the culture doubling time differences. They were included in qualitative mutational signature spectra analysis for more stability, but not considered for mutation count/burden analyses. |

| Replication | Each experimental gene edit has at least 2-5 sub-clones as biological replicates per genotype. They were also all validated with in vivo cancer analyses. |
| Randomization | The question of allocation to experimental groups is not applicable to this study. No randomization was performed. All experimental samples were contrasted against an isogenic unedited/WT control. |
| Blinding | We applied the analysis algorithms to each and every gene edit in the dataset in exactly the same way and without any prior expectations about the desired outcome of the analysis. Therefore, blinding was not required. |

# Reporting for specific materials, systems and methods

We require information from authors about some types of materials, experimental systems and methods used in many studies. Here, indicate whether each material, system or method listed is relevant to your study. If you are not sure if a list item applies to your research, read the appropriate section before selecting a response.

## Materials & experimental systems

| n/a | Involved in the study |
|---|---|
| ☒ | Antibodies |
| ☐ | Eukaryotic cell lines |
| ☒ | Palaeontology and archaeology |
| ☒ | Animals and other organisms |
| ☒ | Clinical data |
| ☒ | Dual use research of concern |
| ☒ | Plants |

## Methods

| n/a | Involved in the study |
|---|---|
| ☒ | ChIP-seq |
| ☒ | Flow cytometry |
| ☒ | MRI-based neuroimaging |

## Eukaryotic cell lines

Policy information about cell lines and Sex and Gender in Research

| Cell line source(s) | The original hTERT RPE-1 are hTERT-immortalized retinal pigment epithelial cells derived by transfecting the RPE-340 cell line with the pGRN145 hTERT-expressing plasmid, which is commercially available from ATCC (https://www.atcc.org/products/crl-4000). This is a near-diploid human cell line of female origin with a modal chromosome number of 46 that occurred in 90% of the cells counted. The specific clone used in this study was originally generated from doi: 10.1038/s41586-018-0291-z. They were gifts from M. Tarsounas, (Department of Oncology, University of Oxford, UK). |
| Authentication | The cell lines were not authenticated in this study. However, we did have the whole genome-sequencing data and had matched SNP genotype profiles to confirm the cell line identities and their isogenicity. |
| Mycoplasma contamination | Stock cell line was tested negative for mycoplasma contamination when banked and used for the first time, but not tested again throughout the mutation accumulation experiment and subsequent single-cell subcloning steps. |
| Commonly misidentified lines (See ICLAC register) | Not applicable as none were used. |

## Plants

| Seed stocks | Not applicable as none were used. |
| Novel plant genotypes | Not applicable as none were used. |
| Authentication | Not applicable as none were used. |

