## [Peer Review File · Nature Genetics]

A redefined indel taxonomy provides insights into mutational signatures

Corresponding Author: Professor Serena Nik-Zainal

This manuscript has been previously reviewed at another journal. This document only contains information relating to versions considered at Nature Genetics.

A version of this paper was originally rejected for publication by Nature Genetics, however that decision was reconsidered after appeal by the authors.

Version 0:

Decision Letter:

7th May 2024

Dear Professor Nik-Zainal,

First, please accept my apologies for the delay in returning this decision to you and also for the confusing emails you received from me! Thank you for bearing with me.

Your Article entitled "Redefined indel taxonomy reveals insights into mutational signatures" has now been seen by Reviewer #1 who was asked to provide feedback on your response to their comments and also those from Reviewer #2 who was unable to re-review. Reviewer #1's feedback is below. While they find your work of potential interest, they have raised ongoing concerns which in our view are sufficiently important that they preclude publication of the work in Nature Genetics, at least in its present form.

While the referees find your work of some interest, they raise concerns about the strength of the novel conclusions that can be drawn at this stage.

Should further experimental data allow you to fully address these criticisms we would be willing to consider an appeal of our decision (unless, of course, something similar has by then been accepted at Nature Genetics or appeared elsewhere). This includes submission or publication of a portion of this work somewhere else.

The required new experiments and data include, but are not limited to those detailed here. We hope you understand that until we have read the revised manuscript in its entirety we cannot promise that it will be sent back for peer review.

If you are interested in attempting to revise this manuscript for submission to Nature Genetics in the future, please contact me to discuss a potential appeal. Otherwise, we hope that you find our referees' comments helpful when preparing your manuscript for resubmission elsewhere.

Although we cannot publish your paper, it may be appropriate for another journal in the Nature Portfolio. If you wish to explore the journals and transfer your manuscript please use our manuscript transfer portal. You will not have to re-supply manuscript metadata and files, unless you wish to make modifications. For more information, please see our [manuscript transfer FAQ](http://www.nature.com/authors/author_resources/transfer_manuscripts.html?WT.mc_id=EMI_NPG_1511_AUTHORTRANSF&WT.ec_id=AUTHOR) page.

Sincerely,

Safia Danovi, PhD
Senior Editor, Nature Genetics
ORCID: 0009-0007-7822-5479

Reviewers' Comments:

Reviewer #1:

Remarks to the Author:

The authors have provided a revised version of their study "Redefined indel taxonomy reveals insights into mutational signatures".

As before, the study begins with mutational patterns in cellular models of post replicative repair deficiency (PRRd) and ends with a revised 89 channel indel taxonomy from which they derive signatures and apply to pan-cancer data, including training a new PRRdetect classifier.

Overall, while some clarifications have been made in the text and rebuttal, there is still a surprising disconnect between two parts of the paper - looking at cell models and cancer genome data analysis. Especially now that the authors have clarified that the revised indel taxonomy is based on cancer genomes, as is the PRRdetect training.

The flow of the paper seems to then use the cell models mainly to show that COSMIC-83 is inadequate, after which these unique data are sort of cast aside as the study moves to a second technical cancer genomics analysis that might stand alone as a separate paper. Importantly, the final classifier PRRDetect seems to offer incremental classification benefit over their previous MRRdetect algorithm and SBS signatures, especially given the rarity of polymerase dysfunction and the ease of classifying it with SBSs. It would seem like their unique cell model dataset and new indel taxonomy system would be able to more precisely diagnose the repair dysfunction (for example to a specific protein defect). Such a demonstration would elevate it much more clearly above the state of the art (including using the burden of SBS6 and 10 to diagnose polymerase deficiency).

Specific critiques

- The indel taxonomy still seems arbitrary, and not really justified by biological principles or data (including the RPE1 data), especially to a general genetics reader. While it is clear from their paper that COSMIC-83 is lacking, it is not convincingly shown that the proposed taxonomy is the natural successor.

* One way to present the new taxonomy is as a modification / improvement to COSMIC-83. There seem to be two key differences with COSMIC: First, longer repeat lengths (up to 9) are captured. Second, the base context to the 5' and 3' of a mononucleotide repeat is considered. Is this true? Can these two choices be concisely justified by main text data analysis or cited biological first principles?

* Along these lines there is data in the supplement, including histograms of repeat lengths which actually show a substantial fraction of indels with repeat lengths of 9 or greater (for example 1bp T indel). Does it really make sense to collapse them? In other words, why draw the line at 9+? If the authors are looking to rethink the indel taxonomy, are they being bold enough?

* Are there any other important changes vs. COSMIC-83? If so, what motivates and justifies them? The remaining differences seem minor: These include collapsing of categories - for example the authors group all length 2 to 5 microhomology deletions into a single category, while COSMIC considers them separately. Complex indels are given their own channel, and ignored in COSMIC-83. There is also a new approach to indel segmentation, which perhaps may help reproducibility across indels called using different conventions. It is not clear whether COSMIC-83 already does this kind of segmentation.

* The average reader should be able to very readily grasp important features of the new taxonomy and key differences with COSMIC 83. The schematic in Figure 2a seems important for this and also to explain the channel bar plots. However, as it stands, it seems overly complicated and main differences with COSMIC-83 are not apparent. Perhaps it could be simplified, and the full version can be moved to extended data.

* How are insertions with microhomology classified? Namely, insertions aligning with a spacer to flanking sequence. It does not seem like they have a category here.

* Since the taxonomy is apparently independent of the RPE1 experiments, the authors may actually want to consider presenting the cancer genome analyses first and ending with the RPE1 data. In addition to more solid motivation for the indel taxonomy, this would require restructuring of the paper, but maybe provide an opportunity to close with some biological insights.

- There still appear to be some circularities in the classifier training and testing, including a concern about data leakage.

* The new 89-channel system is derived from the entirety of pan-cancer data (18522 samples) which seems also to include samples used to test PRRdetect, including ICGC, TCGA, and HMF. Since these test cohorts (ICGC, TCGA, HMF) were used for feature engineering (Extended Fig 4, supplementary note) they are not independent. The feature engineering and training

of signatures and classifiers should be done on a single cohort, for example GEL, leaving the other datasets for testing, including ROC / AUC analysis and calculations of sensitivity and specificity.

* Related to above, it is not clear whether new ICGC breast analysis was done on samples included in the ICGC analyses shown in Extended Fig. 4.

* It is still unclear how gold-standard labels are derived - namely what are the criteria for labeling a sample as PRRd or PRR proficient. Methods currently refer to "manual curation" for GEL samples and for the remaining samples, "true labels were established based on IHC of four MMR proteins and driver mutations". The authors should offer much more detail about these criteria and how frequently each was applied (in how many samples IHC vs. driver mutations vs. combinations was used to determine the label, including whether / how allelic status of driver mutations was assessed).

* Specifically, how were true labels derived for samples / cohorts (like ICGC / HMF) where presumably IHC labels are not available. (for example Figure 4e). I presume it's using driver status. But then the authors refer to the false negative rate of using strictly driver events (66%, line 365).

* Reading through methods, signature analysis seems to have been used to pick PRRd and non PRRd samples for GEL ("samples .. were confirmed to lack driver mutations .. and displayed no evidence of signatures associated abnormalities" (lines 881-883). Were signatures (or the lack of) part of the criteria used to determine PRR proficient samples? If so, was this done only for training or for testing? If so, this could obviously bias either training and testing or both.

* From which cohorts are the numbers in lines 350-358 derived (sensitivity, AUC). Cited figure 4 panels (c,e-f) show ICGC / Hartwig results but unclear how the true labels were defined and hence how sensitivity / AUC were computed for these cohorts. Also 4c labels these cohorts as "application" but it appears that they are being used for testing. If so, then why not include them in the ROC analysis (Figure 4d) ?

* It is unclear which groupings refer to algorithm outputs rather than true labels (meaning those derived from manual curation, IHC, and / or driver analysis). This is additionally complicated by the fact that the authors have published a previous algorithm MMRdetect with overlapping outputs to PRRdetect. For example in 4f and 4g the caption implies that some of the labels (MMRd, Poly-dis) are algorithm outputs, however in the previous panel MMRd and Poly-dis refer to the true labels. Also in Extended figure 10, among other places in the manuscript. It would help to use uniform and unambiguous terminology to make this distinction clear across the manuscript.

- The ICGC breast cohort used for validation (Figure 4d) does not appear to have many positive (PRRd) samples, just judging by the shape of the curves. How many are the results in Figure 4d statistically significant? In other words, is the AUC generated by PRRdetect significantly higher than the previously published methods including MMRdetect?

- The previously published MMRdetect algorithm seems to very comparable outputs to PRRdetect (Figure 4e). This is not a coincidence, since PRRdetect meant to capture MMR dysfunction as well as polymerase dysfunction, as well as a mixture of the two. However polymerase dysfunction is very rare, as extended figure 10 shows, so the number of cases this will affect is small. Hence it begs the question as to what is the incremental advance of PRRdetect, especially as a clinical tool. Does it substantially improve both MMRd and polymerase deficiency detection, or just one of the two?

- Specifically, how much better is PRRdetect at detecting polymerase deficiency than just the total SBS6 + SBS10 mutational burden in clinical tumor samples. While MMR is indeed hard to specifically detect with SBSs only, polymerase dysfunction detection seems more straightforward.

- Similar for cell models - the study focuses on indels, but how much additional accuracy do indels provide on top of SBS6+10 for polymerase dysfunction detection in these models? Extended Figure 2g shows cosine similarity but does not address this specifically.

- How does the new taxonomy, signature system, and classifier perform on mutants profiled in the 2021 Nature Cancer paper? Given the overlap of the previous WGS data on knockouts (including MMR knockouts), it would seem fitting to analyze this.

- Taking a step back, it would seem more interesting to build a classifier to diagnose repair dysfunctions more specifically than what the authors are doing here, rather than just 3 dysfunction categories, two of which seem quite rare (Pol-dys, MMRd/Pol-dys). Why can't the RPE1 data be used (in combination with previously published 2021) and cancer genomics data to build such a classifier? This would tie together the two currently disparate parts of the paper.

- The strand bias signals are interesting, though it seems that more could be made of this. A strand bias not only reflects a difference in mutation rates but also a pyrimidine vs. purine preference for mutations arising on that strand. In other words, while the convention is to align mutational strands to pyrimidines, a leading strand bias implies either an increased pyrimidine mutation rate on the leading strand or an increased purine mutation rate on the lagging strand. Under the biological assumption that dysfunction of POLE and POLD increases mutation rates on the leading and lagging strand, respectively, their data implies that both POLE and POLD dysfunction preferentially causes T rather than A insertions. How do the authors explain this?

- Since the study does not include immunotherapy analyses, it does not seem appropriate to mention immunotherapy in the

abstract. To the reviewer, it does seem likely that the modest improvements in PRRd accuracy provided by the classifier here will improve prediction of immunotherapy outcomes.

- Authors cite space considerations, but it seems that some of the main panels (such as 1f, 1d, 3b, 4a, 4e) could be moved to extended data or substantially condensed to show key signals. Also NG allows for up to 7 figures.

Version 1:

Decision Letter:

IMPORTANT: Please note the reference number: NG-A64708R-Z Nik-Zainal. This number must be quoted whenever you communicate with us regarding this paper.

25th Jul 2024

Dear Dr Nik-Zainal,

First, please accept my apologies for the delay in returning this decision to you. Thank you for your patience.

Thank you for asking us to reconsider our decision on your manuscript "Redefined indel taxonomy reveals insights into mutational signatures". I have now discussed your appeal with my colleagues, and we think that you have some valid points. We therefore invite you to revise your manuscript along the lines that you propose.

When preparing a revision, please ensure that it fully complies with our editorial requirements for format and style; details can be found in the Guide to Authors on our website (<http://www.nature.com/ng/>).

Please be sure that your manuscript is accompanied by a separate letter detailing the changes you have made and your response to the points raised. At this stage we will need you to upload:

1) a copy of the manuscript in MS Word .docx format.

2) The Editorial Policy Checklist:

<https://www.nature.com/documents/nr-editorial-policy-checklist.pdf>

3) The Reporting Summary:

(Here you can read about the role of the Reporting Summary in reproducible science:

<https://www.nature.com/news/announcement-towards-greater-reproducibility-for-life-sciences-research-in-nature-1.22062>)

Please use the link below to be taken directly to the site and view and revise your manuscript:

Link Redacted

With kind wishes,

Safia Danovi, PhD
Senior Editor, Nature Genetics
ORCID: 0009-0007-7822-5479

Version 2:

Decision Letter:

20th Sep 2024

Dear Serena,

First of all, please accept my apologies for the delay in returning this decision to you. Thank you for bearing with me.

Your Article entitled "Redefined indel taxonomy reveals insights into mutational signatures" has now been seen by Reviewer #1, whose comments are below. In the light of their advice we have decided that we cannot offer to publish your manuscript in Nature Genetics.

We feel that these reservations are sufficiently important as to preclude publication of this study in Nature Genetics.

I appreciate that this is not the news you were hoping for (it is not the news I wanted to give either), but given this report, I hope that you can understand our editorial position.

In the interests of time, I didn't consult with any other journals in the Nature family but would be pleased to do so if it would be helpful.

I am sorry that we cannot be more positive on this occasion but hope that you will find our referees' comments helpful when preparing your paper for submission elsewhere.

Sincerely,

Safia

Safia Danovi, PhD
Senior Editor, Nature Genetics
ORCID: 0009-0007-7822-5479

Reviewers' Comments:

Reviewer #1 (Remarks to the Author):

The authors have clarified some points, but deep methodological concerns still stand which have not been given sufficiently serious attention. These include the rigour with which the authors have trained and tested their classifier, the rationale for their feature engineering, and the marginal improvement of their classifier over the state of the art, particularly for predicting polymerase dysfunction.

- The rebuttal (response to point 7) and revised manuscript still do not seem to give the concept of data leakage the serious attention that it deserves.

The authors must be aware that putting an analysis in a different "section" of the manuscript does not make it independent. Notably, PRRdetect is built on set of mutational channels that have been hand-engineered through the analysis of RPE1 and 18K+ pan-cancer data, including the collapsing of channels as the manuscript describes. Hence the training and testing data are not independent, they were used to engineer the features which are the ingredients to PRRdetect.

- The authors do not seem to approach their data labeling with sufficient rigour, and there is a concern that this may skew both training and testing. Though the authors have removed the word "manual review" from the manuscript, there appears to be a (still) poorly described manual process to assess PRRd labels. Closer scrutiny of the supplementary data suggests that this may be impacting their results.

- The labeling primarily appears to rely on assessing genotypes at MMR and POLE1 / POLD, though for MMR they additionally use IHC when that data is available. However in many of their training and testing data, IHC data is not available. The high number of cases that are genotype "driver" negative but MMR positive suggests that there may be many more positives among cases lacking IHC, or for POLE1 / POLD where no orthogonal data is available.

- While IHC is manual, it should be possible to apply some objective and reproducible criteria by which samples are genotyped as Driver = Yes. This includes specifying which variants are included in that criteria (germline, somatic?) and describing if and when LOH or the presence of a second hit was used. Not specifying this process in sufficient detail calls into question both the training and testing of the classifier.

- In their supplementary data there are quite many cases (I count 31) where IHC is abnormal but there is no driver. Are these cases truly MMR? Do these samples not have mutations in any MMR genes? While one or more alleles MLH1 can be occasionally hypermethylated, the frequency here of wild type seems quite high. Is the MMR pattern in these consistent with MLH1 loss? Alternatively have orthogonal (such as PCR based clinical) assays to assess MSI been run on these tumors.

- The high rate of IHC MMR in wild type samples suggests that the bespoke and poorly described genotyping approach is not sufficiently sensitive, either not adequately classifying mutations as VUSs, adequately assessing LOH and copy number, and by not looking at other genes (like EPCAM, PMS1, MLH3) that have also been implicated in MMR. Abnormal MMR IHC can also be false positives - for example when staining is weak but not absent, or when it is not associated with microsatellite instability, as is clinically measured.

- In the rebuttal, the authors state "We know that we are not missing any true PRRd samples. These samples have confident labels. Ground truth is solid for all samples." But if indeed the rate of MMR deficiency is so high for wild type samples, then it calls into question whether the labels for samples that do not have IHC can be fully trusted. Notably, it calls into question the veracity of POLD1 and POLE deficiency labels which are solely reliant on analyzing genotypes.

- Given the very small number of polymerase deficient samples in the validation, it appears that all of the performance increase shown in Figure 4d is coming from better detection of MMR. This is somewhat surprising (even impressive) that they are seeing so much performance gain on a relatively well-trod analytic task, namely predicting MMRd or MSI. But are they actually seeing better prediction of POLD1 / POLE deficiency? It seems that they may not have enough numbers to show this.

- One omission from Figure 4d is MSIsensor - which is the state-of-the-art genomics tool for analyzing microsatellite instability in genome data. Did they test this? How well does it perform here? MSIsensor does not look at indels but examines (subclonal) gapped alignments at microsatellites, which PRRdetect (as far as I can tell) ignores. As I understand it, PRRdetect is getting most of its signal from clonal indels that happen outside of microsatellites, which is in a way orthogonal to what MSIsensor is detecting. Notably, MSI is not quite the same as MMRd, though they can both be consequences of mutations in so MMR genes. Are the additional samples that they are classifying MMR deficient but microsatellite stable? MSI status from a clinical microsatellite assay may be useful to address this question.

Related to this, are the additional samples that are driving this performance gain driven purely by the IHC-only samples, which were noted above?

- The authors justify somewhat arbitrary choices made for their indel channel feature engineering by using COSMIC-83 as a precedent, pointing to the fact that COSMIC-83 also made many arbitrary choices. COSMIC-83 was however proposed when there did not exist a previous system of channels to classify indel mutations and extract signatures, therefore you might argue that the bar was lower.

However, the authors' pointing to the arbitrariness of COSMIC-83 to justify their arbitrary and not clearly data or biology-driven choices in devising their 89 channel system highlights the importance of setting good precedents in the scientific literature, particularly when publishing in influential journals like Nature Genetics. If the study here is published having made similarly arbitrary choices, then future work, published perhaps in less influential journals, will cite it as a precedent to propose similarly arbitrary alternate classifications. I think this study, especially if it is to be published in NG, should set a better example.

- Another reason to demonstrate that the new channel system is not in fact arbitrary is that it would raise less questions that it was overfit to some specific classification task, which right now in combination with the issues cited above regarding training and testing is a major concern at least to this reviewer. Given the many degrees of freedom in designing the channel system (for example choosing cutoffs homopolymer tract lengths), it would be more convincing if the authors could show that their final classification performance was robust to the details of these choices.

- The channel collapsing in particular seems to be driven by an analysis of the full dataset, including samples that were used as "independent" tests of their classifier. How well would their classifier perform without any channel collapsing?

**Although we cannot offer to publish your manuscript, I suggest that you consider Nature Communications as a suitable venue for this work. To transfer your manuscript, please use our manuscript transfer portal. You will not have to re-supply manuscript metadata and files, unless you wish to make modifications. For more information, please see our [manuscript transfer FAQ](http://www.nature.com/authors/author_resources/transfer_manuscripts.html?WT.mc_id=EMI_NPG_1511_AUTHORTRANSF&WT.ec_id=AUTHOR) page.

Version 3:

Decision Letter:

IMPORTANT: Please note the reference number: NG-A64708R2-Z Nik-Zainal. This number must be quoted whenever you communicate with us regarding this paper.

14th Oct 2024

Dear Serena,

As discussed via email, we're willing to reconsider an appeal of the paper. A quirk of our online system means that unfortunately, you have to resubmit everything again. I'm sorry about this as I know it's a faff - sadly I can't do anything about it.

When preparing a revision, please ensure that it fully complies with our editorial requirements for format and style; details can be found in the Guide to Authors on our website (<http://www.nature.com/ng/>).

Please be sure that your manuscript is accompanied by a separate letter detailing the changes you have made and your response to the points raised by Reviewer #1 in the form of a full point-by-point letter that addresses all queries made.

At this stage we will need you to upload:

1) a copy of the manuscript in MS Word .docx format.

2) The Editorial Policy Checklist:

<https://www.nature.com/documents/nr-editorial-policy-checklist.pdf>

3) The Reporting Summary:

(Here you can read about the role of the Reporting Summary in reproducible science:

<https://www.nature.com/news/announcement-towards-greater-reproducibility-for-life-sciences-research-in-nature-1.22062>)

Please use the link below to be taken directly to the site and view and revise your manuscript:

Link Redacted

As discussed, we will not be going back to Reviewer #1 but will be recruiting a new referee. I will make sure that they know to reduce the scope of the review to Reviewer #1's comments in the last round (I'll ask them not to provide a full review).

With kind wishes,

Safia Danovi, PhD

Senior Editor, Nature Genetics

ORCID: 0009-0007-7822-5479

Version 4:

Decision Letter:

3rd Dec 2024

Dear Serena,

How are you? I hope you're well.

I'm so sorry for the delay in returning this decision to you. As you know, we decided to send your article entitled "Redefined indel taxonomy reveals insights into mutational signatures" to an arbitrating reviewer whose comments are below and in the attached. You will see that they broadly agree with most of the points raised by you in your point-by-point rebuttal but have made some requests for further analyses. We are interested in the possibility of publishing your study in Nature Genetics, but would like to consider your response in the form of a revised manuscript before we make a final decision on publication.

We therefore invite you to revise your manuscript taking into account all reviewer comments. Please highlight all changes in the manuscript text file. At this stage we will need you to upload a copy of the manuscript in MS Word .docx or similar editable format.

*2) If you have not done so already please begin to revise your manuscript so that it conforms to our Article format instructions, available

[here](http://www.nature.com/ng/authors/article_types/index.html).

*3) Include a revised version of any required Reporting Summary: <https://www.nature.com/documents/nr-reporting-summary.pdf>

Link Redacted

We hope to receive your revised manuscript within four to eight weeks. If you cannot send it within this time, please let us know.

Sincerely,

Safia Danovi, PhD
Senior Editor, Nature Genetics
ORCID: 0009-0007-7822-5479

[Ed; Reviewer 2 can expertise in cancer genomics, including SVs, mutational signatures and copy number analysis]

Reviewers' Comments:

Reviewer #2 (Remarks to the Author):

Please see attached pdf for my comments on the previous review and rebuttal (marked in green).

Reviewer #2 (Remarks on code availability):

The PRRdetect package installs correctly but I was unable to fully test the code without example data. I suggest example data and a vignette be provided.

There was no documentation provided on how to use functions in the <https://github.com/xqzou/indelsig.tools.lib> repo to reproduce the results found in the paper. This needs to be added.

Links to the browsable data on the <https://signal.mutationalsignatures.com> website are broken.

Version 5:

Decision Letter:

Our ref: NG-A64708R4

22nd Jan 2025

Dear Dr Nik-Zainal,

Thank you for submitting your revised manuscript "Redefined indel taxonomy reveals insights into mutational signatures" (NG-A64708R4). It has now been seen again by Reviewer #2 and their comments are below. The reviewer finds that the paper has improved in revision, and therefore we'll be happy in principle to publish it in Nature Genetics, pending minor revisions to satisfy the referees' final requests (regarding code) and to comply with our editorial and formatting guidelines.

Sincerely,

Safia Danovi, PhD
Senior Editor, Nature Genetics
ORCID: 0009-0007-7822-5479

Reviewer #2 (Remarks to the Author):

The authors have now addressed all of my comments with regards to the manuscript. Congratulations on an excellent piece of work.

Unfortunately however, I have been unable to run the code (see below for comments).

Reviewer #2 (Remarks on code availability):

Can the authors please provide install instructions for the <https://github.com/Nik-Zainal-Group/indelsig.tools.lib> repository? I spent quite a number of hours attempting to track down and resolve unspecified dependencies, but could not get it to work. I suggest the authors test installation on a clean system and specify all dependencies and package versions (or provide a docker image or similar).

RTR.v4

Reviewer #1:

Remarks to the Author:

The authors have provided a revised version of their study "Redefined indel taxonomy reveals insights into mutational signatures".

As before, the study begins with mutational patterns in cellular models of post replicative repair deficiency (PRRd) and ends with a revised 89 channel indel taxonomy from which they derive signatures and apply to pan-cancer data, including training a new PRRdetect classifier.

#Summary Section

Overall, while some clarifications have been made in the text and rebuttal, there is still a surprising disconnect between two parts of the paper - looking at cell models and cancer genome data analysis. Especially now that the authors have clarified that the revised indel taxonomy is based on cancer genomes, as is the PRRdetect training.

The flow of the paper seems to then use the cell models mainly to show that COSMIC-83 is inadequate, after which these unique data are sort of cast aside as the study moves to a second technical cancer genomics analysis that might stand alone as a separate paper.

Thank you for taking the time to review our paper again.

We find it a little perplexing that the experimental analyses and the cancer analyses in our manuscript are considered dichotomous. The principle of performing experiments and then assessing generalisability in a real-world cancer dataset is not uncommon. This step-wise approach of presenting data has been used by others recurrently (PMID 35140396 is a good example of one exploring indel signatures published in Nature (amongst many others)).

In our view, the experimental and cancer analyses are fully integrated; the cell model data are central to the narrative and fully integrated with the cancer data:

- Section 1 on "Diversity of indel patterns in PRRd" – here, data from cell models are the main subject matter.
- Section 2 on "Limitations of current indel taxonomy" – here, data from cell models are the main subject matter and demonstrate the problems with the current taxonomy and that a change is needed. This is an experimentally-driven exercise. It is robust and agnostic.
- Section 3 on "A new framework for classifying indels" – data from cell models are definitely not cast aside. The new indel framework is proposed based on the logical argument of using flanking sequence context (amongst other features).
 - The new framework is assessed in cancers AND
 - then in paragraphs 3-6 of the same section of the manuscript, the experimental data are used alongside cancer data to examine the technical strengths of the new classification system.

Thus, the experimental data are fully and comprehensively utilised and integrated.

Importantly, the final classifier PRRDetect seems to offer incremental classification benefit over their previous MRRdetect algorithm and SBS signatures, especially given the rarity of polymerase dysfunction and the ease of classifying it with SBSs. It would seem like their unique cell model dataset and new indel taxonomy system would be able to more precisely diagnose the repair dysfunction (for example, to a specific protein defect). Such a demonstration would elevate it much more clearly above the state of the art (including using the burden of SBS6 and 10 to diagnose polymerase deficiency).

Here, we humbly disagree with the notion that "incremental benefit" is expected because of clinical rarity of polymerase dysfunction. Rarity is not an adequate argument for dismissing a classification process.

- First, rarity in one tumor-type may not translate to rarity in other tumor-types (e.g., Mismatch repair deficiency).
- Second, 1% of a common disease like cancer, translates to many thousands of people with cancer.

RTR.v4

These are thoughtful deliberations based on the distribution of mutations (actual data is in Extended Figure 3) across multiple cohorts involving most human cancer types and are not arbitrary decisions. Our approach is data-driven and unbiased.

From a large body of published work, we know that indel formation is commonly driven by the underlying sequence motifs. While many in the community have been introduced to COSMIC-83 for signature analysis, it is not as widely adopted as the substitution channels because it has been problematic, and simply because it was the first, and has been introduced by the Sanger/COSMIC team, does not mean that it should be dogma. Many have struggled and found it challenging to interpret the COSMIC ID signature results. This is because COSMIC-ID lacks detailed sequence context information, resulting in peculiar signature assignments, leading to uncertainties in the interpretation of results (PMID: 38361034). This is not only apparent in our RPE1 dataset presented here, but also has been shown by others (PMID: 35320711, 35140396) that the COSMIC-83 is not able to resolve many biological phenotypes.

It is with these biological observations as motivations and also based on additional insights gleaned from many mechanistic studies (PMID: 30982602, 32661091, 35320711, 35140396) showing that a lot of mutational processes and mutagens show a predilection for specific sequence contexts to form adducts/damage and cause indels (e.g. UV at TT, etc), that we expanded on the current COSMIC-83 framework to include the considerations outlined above.

Therefore, we maintain that our classification system is not arbitrary. It is informed by biological observations, and guided by the understanding of the underlying data distribution. It is also noteworthy that we, in addition to capturing all previous COSMIC-83 signatures with confirmed aetiologies, also identified multiple new signatures in the 7 GEL cancer types, while providing additional insights/dimensions to guide signature interpretation.

We stand by the validity, robustness, and biological value of our approach.

#Point 1

1. One way to present the new taxonomy is as a modification / improvement to COSMIC-83. There seem to be two key differences with COSMIC: First, longer repeat lengths (up to 9) are captured. Second, the base context to the 5' and 3' of a mononucleotide repeat is considered. Is this true? Can these two choices be concisely justified by main text data analysis or cited biological first principles?

These are broadly correct but there is more complexity (which the reviewer has mentioned in #Point 3).

We present below an alluvial plot mapping the COSMIC-83 features to our InD-89 system.

RTR.v4

As can be seen above, there is no direct 1-to-1 mapping of COSMIC-83 channels to InD-89 channels. Predominantly, the InD-89 taxonomy expands upon the T indels (where most indel signals are concentrated) and condenses the longer indels (≥ 2 bp) (where the signals are scant). Detailed mapping of the channels between the two classification systems and some examples were provided in **Supplementary Table S4**.

We have inserted the following statement to fall in line with the PCAWG classification:

Line 193: Overall, compared to the COSMIC-83, the 89-channel taxonomy expands upon 1 bp T indels (where most of the signals were) into a larger array of channels, and condenses longer indels and/or genome motifs that are infrequent in the genome (where signals were scant or non-existent) into fewer indel subcategories.

There is, however, far more complexity that makes our system more reproducible and robust, which the reviewer has highlighted in **Point 3** below.

#Point 2

2. Along these lines there is data in the supplement, including histograms of repeat lengths which actually show a substantial fraction of indels with repeat lengths of 9 or greater (for example 1bp T indel). Does it really make sense to collapse them? In other words, why draw the line at 9+? If the authors are looking to rethink the indel taxonomy, are they being bold enough?

The histogram shows indels up to repeat lengths of 9, and *not 9 or greater*. See below for the zoomed-in example of all possible 1bp T deletions. The same applies to all homopolymer, polynucleotide repeats (*i.e.*, repeat # ≤ 9)

1 bp T indel channels (81 subcategories)

It is not possible using today's genomic data pipelines of any institution, to confidently go beyond repeat lengths of > 9 because of the error rates in mutation-calling algorithms. Nearly all commonly-used genomic pipelines (Broad GATK, Dragen, Strelka, Platypus, Pindel, etc) have a post-hoc filter capped at repeat lengths < 10.

Genome distribution of polynucleotide repeat tracts

The contribution of indels at very long homopolymer tract length of >9 to indel signatures could be limited given that there is a much lower prevalence across the genome (see image above). Having said that, we agree that indels at longer homopolymers may be important and should be investigated at a later date when indel data obtained through newer lower-error/long read technologies become available, in quality and quantity. Thus, we have added the following statement:

Lines 432: The possibility of adapting the taxonomy in the future to include features currently not explorable due to the limitations incurred by technological error rates of calling indels using short-read WGS (*i.e.*, at longer simple repeats), could also be revealing.

#Point 3

3. Are there any other important changes vs. COSMIC-83? If so, what motivates and justifies them? The remaining differences seem minor: These include collapsing of categories - for example the authors group all length 2 to 5 microhomology deletions into a single category, while COSMIC considers them separately. Complex indels are given their own channel, and ignored in COSMIC-83. There is also a new approach to indel segmentation, which perhaps may help reproducibility across indels called using different conventions. It is not clear whether COSMIC-83 already does this kind of segmentation.

Yes, the reviewer has highlighted a key difference and is correct in saying that our approach in classifying indels through a “segmentation process” does result in a more unified, reproducible, and standardised classification. Prior to our approach, because a binary threshold-based classification schemes was used (*i.e.*, see “repeat-mediated” vs “microhomology-mediated” deletions in #Point 1 Figure, for example), there were ambiguous cases that arose that could be classified in either category. This has never been addressed previously. To deal with this ambiguity, our “segmentation” strategy explicitly identifies the smallest prefix sequence motif that displays a maximal repetitive relationship with the 3’ sequence context and within the indel itself. This provides a standardized approach for relating indel sequence to the 3’ sequence context. COSMIC-83 does not do that.

Examples:

Indels	InD-89	COSMIC-83
C AAATA AAAATA AAATACCTAGGAAT	Del(3,):U(3):R(3,9)	5:Del:M:5
G CCC CTCAGGATGATTTCCTCCCTGG	Del(2,8):U(1,2):R(2,4)	3:Del:M:1

In the first example, AAATAAAAATA is the deletion motif, AAATA is the smallest prefix sequence motif, the unit is 5 bp, and repeated 3 times. In COSMIC-83, it is classified as long deletions (>5bp) with 5bp microhomology. However, using our strategy, it is interpreted as a loss of 2 of the 3 AAATA pentanucleotide repeats.

In the second example, CCC is the deletion motif, C is the smallest prefix sequence motif, the unit is 1 bp, and repeated 4 times. In COSMIC-83, it is classified as 3bp deletion with 1bp microhomology. However, using our strategy, it is interpreted as a loss of 3 of the 4 x C mononucleotides.

Please also see answer to #Point 1. Details of segmentation and motivations/justifications were previously described in **Supplementary Note 1**. However, we agree that we have not adequately highlighted key differences between the classification systems and have included this in the abstract starting from Line 32:

“we developed an alternative classification system that considers the 5’ and 3’ sequences flanking indels and includes informative motifs (*e.g.*, longer homopolymeric tracts) while removing uninformative channels. Our new cataloguing system ensures that every indel can be unambiguously classified into one of 89 indel subcategories and is capable of disambiguating gene-specific experimentally-generated signatures.”

We have also added text in the final paragraph of the Introduction to ensure that the distinction between our methodology and that of COSMIC-83 is clearly articulated (Lines 74-77):

“Additionally, motifs known to increase mutational vulnerability that are conflated by the current system

RTR.v4

(*i.e.*, homopolymer tracts >6nt), and genome-wide prevalence of such motifs were factored into our proposition. Our approach unambiguously classifies each indel into a specific subcategory.”

Thank you for raising this.

#Point 4

4. The average reader should be able to very readily grasp important features of the new taxonomy and key differences with COSMIC 83. The schematic in Figure 2a seems important for this and also to explain the channel bar plots. However, as it stands, it seems overly complicated and main differences with COSMIC-83 are not apparent. Perhaps it could be simplified, and the full version can be moved to extended data.

We wonder whether moving **Figure 2a** upper panel to the Supplement would make interpretation of the rest of figures challenging. Readers would have to flip back and forth to understand the full complement of the profiles, *e.g.*, in 2d. As **Figure 2a** serves as the basis for the signature plots in **Figure 2d** and **Figure 3**, we believe that having it as a main figure panel is essential.

One option is to contrast the two classification systems to highlight differences (see the alluvial plot shown in **Point 1**). Detailed mapping of the channels between the two classification systems and indel examples were already provided in **Supplementary Table S4**. It is quite an illustrative plot. The alluvial plot contrasting the two classification systems is now provided and referenced as **Extended Data Figure 4**.

#Point 5

5. How are insertions with microhomology classified? Namely, insertions aligning with a spacer to flanking sequence. It does not seem like they have a category here.

Insertions with microhomology are a rare feature (single digits if at all), even in HR-deficient samples, where they are alleged to be seen more frequently. *Critically, they cannot be unambiguously mapped to a single category.*

That’s because templated insertions are:

- often complex (*i.e.*, they are accompanied by deletion at the junction). Majority of them would have been overlooked or mis-annotated using standard indel variant callers (PMID: 26657142).
- could also be templated from inverted repeats (both left and right flanks) and not be immediately at the junction of the indel, and unaccompanied by deletions.

Although we do not designate a category specifically for these complex events, they were segmented and classified using our unified framework into various non-overlapping “insertion” and “complex” categories. Due to the complexity of these events, they are best characterised by specialised algorithms (PMID: 32234782, 26657142).

#Point 6

6. Since the taxonomy is apparently independent of the RPE1 experiments, the authors may actually want to consider presenting the cancer genome analyses first and ending with the RPE1 data. In addition to more solid motivation for the indel taxonomy, this would require restructuring of the paper, but maybe provide an opportunity to close with some biological insights.

The taxonomy *is not independent* of the RPE1 experiments. The RPE1 experiments culminated in the evidence that shows that the current COSMIC-83 is inadequate. Once the taxonomy was designed, it was also tested on the RPE1 data to show the remarkable improvement in discerning the gene-specificity that is present in the data (**Figure 2b-d, Extended Data Figure 5**).

We respectfully disagree with this suggestion. Furthermore, first, it would not make sense to start with an adaptation of the COSMIC-83 system for no reason at all, and then use the experiments to back up our argument. Second, it might be seen as trying to “appear clever” *if we started by philosophising a change*. An alternative reviewer could easily ask in this scenario, “why did they think it was necessary to adapt it in the first place?”.

RTR.v4

Thus, we are concerned that restructuring would, in fact, be detrimental to the manuscript and is not the truth of how the thought process went either.

#Point 7

- There still appear to be some circularities in the classifier training and testing, including a concern about data leakage.

7. The new 89-channel system is derived from the entirety of pan-cancer data (18522 samples) which seems also to include samples used to test PRRdetect, including ICGC, TCGA, and HMF. Since these test cohorts (ICGC, TCGA, HMF) were used for feature engineering (Extended Fig 4, supplementary note) they are not independent. The feature engineering and training of signatures and classifiers should be done on a single cohort, for example GEL, leaving the other datasets for testing, including ROC / AUC analysis and calculations of sensitivity and specificity.

Note that the former **Extended Fig 4** is now revised as the new **Extended Data Figure 3**.

The reviewer has conflated several distinct pieces of work together in the argument above. There are three distinct sections:

- 1) The demonstration of validity of the 89-channel system
- 2) Indel signature extraction from seven cancer types
- 3) The PRRDetect classifier development

1) The 89-channel system

The entirety of the pan-cancer data of 18,822 samples was *simply used to show the indel distribution across all categories from available cancer types across all three cohorts (GEL, ICGC and Hartwig)*. Because the same reviewer previously raised concern that our proposed framework was potentially “biased towards PRRD phenotype”, we presented the plots in **Extended Data Figure 3** to show that:

A/ indel distributions across InD channels are preserved regardless of cohorts or cancer types, and are not biased towards PRRd phenotypes or skewed towards hypermutator samples (**Extended Data Figure 3e**).

B/ Additionally, the new indel classification channels have data in all channels across all cancer types, in contrast to the sparse profiles with COSMIC-83 (**Extended Data Figure 3a-d**).

§ ratio, and importantly, shows that our final channel set does not bias towards hypermutator phenotypes (and hence PRRd samples), and reinforces that our approach is data-driven.

2) Indel signature extraction

In the main text (Line 252), we clearly stated that indel signature extraction *was performed on the 7 selected cancer types within the GEL cohort*. An indel signature extraction has only been performed on 4,775 GEL samples, not on the whole cohort of 18,822 samples. This is unrelated to **Extended Data Figure 3**.

3) PRRDetect

In this also entirely separate section of the manuscript, we have developed PRRDetect as a multinomial classifier for polymerase dysfunction and MMRd.

TRAINING of PRRDetect

It was trained on **571 GEL** samples (a subset of the 4,775 GEL cancers), where

- i. 291 were confirmed to have driver mutations/ IHC staining.
- ii. 280 were negative set.
- iii. The training cohort of 571 went through a ~ 7:3 split for training.

These samples have confident PRRd labels. Ground truth is solid for all samples.

VALIDATION of PRRDetect

RTR.v4

PRRDetect was validated on an **independent cohort of 504 ICGC breast samples for which PRRd labels were available** (Line 360). This cohort was used specifically for validation because we are confident of the PRRd label of all samples, informed by IHC staining and careful curation of all drivers. We know that we are not missing any true PRRd samples. These samples have confident labels. Ground truth is solid for all samples.

Also note that during this revision of the manuscript, we have expanded on our validation cohort to include additional samples (not used for PRRDetect training) for statistical power. See #Point 14 for details.

GENERAL APPLICATION

PRRDetect was then applied on ICGC and Hartwig samples ($n=1,335$) (starting line 367) to make illustrative descriptions of generalisability. For these samples, the ground truth status is not (completely) known. We only have driver information that has been published by others. These samples do not have confident labels. Ground truth is not so solid; hence we have not used them for validation.

We showed, however, the concordance/discordance from different prediction algorithms, and whether predicted PRRd samples have annotated driver mutations or not (**Figure 4e**). Manuscript lines 367-385. Please note, this is not the same thing as testing or training.

Additionally, all samples used (whether in training or validation, whether positive or negative, with driver and/or IHC) are all annotated in **Supplementary Table S7** (former S6).

#Point 8

8. Related to above, it is not clear whether new ICGC breast analysis was done on samples included in the ICGC analyses shown in Extended Fig. 4.

The former **Extended Fig 4** is now revised as the new **Extended Data Figure 3**.

No. The ICGC breast samples used for PRRDetect testing/validation were not used in any analyses in **Extended Data Figure 3**. In fact, **Extended Data Figure 3** has nothing to do with PRRDetect. It is simply showing what the distribution of indels are across all cohorts, according to the COSMIC-83 and the new 89-channel system. See also #Point 7 – “1) The 89-channel system”.

Extended Data Figure 3 is not related to PRRDetect at all. It does not bias the PRRDetect analysis because it not related to the PRRDetect analysis.

#Point 9

9. It is still unclear how gold-standard labels are derived - namely what are the criteria for labeling a sample as PRRd or PRR proficient. Methods currently refer to "manual curation" for GEL samples and for the remaining samples, "true labels were established based on IHC of four MMR proteins and driver mutations". The authors should offer much more detail about these criteria and how frequently each was applied (in how many samples IHC vs. driver mutations vs. combinations was used to determine the label, including whether / how allelic status of driver mutations was assessed).

Regarding the term “manual curation”

Please note the term “manual curation” for identifying drivers is not a negative thing. This means that not only was a driver identified through bioinformatic methods, we will have also:

- checked that the other allele was lost if it is a tumor suppressor and
- we physically inspected the variant using IGV.

This, in our view, is being thorough and doing our due diligence. It is not and should not be viewed in a negative way.

In any case, we have removed the words “manual curation” from the sentence on line 350.

True labels for each sample

Note that a full, comprehensive list of every sample, whether they were used for training or validation, whether they had IHC or drivers is fully available in **Supplementary Table S7** (former S6). Note that:

RTR.v4

- A mutation is only considered a driver in any of the MMR repair proteins (*MLH1*, *MSH2*, *MSH6*, *PMS2*) if it has also lost the wild-type parental allele and is thus biallelically null. Mutations in *POLE/POLD1* are only annotated as drivers if they are previously reported mutations in functionally relevant domains. This is what we mean by “curated” drivers - we have gone the extra mile to ensure that these are not simply passenger mutations that have hit these genes by chance.
- Samples were labelled with an IHC staining result if one was available. Not all samples have had IHC.

Thus, a positive PRRd label is assigned to a sample if one of the two criteria (driver or an abnormal IHC) are met.

That we have then additionally checked to ensure that these gold-standard labels are unambiguous, must surely be a strength, not a weakness. Samples that do not have the right gold-standard label and are not in the right category would be detrimental to training an algorithm. Hence, we also double checked that samples in the training cohort of 571 samples were unambiguously PRR-deficient or PRR-proficient by ensuring that the relevant signatures matched the labels, which they did.

#Point 10

10. Specifically, how were true labels derived for samples / cohorts (like ICGC / HMF) where presumably IHC labels are not available. (for example, Figure 4e). I presume it's using driver status. But then the authors refer to the false negative rate of using strictly driver events (66%, line 365).

The reviewer was correct in saying that IHC labels were not available for ICGC/Hartwig cohorts. Hence, in line 369, we stated that “**Based solely on available published driver information, ...**”. Where IHC is not available, we are dependent on driver statuses reported by published ICGC or HMF studies. Note that many samples may be MMR-deficient through promoter hypermethylation of *MLH1* and will not thus show a genetic “driver”. Hence, the point at the end of the paragraph “**should customary targeted sequencing strategies be solely used to identify driver events associated with these deficiencies, a significant proportion of cases (66%) could be missed.**”

That’s because targeted panels will only “see” the genetic driver mutations in ~33% of cases (17 of the 50 predicted positive cases had reported drivers) and will miss all the MMRd/polymerase dysfunctional cases that are due to promoter hypermethylation or other causes.

This paragraph, by the way, is not a training or validation point for PRRDetect. This is simply a demonstration of the biological/clinical insight that if we stuck with targeted panels of MMR/polymerase genes, we would miss a significant proportion of the cases. Given the lack of clarity, we have adapted the manuscript to clarify the assumed context: “the assumption of PRRDetect predicted results being true”:

line 383: “**Should PRRDetect predictions be all true, and customary targeted sequencing strategies be solely used to identify driver events associated with these deficiencies, a significant proportion of cases (66%) could be missed.**”

#Point 11

11. Reading through methods, signature analysis seems to have been used to pick PRRd and non PRRd samples for GEL (“samples .. were confirmed to lack driver mutations .. and displayed no evidence of signatures associated abnormalities” (lines 881-883). Were signatures (or the lack of) part of the criteria used to determine PRR proficient samples? If so, was this done only for training or for testing? If so, this could obviously bias either training and testing or both.

Please see response to **#Point 9** above. **To define the ground truth set of PRRD and PRR-proficient samples, high-confidence driver mutations (including allelic status) and/or IHC were used.**

Having a clear ground truth set is incredibly important. Thus, to be exacting and highly conservative, we ensured that there was no ambiguity at all in the ground truth set: So, we manually checked and were sure that:

- True positives really did have drivers and the associated signatures.
- True negatives really did not have drivers or any hint of any similar signatures.

It was not a criterion. It was a simply a check. We are unclear why this is an issue. This would usually be preferable
over the alternative, of not checking.

**#Point 12**

12. From which cohorts are the numbers in lines 350-358 derived (sensitivity, AUC). Cited figure 4 panels (c,e-f)
show ICGC / Hartwig results but unclear how the true labels were defined and hence how sensitivity / AUC were
computed for these cohorts. Also 4c labels these cohorts as "application" but it appears that they are being used
for testing. If so, then why not include them in the ROC analysis (Figure 4d) ?

- • **Figure 4c** is a diagram of the workflow. It is not a result. It is simply telling the reader what samples and
numbers are in the various cohorts, and which cohort is used for which purpose.
- • **Figure 4d** is the validation of PRRDetect performed on 504 independent ICGC breast samples as this is the
cohort with highly curated, confident PRRD labels for us to calculate an accurate AUC. **Figure 4d** has now been
revised to include more samples (see **#Point 14** below).
- • **Figure 4e** is not validation – it is simply a graphical representation of PRRDetect predictions contrasted to TMB,
MSIseq, MMRDetect and drivers in ICGC/Hartwig. These samples are not being used for PRRDetect
training/testing or validation at all (see **#Point 7**). The numbers in the original lines 350-358 relate to the
‘compare and contrast’ between the different classifiers.

As this was confusing, we have made the following changes to the manuscript, starting from line 369:

“Based solely on available published driver information for PRRd status, PRRDetect could distinguish Pol-dys and
mixed samples from polymerase-proficient samples with an AUC of 0.999, compared to an AUC of 0.83 if TMB (10
muts/Mb) was used. Overall, PRRDetect performed superiorly to current biomarker strategies, correctly predicting
all Pol-dys, MMRd/Pol-dys samples and missing two subclonal MMRd samples. MSIseq missed six of 43 PRRDetect-
predicted MMRd, two of four mixed MMRd/Pol-dys cases while displaying poor concordance for detecting pure
Pol-dys cases. Unsurprisingly, PRRDetect captured all MMRDetect positive cases. However, MMRDetect failed to
identify all PRRd cases as it was not designed to detect Pol-dys/mixed phenotypes and missed 7 MMRd samples.
We also noted multiple instances of PRRDetect positive cases with no associated driver, which carries clinical
significance: of 50 PRRDetect positive cases, 39 were MMRd (eight had an associated driver mutation), seven were
Pol-dys (all had driver mutations in polymerase proofreading domains), four were predicted as mixed MMRd/Pol-
dys (two had exonuclease mutations, none had MMR drivers). Should PRRDetect predictions be all true, and
customary targeted sequencing strategies be solely used to identify driver events associated with these
deficiencies, a significant proportion of cases (66%) could be missed.”.

**#Point 13**

13. It is unclear which groupings refer to algorithm outputs rather than true labels (meaning those derived from
manual curation, IHC, and / or driver analysis). This is additionally complicated by the fact that the authors have
published a previous algorithm MMRdetect with overlapping outputs to PRRdetect. For example in 4f and 4g the
caption implies that some of the labels (MMRd, Poly-dis) are algorithm outputs, however in the previous panel
MMRd and Poly-dis refer to the true labels. Also in Extended figure 10, among other places in the manuscript. It
would help to use uniform and unambiguous terminology to make this distinction clear across the manuscript.

**Figures 4e,f,g and Extended Data Figure 10** are all PRRDetect algorithm outputs. These were all specified in the
figure legends, starting from line 552 and are copied here for you:

“**e, PRRDetect results** of $n=1,335$ ICGC and Hartwig cancers, ordered from the lowest to the highest prediction
probability across the x axis (left to right) for polymerase-dysfunctional samples (orange), combined mismatch-
repair deficient (MMRd) samples (blue), and MMRd samples (purple). Negative samples were ordered by TMB in
increasing order from left to right. Results of MSIseq, MMRDetect, cancer gene driver annotation, and cancer tissue
origin are labelled at the bottom tracks. Dashed rectangle highlights the extent of false positive overcalling if using
TMB > 10 muts/Mb as a cutoff.

**f**, Concordance of calls among TMB-H (> 10 mutations/Mb), positive exposure to SBS signatures that impart
hypermutation, and **PRRDetect prediction** across $n = 1,335$ ICGC and Hartwig cancers.

RTR.v4

g, Concordance of calls among TMB-H (> 10 mutations/Mb), positive exposure to SBS signatures that impart hypermutation, and **PRRDetect prediction** across n = 4,775 GEL tumors.”

However, given the query and the potential lack of clarity, we additionally labelled **Figure 4** panels and **Extended Data Figure 10** with “PRRDetect” on the figure panels:

To make it impossible to miss, we also incorporated main text changes to clarify PRRd predicted labels in:

Line 389: “just over a tenth of 459 cases classified as TMB-High (50/459, 10.9%) had **predicted** PRR dysfunction”.
Line 398: “We found that out of 1,371 TMB-H cases, just under half the cohort (677, 49.4%) **was predicted** as having MMRd and/or polymerase dysfunction (Fig. 4g).”

#Point 14

14. The ICGC breast cohort used for validation (Figure 4d) does not appear to have many positive (PRRd) samples, just judging by the shape of the curves. How many Are the results in Figure 4d statistically significant? In other words, is the AUC generated by PRRdetect significantly higher than the previously published methods including MMRdetect?

Indeed, the ICGC breast cohort has only 10 PRRd samples. PRRd is rare in other cancer types but we chose to use a different, independent cohort from the training set for validation, and where we have solid, confident PRRd status of the samples.

PRRd is more common in colorectal and uterine cancers. However:

- We cannot reuse the colorectal/uterine cancers from GEL cohort because as the reviewer has argued above, we shouldn’t reuse the training set for validation.
- ICGC colorectal (n=52) and uterine cohorts (n=44) are very small, because WGS has not been performed much in these tumor types, and crucially, there are no MMRd samples (n=0) based on driver information. There are 7 colorectal cancers and 1 uterine cancer with polymerase driver mutation. That’s fewer than the validation cohort that we did use.
- The colorectal and uterine cohorts of Hartwig are also not large (n=379 and n=42, respectively).
- More critically, the ground truth labels of samples in these additional cohorts are not solid (allelic status not defined, IHC not available, and risks not being informative).

However, to increase the sample size of the validation cohort, and hence, the power of PRRDetect validation, we have now added to the n=504 breast cohort, an additional n=847 GEL cancers (see revised **Figure 4c** below) with confirmed PRRd status for 104 positive and 743 negative samples. Critically, these samples are completely independent, and never used in PRRDetect training previously.

With more samples (also more PRRd cases), we could now see clear separation of the AUCs (revised Figure 4d above). PRRDetect does perform significantly better than MMRDetect ($P < 2.2e-16$), MSIsseq ($P = 9.773e-05$), and TMB (10 muts/Mb) ($P < 2.2e-16$).

#Point 15

15. The previously published MMRdetect algorithm seems to very comparable outputs to PRRdetect (Figure 4e). This is not a coincidence, since PRRdetect meant to capture MMR dysfunction as well as polymerase dysfunction, as well as a mixture of the two. However, polymerase dysfunction is very rare, as extended figure 10 shows, so the number of cases this will affect is small. Hence it begs the question as to what is the incremental advance of PRRdetect, especially as a clinical tool. Does it substantially improve both MRRd and polymerase deficiency detection, or just one of the two?

First, MMRDetect was only designed to detect MMRd phenotypes. By contrast, PRRDetect detects three phenotypes: MMRd, polymerase dysfunction / (pol-dys), and mixed MMRd/pol-dys phenotypes. This is already an advantage.

Second, compared to MMRDetect, PRRDetect improves on both MMRd and Pol-dys detection.

a) In the ICGC/Hartwig cohort, MMRDetect classified 34 samples as MMRd. In fact, 2 out of 34 were mixed MMRd/Pol-dys and identified as such by PRRDetect. By contrast, PRRDetect classified 39 samples as pure MMRd. Thus, MMRDetect missed the following 7 MMRd samples:

- 1) 9fc5b5c7-3973-42b4-8710-454de0cb5b50 Skin (SBS7 dominated)
- 2) HMF003589A - CNS
- 3) HMF003436A - CNS
- 4) HMF001534A - Skin
- 5) HMF002693A - Lung
- 6) HMF000572A - Colorectal
- 7) HMF000104A - Colorectal

So, PRRDetect has increased sensitivity and specificity over MMRDetect. Yes, this is a small number, but this is a limited study. If this were extended to a population level incidence of cancer, the numbers would be more substantial in clinical practice.

b) In the validation cohort, MMRDetect missed the following 4 samples:

1. PD5937a (APOBEC hypermutator)
2. GEL-2414040-11 (SBS6)
3. GEL-2495139-11 (mixed)
4. GEL-2925291-11 (mixed)

These samples were missed by MMRDetect because MMRDetect uses the cosine similarity between the SBS profile of tumor and that of MMR knockouts as one of its input parameters. In addition to having the expected MMRd-

RTR.v4

associated SBS, these samples were either subclonal for MMRd or were also dominated by other processes, such as UV exposure SBS7 or APOBEC SBS2/13 hypermutation, causing the MMRDetect-calculated probability to be penalised, resulting in false negative calls by MMRDetect.

Additionally, MMRDetect uses only the experimentally derived MMRd-SBS signatures (*i.e.*, SBS26 and SBS44), precluding the possibility of identifying samples with MMRd-SBS signatures other than SBS26/44, such as SBS15 and SBS97. PRRDetect uses SBS6, 26, 44, 15 and 97 – all SBS signatures that have been associated with MMRd, for MMRd prediction and indel signatures as well.

Critically, all MMRDetect-predicted MMRd samples are also captured by PRRDetect. The prediction results of all algorithms used, wherever possible, of all samples analysed in the manuscript, were provided in **Supplementary Table S7 (former S6)**.

Rarity is not a reason to dismiss classification. Indeed, the whole purpose of a tumor-agnostic process such as whole genome sequencing is to permit one assay to be read by many different tools to give a comprehensive read-out. Indeed, this is already in action today. Following WGS, each genome is read for all its drivers, gene fusions, germline mutations, somatic signatures, TMB, Myriad LOH score, HRDetect score, and PRRdetect scores as a matter of course.

It means that a clinician does not have to specify the assay. All assays can be run and the clinician receives a full genome report with all possibilities reported. Thus, while *POLE*-mutated cancers may be relatively rare, they are not negligible and make up about several hundred cancers per year in the UK. For the rare patient whose cancer is in fact, *POLE*-mutated, the precise diagnosis may make the difference to receiving the right chemotherapeutic option of otherwise.

We would like to thank the reviewer for raising this point, however, as it has led us to make some changes to the manuscript lines 378-385, line 399: “**only ~50% of them have an identified driver**”.

#Point 16

16. Specifically, how much better is PRRDetect at detecting polymerase deficiency than just the total SBS6 + SBS10 mutational burden in clinical tumor samples. While MMR is indeed hard to specifically detect with SBSs only, polymerase dysfunction detection seems more straightforward.

SBS6 is not due to polymerase dysfunction. Perhaps the reviewer was referring to using solely SBS10a (*POLE*) and SBS10d (*POLD1*) to detect polymerase dysfunctional samples. In fact, we did contrast PRRDetect’s performance against all other possibilities. The results were provided in **Extended Data Table 1**, and copied below for you:

Extended Data Table1 Performance metrics of attempted prediction models using different input parameters

No	Model	Normalization	Training ACC WM	Test Set MED.AC C	α std	λ std	Med Coef. Std	Median Multiclass AUC on Test Set	Best α WM	Best λ WM	Multiclass AUC
Sigs	Prop	0.9947	0.988	0.46	0.0005	0.551	0.9955	1	0.001	0.9975
Sigs+R	Prop	0.9947	0.99127	0.433	0.00052	0.4611	0.9961	0.05	1E-07	0.9977
Sigs+TMB	Prop	0.3748	0.377	0.2	0.02	0	0.5	0.65	0.099	0.5
Only_IND	Prop	0.9807	0.9825	0.333	0.00067	0.708	0.9844	0.1	1E-07	0.9932
Only_SBS	Prop	0.965	0.9651	0	0	0.16	0.9734	1	1E-07	0.968

614
615

RTR.v4

Shaded row shows the model selected as the final PRRDetect classifier.

Model:

Sigs, using both SBS and InD signatures;

Sigs+R, using SBS, InD signatures, and (total indel count/total SNV count) ratio;

Sigs+TMB, using SBS, InD signatures, and tumour mutation burden (TMB);

Only IND, using InD signatures only;

Only SBS, using SBS signatures only.

Current PRRDetect model (no.2 in the table above) was chosen because of its highest test set median accuracy, AUC, and stability over other models with single parameters (*i.e.*, SBS only, InD only).

#Point 17

17. Similar for cell models - the study focuses on indels, but how much additional accuracy do indels provide on top of SBS6+10 for polymerase dysfunction detection in these models? Extended Figure 2g shows cosine similarity but does not address this specifically.

This question is very similar to #Point 16. SBS6 is not used to define polymerase dysfunction. SBS10a may be used on its own to denote polymerase epsilon dysfunction. However, like any other classifiers, with additional strands of orthogonal evidence, (*e.g.*, another signature from another class of mutation), the ability to be more specific is higher. This is a well-known principle – multiple strands increase specificity of signal. In situations where the polymerase dysregulation is in a sample with low tumor cellularity or is in a subclone, the addition of orthogonal signatures is critical for accurate classification.

#Point 18

18. How does the new taxonomy, signature system, and classifier perform on mutants profiled in the 2021 Nature Cancer paper? Given the overlap of the previous WGS data on knockouts (including MMR knockouts), it would seem fitting to analyze this.

This analysis was performed previously and included in the manuscript already.

Please see **Extended Data Figure 8b** and **Extended Data Figure 9** where we showed that across knockouts of overlapping genes in two different human biological systems (RPE1 and iPSC), the genotype-specific experimentally derived signatures are extremely similar. This is demonstrated plainly through hierarchical clustering in **Extended Data Fig 9**, which puts all the relevant iPSC and RPE1 genotypes together, with cancer derived InD signatures.

#Point 19

19. Taking a step back, it would seem more interesting to build a classifier to diagnose repair dysfunctions more specifically than what the authors are doing here, rather than just 3 dysfunction categories, two of which seem quite rare (Pol-dys, MMRd/Pol-dys). Why can't the RPE1 data be used (in combination with previously published 2021) and cancer genomics data to build such a classifier? This would tie together the two currently disparate parts of the paper.

We agree that academically, it is terrifically interesting that there are such marked gene-specific signatures. Mismatch repair deficiency is not simply “MSI” or “a high TMB”.

- We show this experimentally
- We find evidence of this in the cancer datasets and
- We made this assertion in paragraph starting lines 295 and 428, “eight gene-specific MMRd and polymerase-dysfunction InDs. ...”

However, clinically, gene specificities are not required for treatment. All MMR gene defects are treated equally. Whether this is the best way to treat patients or not is another matter that is beyond our ability to resolve/and the scope of the manuscript. We see the same situation with *BRCA1/BRCA2*. Currently, all the customary assays simply denote tumours as *BRCA1/2*-deficient or proficient and do not require distinguishing between *BRCA1/BRCA2*

RTR.v4

mutated cancers. Yet, they are different genes with different physiological functions, producing subtly different
signatures. This is a known generic problem, and not something that we can address.

However, it is an important philosophical point. Perhaps the day will come where we need to highlight to
clinicians/trialists, that there are gene-specificities and perhaps this plays a role in treatment outcomes. If the
reviewer is happy with this, we are happy to make the point in the discussion which we think will elevate the
argument.

**Line 446: "Finally, our classifier currently does not distinguish between the MMRd genotypes despite clear
differences between *MLH1*, *MSH2*, *MSH6* and *PMS2*, for example, as there is no clinical indication to do
so. However, should it become clinically important to distinguish between these genes, it shall be possible
to do so."**

**#Point 20**

20. The strand bias signals are interesting, though it seems that more could be made of this. A strand bias not only
reflects a difference in mutation rates but also a pyrimidine vs. purine preference for mutations arising on that
strand. In other words, while the convention is to align mutational strands to pyrimidines, a leading strand bias
implies either an increased pyrimidine mutation rate on the leading strand or an increased purine mutation rate on
the lagging strand. Under the biological assumption that dysfunction of POLE and POLD increases mutation rates on
the leading and lagging strand, respectively, their data implies that both POLE and POLD dysfunction preferentially
causes T rather than A insertions. How do the authors explain this?

When strand asymmetry is observed, there are two possible reasons (covered elsewhere extensively). There may
be a bias of DNA damage or there is a bias in DNA repair.

For environmental mutagens which are known to cause DNA adducts that require the activity of transcription-
coupled repair, we can confidently assert whether the DNA damage occurred on a purine or a pyrimidine base,
based on **transcriptional strand asymmetries** (and not replication strand asymmetries).

- • For example, C>T mutations of UV light, when inspected for transcriptional strand bias, shows an excess of C>T
mutations on the non-transcribed strand more than the transcribed strand. This is due to the activity of
transcription-couple repair (TCR) preferentially repairing DNA damage on the transcribed strand. This also fits
with the known information that UV causes covalent modifications on pyrimidines.
- • By contrast, C>A mutations of tobacco smoke, when inspected for transcriptional strand bias, shows an excess
of C>A mutations on the transcribed strand more than the non-transcribed strand. Because the imprint of TCR
activity is the excess of mutations on the non-transcribed than the transcribed strands, we can know that the
true DNA damage must be on the G (purine), and not on the C. This is in keeping with known biochemistry of
guanine adduct formation caused by polycyclic aromatic hydrocarbons.

For endogenous mutagenesis caused by defective repair processes, we cannot quite make the same assertion as
above. However, we have argued and shown (which we think the reviewer agrees with), that there is a bias of
mainly T insertions on the leading strand for *POLE-mut* (which is known to be the major replicative polymerase of
the leading strand, and the opposite is seen for *POLD1* (which preferentially replicates and proof-reads the lagging
strand). Biochemical analyses have shown that B-family polymerases are particularly proficient at discriminating
between correctly and incorrectly paired bases at template adenine (PMID: 21036870), *i.e.*, less accurate on
replicating template thymine.

In agreement with published work (PMID: 33764464), our data suggests that the *POLE/POLD1* polymerase mutants
frequently accumulate +A insertions (on the nascent strand) particularly when replicating through T-homopolymer
of specific tract lengths (5-7nts) (**Supplementary Table S3**). We note that we have not overtly made the point
about T insertions in the manuscript. Given that the reviewer has kindly raised it, we have now included it in:

**Line 119: ..., specifically T insertions at polynucleotide repeat lengths of 5-7 nts (Supplementary Table 3).**

**Line 120: This is in keeping with the hypothesized preferential activity of Pol ϵ and Pol δ in leading and lagging
strand synthesis respectively, suggesting that *POLE/POLD1* mutants tend to accumulate 1bp A insertions on the
nascent strand while replicating through 5-7 nts poly-T-tracts (PMID: 33764464, 21036870). This aligns with the**

RTR.v4

proposition that polymerase ϵ and δ are more proficient at detecting incorrectly paired bases at template adenine (PMID: 21036870).

#Point 21

21. Since the study does not include immunotherapy analyses, it does not seem appropriate to mention immunotherapy in the abstract. To the reviewer, it does seem likely that the modest improvements in PRRd accuracy provided by the classifier here will improve prediction of immunotherapy outcomes.

Clinically today, the reason for identifying MMR-deficient and PRR-deficient cases is because these patients are candidates for immunotherapies. Hence, we have stated originally in the abstract:

“Furthermore, we develop a highly specific classifier of PRRd status in tumors, PRRDetect, with potential implications for immunotherapies”.

We have **not** said “...with superior performance in immunotherapies”, which would have required an analysis. The use of the word “potential” communicates the tentativeness; it implies that it is possible or likely, but has not yet been realised or confirmed.

We are happy to defer to the editor on this point but we are often encouraged to say what “impact” a manuscript has. Perhaps we could suggest this change:

“Furthermore, we develop a highly specific classifier of PRRd status in tumors, PRRDetect, ~~with~~ given potential implications for immunotherapies”.

#Point 22

22. Authors cite space considerations, but it seems that some of the main panels (such as 1f, 1d, 3b, 4a, 4e) could be moved to extended data or substantially condensed to show key signals. Also NG allows for up to 7 figures.

We’re happy to discuss the path forward with the editor/reviewer.

We thank you for all your feedback and comments, which have helped us clarify issues and improve the manuscript.

RTR.v4

Reviewer #1:

Remarks to the Author:

The authors have provided a revised version of their study "Redefined indel taxonomy reveals insights into mutational signatures".

As before, the study begins with mutational patterns in cellular models of post replicative repair deficiency (PRRd) and ends with a revised 89 channel indel taxonomy from which they derive signatures and apply to pan-cancer data, including training a new PRRdetect classifier.

#Summary Section

Overall, while some clarifications have been made in the text and rebuttal, there is still a surprising disconnect between two parts of the paper - looking at cell models and cancer genome data analysis. Especially now that the authors have clarified that the revised indel taxonomy is based on cancer genomes, as is the PRRdetect training.

The flow of the paper seems to then use the cell models mainly to show that COSMIC-83 is inadequate, after which these unique data are sort of cast aside as the study moves to a second technical cancer genomics analysis that might stand alone as a separate paper.

Thank you for taking the time to review our paper again.

We find it a little perplexing that the experimental analyses and the cancer analyses in our manuscript were viewed in a dichotomous way. The principle of performing experiments and then assessing generalisability in a real-world cancer dataset is not uncommon. This step-wise approach of presenting data has been used by others recurrently (PMID 35140396 is a good example of one exploring indel signatures published in Nature (amongst many others)).

A dichotomous description casts a slightly negative aspersion – and here we would respectfully disagree that the cell model data are used distinctly from the cancer data. The analyses are integrated; the cell model data are central to the narrative and fully integrated with the cancer data:

- Section 1 on “Diversity of indel patterns in PRRd” – here, data from cell models are the main subject matter.
- Section 2 on “Limitations of current indel taxonomy” – here, data from cell models are the main subject matter and demonstrate the problems with the current taxonomy and that a change is needed. This is an experimentally-driven exercise. It is robust and agnostic.
- Section 3 on “A new framework for classifying indels” – data from cell models are definitely not cast aside. The new indel framework is proposed based on the logical argument of using flanking sequence context (amongst other features).
 - The new framework *is assessed in cancers* AND
 - then in paragraphs 3-6 of the same section of the manuscript, *the experimental data are used alongside cancer data to examine the technical strengths of the new classification system.*

Thus, the experimental data are fully and comprehensively utilised and integrated.

Importantly, the final classifier PRRDetect seems to offer incremental classification benefit over their previous MRRdetect algorithm and SBS signatures, especially given the rarity of polymerase dysfunction and the ease of classifying it with SBSs. It would seem like their unique cell model dataset and new indel taxonomy system would be able to more precisely diagnose the repair dysfunction (for example to a specific protein defect). Such a demonstration would elevate it much more clearly above the state of the art (including using the burden of SBS6 and 10 to diagnose polymerase deficiency).

Here, we humbly disagree with the notion that “incremental benefit” is expected because of clinical rarity of polymerase dysfunction. Rarity is not an adequate argument for dismissing a classification process.

- First, rarity in one tumor-type may not translate to rarity in other tumor-types (e.g., Mismatch repair deficiency).

RTR.v4

- Second, 1% of a common disease like cancer, translates to many thousands of people with cancer.
- Third, most genetic diseases are extremely rare, but this does not warrant precluding them from scientific investigation or accurate classification. To date, there are no models to classify polymerase dysfunction or combined polymerase dysfunction and MMR deficiency in a single assay. PRRDetect is the only classifier that does so, and performs better than existing biomarker (*i.e.*, TMB) and algorithms in detecting samples with MSI.

We do agree that gene specificities are a very interesting feature of mutational signatures and we acknowledge this in the discussion. *However, to date, there is no clinical indication to distinguish between gene-specific signatures in MMR deficiency (MMRd). All MMRd samples regardless of their originating genotype, receive the same therapy. Also see answer to #Point 19.*

Specific critiques

- The indel taxonomy still seems arbitrary, and not really justified by biological principles or data (including the RPE1 data), especially to a general genetics reader. While it is clear from their paper that COSMIC-83 is lacking, it is not convincingly shown that the proposed taxonomy is the natural successor.

If we could discuss the logic here: By extension of the reviewer's argument, the COSMIC-83 indel classification would be entirely arbitrary and not justified by biological principles/data, because:

- It is not constructed through examination of any underlying data, as plainly demonstrated in panel a of **Extended Data Figure 3 (Former Extended Data Figure 4)**.
- Because of limited power, the channels were arbitrarily capped at homopolymers of 6bp in lengths.

We wonder why that is considered reasonable. Yet our position, which is a systematic experimental demonstration of the inadequacies of the current indel classification system, with logical suggestions of improving it by extending the homopolymer run to 9bp, introducing flanking sequence context (a system used for substitutions), justifying it in the experimental data and showing that it works in cancer, is not considered adequate and is dismissed rather easily.

We are suggesting our taxonomy as an alternative methodology for indel classification, which could reveal more biological granularity and is able to disambiguate PRRd signatures and/or uncover new signatures, as demonstrated by our RPE1 experimental data and cancer data. We have not, at any point, pitched the new classification system as the "natural successor" to COSMIC-83. Further, we acknowledge in line 454 that "**Nevertheless, optimal classification remains an active research area, and alternative schema could unveil additional mutational processes in the future**".

Our proposed classification expands on the current COSMIC-83 taxonomy, with key additions, considerations and modifications as follows:

1/ Incorporating the 5' and 3' flanking bases for 1bp indels.

2/ Extending the cap on homopolymer length from 6 to 9bp, to allow for better discriminatory power as we observed that there are more indel signals at longer homopolymers (**Extended Data Figure 3a,e**), which may offer the resolution required to tease out distinct biological phenotypes. Note that the frequency of homopolymers in the genome decreases with increasing homopolymer length (see **#Point 2** figure). There's a trade-off between being exhaustive to account for all the homopolymer/indel lengths and conserving power for signature extraction.

3/ Though longer repeats (up to 9bp) are captured in our classification, to improve the indel *signal-to-channel* ratio, we grouped the homopolymer lengths into short (0-4bp), medium (6-7bp), long (8-9bp) based on how they are distributed (**Figure 2a, Supplementary Note 1, Extended Data Figure 3**), to prevent over-sparsity. This was an informed decision based on the distribution of indels/homopolymer tracts in the reference genome. This was not arbitrary.

4/ We aggregated channels where indel signals were low across the pan-cancer cohort data because they would contribute limited value (*i.e.*, no point in having empty channels) while costing statistical power for analyses.

RTR.v4

These are thoughtful deliberations based on the distribution of mutations (actual data is in **Extended Figure 3**) across
multiple cohorts involving most human cancer types and are not arbitrary decisions. Our approach is data-driven and
unbiased.

From a large body of published work, we know that indel formation is commonly driven by the underlying sequence
motifs. While many in the community have been introduced to COSMIC-83 for signature analysis, it is not as widely
adopted as the substitution channels because it has been problematic, and simply because it was the first, and has
been introduced by the Sanger/COSMIC team, does not mean that it should be dogma. Many have struggled and
found it challenging to interpret the COSMIC ID signature results. This is because COSMIC-ID lacks detailed sequence
context information, resulting in peculiar signature assignments, leading to uncertainties in the interpretation of
results (PMID: 38361034). This is not only apparent in our RPE1 dataset presented here, but also has been shown by
others (PMID: 35320711, 35140396) that the COSMIC-83 is not able to resolve many biological phenotypes.

It is with these biological observations as motivations and also based on additional insights gleaned from many
mechanistic studies (PMID: 30982602, 32661091, 35320711, 35140396) showing that a lot of mutational processes
and mutagens show a predilection for specific sequence contexts to form adducts/damage and cause indels (*e.g.* UV
at TT, etc), that we expanded on the current COSMIC-83 framework to include the considerations outlined above.

Therefore, we maintain that our classification system is not arbitrary. It is informed by biological observations, and
guided by the understanding of the underlying data distribution. It is also noteworthy that we, in addition to
capturing all previous COSMIC-83 signatures with confirmed aetiologies, also identified multiple new signatures in
the 7 GEL cancer types, while providing additional insights/dimensions to guide signature interpretation.

We stand by the validity, robustness, and biological value of our approach.

**#Point 1**

1. One way to present the new taxonomy is as a modification / improvement to COSMIC-83. There seem to be two
key differences with COSMIC: First, longer repeat lengths (up to 9) are captured. Second, the base context to the 5'
and 3' of a mononucleotide repeat is considered. Is this true? Can these two choices be concisely justified by main
text data analysis or cited biological first principles?

These are broadly correct but there is more complexity (which the reviewer has mentioned in **#Point 3**).

We present below an alluvial plot mapping the COSMIC-83 features to our InD-89 system, which we made it into
the new **Extended Figure 4** (and took out the former Extended Data Figure on strand bias plot, and instead
presented strand bias results as the new Supplementary Table 3 to keep to the figure limit). We kindly seek your
guidance on this.

RTR.v4

As can be seen above, there is no direct 1-to-1 mapping of COSMIC-83 channels to InD-89 channels. Predominantly, the InD-89 taxonomy expands upon the T indels (where most indel signals are concentrated) and condenses the longer indels ($\geq 2bp$) (where the signals are scant). Detailed mapping of the channels between the two classification systems and some examples were provided in **Supplementary Table S4 (former S3)**.

We have inserted the following statement to fall in line with the PCAWG classification:

Line 193: Overall, compared to the COSMIC-83, the 89-channel taxonomy expands upon 1 bp T indels (where most of the signals were) into a larger array of channels, and condenses longer indels and/or genome motifs that are infrequent in the genome (where signals were scant or non-existent) into fewer indel subcategories.

There is, however, far more complexity that makes our system more reproducible and robust, which the reviewer has highlighted in **Point 3** below.

#Point 2

2. Along these lines there is data in the supplement, including histograms of repeat lengths which actually show a substantial fraction of indels with repeat lengths of 9 or greater (for example 1bp T indel). Does it really make sense to collapse them? In other words, why draw the line at 9+? If the authors are looking to rethink the indel taxonomy, are they being bold enough?

This is not correct. The histogram shows indels up to repeat lengths of 9, and not 9 or greater. See below for the zoomed-in example of all possible 1bp T deletions. The same applies to all homopolymer, polynucleotide repeats (*i.e.*, repeat # ≤ 9)

1 bp T indel channels (81 subcategories)

It is not possible using today's genomic data pipelines of any institution, to confidently go beyond repeat lengths of > 9 because of the error rates in mutation-calling algorithms. Nearly all commonly-used genomic pipelines (Broad GATK, Dragen, Strelka, Platypus, Pindel, etc) have a post-hoc filter capped at repeat lengths < 10.

The contribution of indels at very long homopolymer tract length of >9 to indel signatures could be limited given that there is a much lower prevalence across the genome (see image above). Having said that, we agree that indels at longer homopolymers may be important and should be investigated at a later date when indel data obtained through newer lower-error/long read technologies become available, in quality and quantity. Thus, we have added the following statement:

Lines 432: The possibility of adapting the taxonomy in the future, to include features that are currently not explorable due to the limitations incurred by technological error rates, could also be revealing.

#Point 3

3. Are there any other important changes vs. COSMIC-83? If so, what motivates and justifies them? The remaining differences seem minor: These include collapsing of categories - for example the authors group all length 2 to 5 microhomology deletions into a single category, while COSMIC considers them separately. Complex indels are given their own channel, and ignored in COSMIC-83. There is also a new approach to indel segmentation, which perhaps may help reproducibility across indels called using different conventions. It is not clear whether COSMIC-83 already does this kind of segmentation.

Yes, the reviewer has highlighted a key difference and is correct in saying that our approach in classifying indels through a "segmentation process" does result in a more unified, reproducible, and standardised classification. Prior to our approach, because a binary threshold-based classification schemes was used (i.e., see "repeat-mediated" vs "microhomology-mediated" deletions in #Point 1 Figure, for example), there were ambiguous cases that arose that could be classified in either category. This has never been addressed previously. To deal with this ambiguity, our

RTR.v4

“segmentation” strategy explicitly identifies the smallest prefix sequence motif that displays a maximal repetitive relationship with the 3' sequence context and within the indel itself. This provides a standardized approach for relating indel sequence to the 3' sequence context. COSMIC-83 does not do that.

Examples:

Indels	InD-89	COSMIC-83
C AAATAAAATA AAATACCTAGGAAT	Del(3,):U(3,):R(3,9)	5:Del:M:5
G CCC CTCAGGATGATTCCTCCCTGG	Del(2,8):U(1,2):R(2,4)	3:Del:M:1

In the first example, AAATAAAATA is the deletion motif, AAATA is the smallest prefix sequence motif, the unit is 5 bp, and repeated 3 times. In COSMIC-83, it is classified as long deletions (>5bp) with 5bp microhomology. However, using our strategy, it is interpreted as a loss of 2 of the 3 AAATA pentanucleotide repeats.

In the second example, CCC is the deletion motif, C is the smallest prefix sequence motif, the unit is 1 bp, and repeated 4 times. In COSMIC-83, it is classified as 3bp deletion with 1bp microhomology. However, using our strategy, it is interpreted as a loss of 3 of the 4 x C mononucleotides.

Please also see answer to #Point 1. Details of segmentation and motivations/justifications were previously described in Supplementary Note 1. However, we agree that we have not adequately highlighted key differences between the classification systems and have included this in the abstract (Lines 32-35):

“we developed an alternative classification system that considers the 5' and 3' sequences flanking indels and includes informative motifs (*e.g.*, longer homopolymeric tracts) while removing uninformative channels. Our new cataloguing system ensures that every indel can be unambiguously classified into one of 89 indel subcategories and is capable of disambiguating gene-specific experimentally-generated signatures.”

We have also added text in the final paragraph of the Introduction to ensure that the distinction between our methodology and that of COSMIC-83 is clearly articulated (Lines 72-77):

“Additionally, motifs known to increase mutational vulnerability that are conflated by the current system (*i.e.*, homopolymer tracts >6nt), and genome-wide prevalence of such motifs were factored into our proposition. Our approach unambiguously classifies each indel into a specific subcategory.”

Thank you for raising this.

#Point 4

4. The average reader should be able to very readily grasp important features of the new taxonomy and key differences with COSMIC 83. The schematic in Figure 2a seems important for this and also to explain the channel bar plots. However, as it stands, it seems overly complicated and main differences with COSMIC-83 are not apparent. Perhaps it could be simplified, and the full version can be moved to extended data.

We wonder whether moving Figure 2a upper panel to the Supplement would make interpretation of the rest of figures challenging. Readers would have to flip back and forth to understand the full complement of the profiles, *e.g.*, in 2d. As Figure 2a serves as the basis for the signature plots in Figure 2d and Figure 3, we believe that having it as a main figure panel is essential.

One option is to contrast the two classification systems to highlight differences (see the alluvial plot shown in #Point 1). Detailed mapping of the channels between the two classification systems and indel examples were already provided in Supplementary Table S4. It is quite an illustrative plot. The alluvial plot contrasting the two classification systems is now provided and referenced as Extended Data Figure 4.

#Point 5

5. How are insertions with microhomology classified? Namely, insertions aligning with a spacer to flanking sequence. It does not seem like they have a category here.

RTR.v4

Insertions with microhomology are a rare feature (single digits if at all), even in HR-deficient samples, where they are alleged to be seen more frequently. *Critically, they cannot be unambiguously mapped to a single category.*

That's because templated insertions are:

- often complex (*i.e.*, they are accompanied by deletion at the junction). Majority of them would have been overlooked or mis-annotated using standard indel variant callers (PMID: 26657142).
- could also be templated from inverted repeats (both left and right flanks) and not be immediately at the junction of the indel, and unaccompanied by deletions.

Although we do not designate a category specifically for these complex events, they were segmented and classified using our unified framework into various non-overlapping "insertion" and "complex" categories. Due to the complexity of these events, they are best characterised by specialised algorithms (PMID: 32234782, 26657142).

#Point 6

6. Since the taxonomy is apparently independent of the RPE1 experiments, the authors may actually want to consider presenting the cancer genome analyses first and ending with the RPE1 data. In addition to more solid motivation for the indel taxonomy, this would require restructuring of the paper, but maybe provide an opportunity to close with some biological insights.

The taxonomy *is not independent* of the RPE1 experiments. The RPE1 experiments culminated in the evidence that shows that the current COSMIC-83 is inadequate. Once the taxonomy was designed, it was also tested on the RPE1 data to show the remarkable improvement in discerning the gene-specificity that is present in the data (**Figure 2b-d, Extended Data Figure 5**).

We respectfully disagree with this suggestion. Furthermore, first, it would not make sense to start with an adaptation of the COSMIC-83 system for no reason at all, and then use the experiments to back up our argument. Second, it might be seen as trying to "appear clever" *if we started by philosophising a change*. An alternative reviewer could easily ask in this scenario, "why did they think it was necessary to adapt it in the first place?".

Thus, we are concerned that restructuring would, in fact, be detrimental to the manuscript and is not the truth of how the thought process went either.

#Point 7

- There still appear to be some circularities in the classifier training and testing, including a concern about data leakage.

7. The new 89-channel system is derived from the entirety of pan-cancer data (18522 samples) which seems also to include samples used to test PRRdetect, including ICGC, TCGA, and HMF. Since these test cohorts (ICGC, TCGA, HMF) were used for feature engineering (Extended Fig 4, supplementary note) they are not independent. The feature engineering and training of signatures and classifiers should be done on a single cohort, for example GEL, leaving the other datasets for testing, including ROC / AUC analysis and calculations of sensitivity and specificity.

Note that the former **Extended Fig 4** is now revised as the new **Extended Data Figure 3**.

The reviewer has conflated several distinct pieces of work together in the argument above. There are three distinct pieces of work:

- 1) The demonstration of validity of the 89-channel system
- 2) Indel signature extraction from seven cancer types
- 3) The PRRDetect classifier development

These are different analyses. Historically, a paper would be done on (1) and (2). For example, the substitution signatures (Cell 2012a, Nature 2013) are an example of analyses similar to items (1) and (2). Later, some type of classifier might be created (often using past publicly-available data because that's what is available and we need something as a "gold standard"). For example, HRDetect (Nature Medicine 2017) – this would be similar to item (3). These are different analyses resulting in different papers. In this manuscript, we have argued for an alternative

RTR.v4

classification taxonomy (1) and shown how it would be used to find signatures (2). Item (3) is just like HRDetect, it is a separate exercise, and could easily have been a separate paper. These are simply different analyses.

1) The 89-channel system

The entirety of the pan-cancer data of 18,822 samples was *simply used to show the indel distribution across all categories from available cancer types across all three cohorts (GEL, ICGC and Hartwig)*. Because the same reviewer previously raised concern that our proposed framework was potentially “biased towards PRRD phenotype”, we presented the plots in the now **Extended Data Figure 3** to show that:

A/ indel distributions across InD channels are preserved regardless of cohorts or cancer types, and are not biased towards PRRd phenotypes or skewed towards hypermutator samples (Extended Data Figure 3e).

B/ Additionally, the new indel classification channels have data in all channels across all cancer types, in contrast to the sparse profiles with COSMIC-83 (**Extended Data Figure 3a-d**).

§ ratio, and importantly, shows that our final channel set does not bias towards hypermutator phenotypes (and hence PRRd samples), and reinforces that our approach is data-driven.

2) Indel signature extraction

In the main text (starting from line 251), we clearly stated that indel signature extraction *was performed on the 7 selected cancer types within the GEL cohort*. An indel signature extraction has only been performed on 4,775 GEL samples, not on the whole cohort of 18,822 samples. This is unrelated to **Extended Data Figure 3**.

3) PRRDetect

In this also entirely separate section of the manuscript, we have developed PRRDetect as a multinomial classifier for polymerase dysfunction and MMRd.

TRAINING of PRRDetect

It was trained on **571 GEL** samples (a subset of the 4,775 GEL cancers), where

- i. 291 were confirmed to have driver mutations/ IHC staining.
- ii. 280 were negative set.
- iii. The training cohort of 571 went through a ~ 7:3 split for training.

These samples have confident PRRd labels. Ground truth is solid for all samples.

VALIDATION of PRRDetect

PRRDetect was *validated on an independent cohort of 504 ICGC breast samples for which PRRd labels were available (Line 360)*. This cohort was used specifically for validation because we are confident of the PRRd label of all samples, informed by IHC staining and careful curation of all drivers. We know that we are not missing any true PRRd samples. These samples have confident labels. Ground truth is solid for all samples.

Also note that during this revision of the manuscript, we have expanded on our validation cohort to include additional samples (not used for PRRDetect training) for statistical power. See **Point 34 for details.**

GENERAL APPLICATION

PRRDetect was then applied on ICGC and Hartwig samples (starting line 367, $n=1,335$) to make illustrative descriptions of generalisability. For these samples, the ground truth status is not (completely) known. We only have driver information that has been published by others. These samples do not have confident labels. Ground truth is not so solid; hence we have not used them for validation.

We showed, however, the concordance/discordance from different prediction algorithms, and whether predicted PRRd samples have annotated driver mutations or not (**Figure 4e**). Manuscript lines 359-385. Please note, this is not the same thing as testing or training.

Additionally, all samples used (whether in training or validation, whether positive or negative, with driver and/or IHC) are all annotated in **Supplementary Table S7 (former S6)**.

RTR.v4

#Point 8

8. Related to above, it is not clear whether new ICGC breast analysis was done on samples included in the ICGC analyses shown in Extended Fig. 4.

The former **Extended Fig 4** is now revised as the new **Extended Data Figure 3**.

No. The ICGC breast samples used for PRRDetect testing/validation were not used in any analyses in **Extended Data Figure 3**. In fact, **Extended Data Figure 3** has nothing to do with PRRDetect. It is simply showing what the distribution of indels are across all cohorts, according to the COSMIC-83 and the new 89-channel system. See also **#Point 7** – “1) The 89-channel system”.

Extended Data Figure 3 is not related to PRRDetect at all. It does not bias the PRRDetect analysis because it not related to the PRRDetect analysis.

#Point 9

9. It is still unclear how gold-standard labels are derived - namely what are the criteria for labeling a sample as PRRd or PRR proficient. Methods currently refer to "manual curation" for GEL samples and for the remaining samples, "true labels were established based on IHC of four MMR proteins and driver mutations". The authors should offer much more detail about these criteria and how frequently each was applied (in how many samples IHC vs. driver mutations vs. combinations was used to determine the label, including whether / how allelic status of driver mutations was assessed).

Regarding the term “manual curation”

Please note the term “manual curation” for identifying drivers is not a negative thing. This means that not only was a driver identified through bioinformatic methods, we will have also:

- checked that the other allele was lost if it is a tumor suppressor and
- we physically inspected the variant using IGV.

This, in our view, is being thorough and doing our due diligence. It is not and should not be viewed in a negative way.

In any case, we have removed the words “manual curation” from paragraph starting from line 349.

True labels for each sample

Note that a full, comprehensive list of every sample, whether they were used for training or validation, whether they had IHC or drivers is fully available in **Supplementary Table S7 (former S6)**. Note that:

- A mutation is only considered a driver in any of the MMR repair proteins (*MLH1*, *MSH2*, *MSH6*, *PMS2*) if it has also lost the wild-type parental allele and is thus biallelically null. Mutations in *POLE/POLD1* are only annotated as drivers if they are previously reported mutations in functionally relevant domains. This is what we mean by “curated” drivers - we have gone the extra mile to ensure that these are not simply passenger mutations that have hit these genes by chance.
- Samples were labelled with an IHC staining result if one was available. Not all samples have had IHC.

Thus, a positive PRRd label is assigned to a sample if one of the two criteria (driver or an abnormal IHC) are met.

That we have then additionally checked to ensure that these gold-standard labels are unambiguous, must surely be a strength, not a weakness. Samples that do not have the right gold-standard label and are not in the right category would be detrimental to training an algorithm. Hence, we also double checked that samples in the training cohort of 571 samples were unambiguously PRR-deficient or PRR-proficient by ensuring that the relevant signatures matched the labels, which they did.

#Point 10

10. Specifically, how were true labels derived for samples / cohorts (like ICGC / HMF) where presumably IHC labels are not available. (for example, Figure 4e). I presume it's using driver status. But then the authors refer to the false negative rate of using strictly driver events (66%, line 365).

The reviewer was correct in saying that IHC labels were not available for ICGC/Hartwig cohorts. Hence, in line 369, we stated that “Based solely on available published driver information, ...”. Where IHC is not available, we are dependent on driver statuses reported by published ICGC or HMF studies. Note that many samples may be MMR-deficient through promoter hypermethylation of *MLH1* and will not thus show a genetic “driver”. Hence, the point at the end of the paragraph “should customary targeted sequencing strategies be solely used to identify driver events associated with these deficiencies, a significant proportion of cases (66%) could be missed.”

That’s because targeted panels will only “see” the genetic driver mutations in ~33% of cases (17 of the 50 predicted positive cases had reported drivers) and will miss all the MMRd/polymerase dysfunctional cases that are due to promoter hypermethylation or other causes.

This paragraph, by the way, is not a training or validation point for PRRDetect. This is simply a demonstration of the biological/clinical insight that if we stuck with targeted panels of MMR/polymerase genes, we would miss a significant proportion of the cases. Given the lack of clarity, we have adapted the manuscript to clarify the assumed context: “the assumption of PRRDetect predicted results being true”:

Line 383: “Should PRRDetect predictions be all true, and customary targeted sequencing strategies be solely used to identify driver events associated with these deficiencies, a significant proportion of cases (66%) could be missed.”

#Point 11

11. Reading through methods, signature analysis seems to have been used to pick PRRd and non PRRd samples for GEL (“samples .. were confirmed to lack driver mutations .. and displayed no evidence of signatures associated abnormalities” (lines 881-883). Were signatures (or the lack of) part of the criteria used to determine PRR proficient samples? If so, was this done only for training or for testing? If so, this could obviously bias either training and testing or both.

Please see response to **#Point 9** above. **To define the ground truth set of PRRD and PRR-proficient samples, high-confidence driver mutations (including allelic status) and/or IHC were used.**

Having a clear ground truth set is incredibly important. Thus, to be exacting and highly conservative, we ensured that there was no ambiguity at all in the ground truth set: So, we manually checked and were sure that:

- True positives really did have drivers and the associated signatures.
- True negatives really did not have drivers or any hint of any similar signatures.

It was not a criterion. It was a simply a check. We are unclear why this is an issue. This would usually be preferable over the alternative, of not checking.

#Point 12

12. From which cohorts are the numbers in lines 350-358 derived (sensitivity, AUC). Cited figure 4 panels (c,e-f) show ICGC / Hartwig results but unclear how the true labels were defined and hence how sensitivity / AUC were computed for these cohorts. Also 4c labels these cohorts as “application” but it appears that they are being used for testing. If so, then why not include them in the ROC analysis (Figure 4d) ?

- **Figure 4c** is a diagram of the workflow. It is not a result. It is simply telling the reader what samples and numbers are in the various cohorts, and which cohort is used for which purpose.
- **Figure 4d** is the validation of PRRDetect performed on 504 independent ICGC breast samples as this is the cohort with highly curated, confident PRRD labels for us to calculate an accurate AUC. **Figure 4d** has now been revised to include more samples (see **#Point 14** below).
- **Figure 4e** is not validation – it is simply a graphical representation of PRRDetect predictions contrasted to TMB, MSIsq, MMRDetect and drivers in ICGC/Hartwig. These samples are not being used for PRRDetect training/testing or validation at all (see **#Point 7**). The numbers in the original lines 350-358 relate to the ‘compare and contrast’ between the different classifiers.

RTR.v4

As this was confusing, we have made the following changes to the manuscript, starting from line 369:

“Based solely on available published driver information for PRRd status, PRRDetect could distinguish Pol-dys and mixed samples from polymerase-proficient samples with an AUC of 0.999, compared to an AUC of 0.83 if TMB (10 muts/Mb) was used. Overall, PRRDetect performed superiorly to current biomarker strategies, correctly predicting all Pol-dys, MMRd/Pol-dys samples and missing two subclonal MMRd samples. MSIsq missed six of 43 PRRDetect-predicted MMRd, two of four mixed MMRd/Pol-dys cases while displaying poor concordance for detecting pure Pol-dys cases. Unsurprisingly, PRRDetect captured all MMRDetect positive cases. However, MMRDetect failed to identify all PRRd cases as it was not designed to detect Pol-dys/mixed phenotypes and missed 7 MMRd samples. We also noted multiple instances of PRRDetect positive cases with no associated driver, which carries clinical significance: of 50 PRRDetect positive cases, 39 were MMRd (eight had an associated driver mutation), seven were Pol-dys (all had driver mutations in polymerase proofreading domains), four were predicted as mixed MMRd/Pol-dys (two had exonuclease mutations, none had MMR drivers). Should PRRDetect predictions be all true, and customary targeted sequencing strategies be solely used to identify driver events associated with these deficiencies, a significant proportion of cases (66%) could be missed.”

#Point 13

13. It is unclear which groupings refer to algorithm outputs rather than true labels (meaning those derived from manual curation, IHC, and / or driver analysis). This is additionally complicated by the fact that the authors have published a previous algorithm MMRdetect with overlapping outputs to PRRdetect. For example, in 4f and 4g the caption implies that some of the labels (MMRd, Poly-dis) are algorithm outputs, however in the previous panel MMRd and Poly-dis refer to the true labels. Also in Extended figure 10, among other places in the manuscript. It would help to use uniform and unambiguous terminology to make this distinction clear across the manuscript.

Figures 4e,f,g and Extended Data Figure 10 are all PRRDetect algorithm outputs. These were all specified in the figure legends, starting from line 553 and are copied here for you:

e, PRRDetect results of n=1,335 ICGC and Hartwig cancers, ordered from the lowest to the highest prediction probability across the x axis (left to right) for polymerase-dysfunctional samples (orange), combined mismatch-repair deficient (MMRd) samples (blue), and MMRd samples (purple). Negative samples were ordered by TMB in increasing order from left to right. Results of MSIsq, MMRDetect, cancer gene driver annotation, and cancer tissue origin are labelled at the bottom tracks. Dashed rectangle highlights the extent of false positive overcalling if using TMB > 10 muts/Mb as a cutoff.

f, Concordance of calls among TMB-H (> 10 mutations/Mb), positive exposure to SBS signatures that impart hypermutation, and PRRDetect prediction across n = 1,335 ICGC and Hartwig cancers.

g, Concordance of calls among TMB-H (> 10 mutations/Mb), positive exposure to SBS signatures that impart hypermutation, and PRRDetect prediction across n = 4,775 GEL tumors.”

However, given the query and the potential lack of clarity, we additionally labelled Figure 4 panels and Extended Data Figure 10 with “PRRDetect” on the figure panels:

To make it impossible to miss, we also incorporated main text changes to clarify PRRd predicted labels in:

Line 389: “just over a tenth of 459 cases classified as TMB-High (50/459, 10.9%) had predicted PRR dysfunction”.

Line 397: "Among the 1,371 TMB-H cases, nearly half the cohort (677, 49.4%) **was predicted** as having MMRd and/or polymerase dysfunction (Fig. 4g)."

#Point 14

14. The ICGC breast cohort used for validation (Figure 4d) does not appear to have many positive (PRRd) samples, just judging by the shape of the curves. How many Are the results in Figure 4d statistically significant? In other words, is the AUC generated by PRRdetect significantly higher than the previously published methods including MMRdetect?

Indeed, the ICGC breast cohort has only 10 PRRd samples. PRRd is rare in other cancer types but we chose to use a different, independent cohort from the training set for validation, and where we have solid, confident PRRd status of the samples.

PRRd is more common in colorectal and uterine cancers. However:

- We cannot reuse the colorectal/uterine cancers from GEL cohort because as the reviewer has argued above, we shouldn't reuse the training set for validation.
- ICGC colorectal (n=52) and uterine cohorts (n=44) are very small, because WGS has not been performed much in these tumor types, and crucially, there are no MMRd samples (n=0) based on driver information. There are 7 colorectal and 1 uterine cancers with polymerase driver mutation. That's fewer than the validation cohort that we did use.
- The colorectal and uterine cohorts of Hartwig are also not large (n=379 and n=42, respectively).
- More critically, the ground truth labels of samples in these additional cohorts are not solid (allelic status not defined, IHC not available, and risks not being informative).

However, to increase the sample size of the validation cohort, and hence, the power of PRRDetect validation, we have now added to the n=504 breast cohort, an additional n=847 GEL cancers (see revised Figure 4c below) with confirmed PRRd status for 104 positive and 743 negative samples. Critically, these samples are completely independent, and never used in PRRDetect training previously.

With more samples (also more PRRd cases), we could now see clear separation of the AUCs (revised Figure 4d above). PRRDetect does perform significantly better than MMRDetect ($P < 2.2 \times 10^{-16}$), MSIsseq ($P = 9.773 \times 10^{-5}$), and TMB (10 muts/Mb) ($P < 2.2 \times 10^{-16}$).

#Point 15

15. The previously published MMRdetect algorithm seems to very comparable outputs to PRRdetect (Figure 4e). This is not a coincidence, since PRRdetect meant to capture MMR dysfunction as well as polymerase dysfunction, as well as a mixture of the two. However, polymerase dysfunction is very rare, as extended figure 10 shows, so the number of cases this will affect is small. Hence it begs the question as to what is the incremental advance of PRRdetect, especially as a clinical tool. Does it substantially improve both MRRd and polymerase deficiency detection, or just one of the two?

First, MMRDetect was only designed to detect MMRd phenotypes. By contrast, PRRDetect detects three
 phenotypes: MMRd, polymerase dysfunction /(pol-dys), and mixed MMRd/pol-dys phenotypes. This is already an
 advantage.

Second, compared to MMRDetect, PRRDetect improves on both MMRd and Pol-dys detection.

a) In the ICGC/Hartwig cohort, MMRDetect classified 34 samples as MMRd. In fact, 2 out of 34 were mixed
 MMRd/Pol-dys and identified as such by PRRDetect. By contrast, PRRDetect classified 39 samples as pure
 MMRd. Thus, MMRDetect missed the following 7 MMRd samples:

- 1) 9fc5b5c7-3973-42b4-8710-454de0cb5b50 Skin (SBS7 dominated)
- 2) HMF003589A - CNS
- 3) HMF003436A - CNS
- 4) HMF001534A - Skin
- 5) HMF002693A - Lung
- 6) HMF000572A - Colorectal
- 7) HMF000104A - Colorectal

So, PRRDetect has increased sensitivity and specificity over MMRDetect. Yes, this is a small number, but this is a
 limited study. If this were extended to a population level incidence of cancer, the numbers would be more
 substantial in clinical practice.

b) In the validation cohort, MMRDetect missed the following 4 samples:

- 1. PD5937a (APOBEC hypermutator)
- 2. GEL-2414040-11 (SBS6)
- 3. GEL-2495139-11 (mixed)
- 4. GEL-2925291-11 (mixed)

These samples were missed by MMRDetect because MMRDetect uses the cosine similarity between the SBS profile
 of tumor and that of MMR knockouts as one of its input parameters. In addition to having the expected MMRd-
 associated SBS, these samples were either subclonal for MMRd or were also dominated by other processes, such as
 UV exposure SBS7 or APOBEC SBS2/13 hypermutation, causing the MMRDetect-calculated probability to be
 penalised, resulting in false negative calls by MMRDetect.

Additionally, MMRDetect uses only the experimentally derived MMRd-SBS signatures (*i.e.*, SBS26 and SBS44),
 precluding the possibility of identifying samples with MMRd-SBS signatures other than SBS26/44, such as SBS15
 and SBS97. PRRDetect uses SBS6, 26, 44, 15 and 97 – all SBS signatures that have been associated with MMRd, for
 MMRd prediction and indel signatures as well.

Critically, all MMRDetect-predicted MMRd samples are also captured by PRRDetect. The prediction results of all
 algorithms used, wherever possible, of all samples analysed in the manuscript, were provided in **Supplementary
 Table S7 (formerly S6)**.

Rarity is not a reason to dismiss classification. Indeed, the whole purpose of a tumor-agnostic process such as
 whole genome sequencing is to permit one assay to be read by many different tools to give a comprehensive read-
 out. Indeed, this is already in action today. Following WGS, each genome is read for all its drivers, gene fusions,
 germline mutations, somatic signatures, TMB, Myriad LOH score, HRDetect score, and PRRdetect scores as a matter
 of course.

It means that a clinician does not have to specify the assay. All assays can be run and the clinician receives a full
 genome report with all possibilities reported. Thus, while *POLE*-mutated cancers may be relatively rare, they are
 not negligible and make up about several hundred cancers per year in the UK. For the rare patient whose cancer is
 in fact, *POLE*-mutated, the precise diagnosis may make the difference to receiving the right chemotherapeutic
 option of otherwise.

RTR.v4

We would like to thank the reviewer for raising this point, however, as it has led us to make some changes to the manuscript lines 378-385, and in line 399: "only ~50% of them have an identified driver".

#Point 16

16. Specifically, how much better is PRRDetect at detecting polymerase deficiency than just the total SBS6 + SBS10 mutational burden in clinical tumor samples. While MMR is indeed hard to specifically detect with SBSs only, polymerase dysfunction detection seems more straightforward.

SBS6 is not due to polymerase dysfunction. Perhaps the reviewer was referring to using solely SBS10a (*POLE*) and SBS10d (*POLD1*) to detect polymerase dysfunctional samples. In fact, we did contrast PRRDetect's performance against all other possibilities. The results were provided in **Extended Data Table 1**, and copied below for you:

Extended Data Table 1 Performance metrics of attempted prediction models using different input parameters

No	Model	Normalization	Training ACC WM	Test Set MED.ACC	α std	λ std	Med Coef. Std	Median Multiclass AUC on Test Set	Best α WM	Best λ WM	Multiclass AUC
Sigs	Prop	0.9947	0.988	0.46	0.0005	0.551	0.9955	1	0.001	0.9975
Sigs+R	Prop	0.9947	0.99127	0.433	0.00052	0.4611	0.9961	0.05	1E-07	0.9977
Sigs+TMB	Prop	0.3748	0.377	0.2	0.02	0	0.5	0.65	0.099	0.5
Only_IND	Prop	0.9807	0.9825	0.333	0.00067	0.708	0.9844	0.1	1E-07	0.9932
Only_SBS	Prop	0.965	0.9651	0	0	0.16	0.9734	1	1E-07	0.968

Shaded row shows the model selected as the final PRRDetect classifier.

Model:

- Sigs, using both SBS and InD signatures;
- Sigs+R, using SBS, InD signatures, and (total indel count/total SNV count) ratio;
- Sigs+TMB, using SBS, InD signatures, and tumour mutation burden (TMB);
- Only_IND, using InD signatures only;
- Only_SBS, using SBS signatures only.

Current PRRDetect model (no.2 in the table above) was chosen because of its highest test set median accuracy, AUC, and stability over other models with single parameters (i.e., SBS only, InD only).

#Point 17

17. Similar for cell models - the study focuses on indels, but how much additional accuracy do indels provide on top of SBS6+10 for polymerase dysfunction detection in these models? Extended Figure 2g shows cosine similarity but does not address this specifically.

This question is very similar to #Point 16. SBS6 is not used to define polymerase dysfunction. SBS10a may be used on its own to denote polymerase epsilon dysfunction. However, like any other classifiers, with additional strands of orthogonal evidence, (e.g., another signature from another class of mutation), the ability to be more specific is higher. This is a well-known principle – multiple strands increase specificity of signal. In situations where the polymerase dysregulation is in a sample with low tumor cellularity or is in a subclone, the addition of orthogonal signatures is critical for accurate classification.

#Point 18

18. How does the new taxonomy, signature system, and classifier perform on mutants profiled in the 2021 Nature

RTR.v4

Cancer paper? Given the overlap of the previous WGS data on knockouts (including MMR knockouts), it would seem fitting to analyze this.

This analysis was performed previously and included in the manuscript already.

Please see **Extended Data Figure 8b** and **Extended Data Figure 9** where we showed that across knockouts of overlapping genes in two different human biological systems (RPE1 and iPSC), the genotype-specific experimentally derived signatures are extremely similar. This is demonstrated plainly through hierarchical clustering in **Extended Data Fig 9**, which puts all the relevant iPSC and RPE1 genotypes together, with cancer derived InD signatures.

#Point 19

19. Taking a step back, it would seem more interesting to build a classifier to diagnose repair dysfunctions more specifically than what the authors are doing here, rather than just 3 dysfunction categories, two of which seem quite rare (Pol-dys, MMRd/Pol-dys). Why can't the RPE1 data be used (in combination with previously published 2021) and cancer genomics data to build such a classifier? This would tie together the two currently disparate parts of the paper.

We agree that academically, it is terrifically interesting that there are such marked gene-specific signatures. Mismatch repair deficiency is not simply "MSI" or "a high TMB".

- We show this experimentally
- We find evidence of this in the cancer datasets and
- We made this assertion in paragraph starting line 295/428, "eight gene-specific MMRd and polymerase-dysfunction InDs. ..."

However, clinically, gene specificities are not required for treatment. All MMR gene defects are treated equally. Whether this is the best way to treat patients or not is another matter that is beyond our ability to resolve/and the scope of the manuscript. We see the same situation with *BRCA1/BRCA2*. Currently, all the customary assays simply denote tumours as *BRCA1/2*-deficient or proficient and do not require distinguishing between *BRCA1/BRCA2* mutated cancers. Yet, they are different genes with different physiological functions, producing subtly different signatures. This is a known generic problem, and not something that we can address.

However, it is an important philosophical point. Perhaps the day will come where we need to highlight to clinicians/trialists, that there are gene-specificities and perhaps this plays a role in treatment outcomes. If the reviewer is happy with this, we are happy to make the point in the discussion which we think will elevate the argument.

Line 446: "Finally, our classifier does not distinguish between the MMR-deficiency genotypes although there are clear differences between *MLH1/MSH2/MSH6* and *PMS2*, for example. Currently, there is no clinical indication to do so. However, should it become clinically important to distinguish between these genes, it shall be possible to do so."

#Point 20

20. The strand bias signals are interesting, though it seems that more could be made of this. A strand bias not only reflects a difference in mutation rates but also a pyrimidine vs. purine preference for mutations arising on that strand. In other words, while the convention is to align mutational strands to pyrimidines, a leading strand bias implies either an increased pyrimidine mutation rate on the leading strand or an increased purine mutation rate on the lagging strand. Under the biological assumption that dysfunction of POLE and POLD increases mutation rates on the leading and lagging strand, respectively, their data implies that both POLE and POLD dysfunction preferentially causes T rather than A insertions. How do the authors explain this?

When strand asymmetry is observed, there are two possible reasons (covered elsewhere extensively). There may be a bias of DNA damage or there is a bias in DNA repair.

For environmental mutagens which are known to cause DNA adducts that require the activity of transcription-coupled repair, we can confidently assert whether the DNA damage occurred on a purine or a pyrimidine base, based on **transcriptional strand asymmetries** (and not replication strand asymmetries).

RTR.v4

- For example, C>T mutations of UV light, when inspected for transcriptional strand bias, shows an excess of C>T mutations on the non-transcribed strand more than the transcribed strand. This is due to the activity of transcription-couple repair (TCR) preferentially repairing DNA damage on the transcribed strand. This also fits with the known information that UV causes covalent modifications on pyrimidines.
- By contrast, C>A mutations of tobacco smoke, when inspected for transcriptional strand bias, shows an excess of C>A mutations on the transcribed strand more than the non-transcribed strand. Because the imprint of TCR activity is the excess of mutations on the non-transcribed than the transcribed strands, we can know that the true DNA damage must be on the G (purine), and not on the C. This is in keeping with known biochemistry of guanine adduct formation caused by polycyclic aromatic hydrocarbons.

For endogenous mutagenesis caused by defective repair processes, we cannot quite make the same assertion as above. However, we have argued and shown (which we think the reviewer agrees with), that there is a bias of mainly T insertions on the leading strand for *POLE-mut* (which is known to be the major replicative polymerase of the leading strand, and the opposite is seen for *POLD1* (which preferentially replicates and proof-reads the lagging strand). Biochemical analyses have shown that B-family polymerases are particularly proficient at discriminating between correctly and incorrectly paired bases at template adenine (PMID: 21036870), *i.e.*, less accurate on replicating template thymine.

In agreement with published work (PMID: 33764464), our data suggests that the *POLE/POLD1* polymerase mutants frequently accumulate +A insertions (on the nascent strand) particularly when replicating through T-homopolymer of specific tract lengths (5-7nts). We note that we have not overtly made the point about T insertions in the manuscript. Given that the reviewer has kindly raised it, we have now included it in:

Line 119: ..., specifically T insertions at polynucleotide repeat lengths of 5-7 nts (Extended Data Fig. 3).

Line 120: This is in keeping with the hypothesized preferential activity of Pol ϵ and Pol δ in leading and lagging strand synthesis respectively^{23,24}, suggesting that *POLE/POLD1* mutants tend to accumulate 1bp A insertions on the nascent strand while replicating through 5-7 nts poly-T-tracts (PMID: 33764464, 21036870). This aligns with the proposition that polymerase ϵ and δ are more proficient at detecting incorrectly paired bases at template adenine (PMID: 21036870).

#Point 21

21. Since the study does not include immunotherapy analyses, it does not seem appropriate to mention immunotherapy in the abstract. To the reviewer, it does seem likely that the modest improvements in PRRd accuracy provided by the classifier here will improve prediction of immunotherapy outcomes.

Clinically today, the reason for identifying MMR-deficient and PRR-deficient cases is because these patients are candidates for immunotherapies. Hence, we have stated originally in the abstract:

“Furthermore, we develop a highly specific classifier of PRRd status in tumors, PRRDetect, with potential implications for immunotherapies”.

We have **not** said “...with superior performance in immunotherapies”, which would have required an analysis. The use of the word “potential” communicates the tentativeness; it implies that it is possible or likely, but has not yet been realised or confirmed.

We are happy to defer to the editor on this point but we are often encouraged to say what “impact” a manuscript has. Perhaps we could suggest this change:

“Furthermore, we develop a highly specific classifier of PRRd status in tumors, PRRDetect, ~~with~~ given potential ~~implications~~ for immunotherapies”.

#Point 22

22. Authors cite space considerations, but it seems that some of the main panels (such as 1f, 1d, 3b, 4a, 4e) could be moved to extended data or substantially condensed to show key signals. Also NG allows for up to 7 figures.

RTR.v4

759 We're happy to discuss the path forward with the editor/reviewer.

760

761 We thank you for all your feedback and comments, which have helped us clarify issues and improve the
762 manuscript.

RTR.v4

Please find below my comments on the previous review. Overall, I am supportive of this work as it improves our
understanding of indel generating processes immensely.

I only have one major point that needs addressing (other suggestions are minor). The PRRdetect performance
evaluation requires important clarifications regarding the validation set labelling before I can fully assess its utility
(see below). In a related point, I believe any statements implying *clinical* utility need to be moved from the results
to the discussion, as this is more speculative at this stage. To claim clinical utility, at a minimum PRRdetect would
have to be assessed against current clinical diagnostic standards on the same samples (for example, IHC +
microsatellite PCR), especially as TMB has recently been shown to be inadequate. Please also remove the word
clinical from the section header.

Reviewer #1:

Remarks to the Author:

The authors have provided a revised version of their study "Redefined indel taxonomy reveals insights into
mutational signatures".

As before, the study begins with mutational patterns in cellular models of post replicative repair deficiency
(PRRd) and ends with a revised 89 channel indel taxonomy from which they derive signatures and apply to pan-
cancer data, including training a new PRRdetect classifier.

**#Summary Section**

Overall, while some clarifications have been made in the text and rebuttal, there is still a surprising disconnect
between two parts of the paper - looking at cell models and cancer genome data analysis. Especially now that the
authors have clarified that the revised indel taxonomy is based on cancer genomes, as is the PRRdetect training.

The flow of the paper seems to then use the cell models mainly to show that COSMIC-83 is inadequate, after which
these unique data are sort of cast aside as the study moves to a second technical cancer genomics analysis that
might stand alone as a separate paper.

Thank you for taking the time to review our paper again.

We find it a little perplexing that the experimental analyses and the cancer analyses in our manuscript were viewed
in a dichotomous way. The principle of performing experiments and then assessing generalisability in a real-world
cancer dataset is not uncommon. This step-wise approach of presenting data has been used by others recurrently
(PMID 35140396 is a good example of one exploring indel signatures published in Nature (amongst many others)).

A dichotomous description casts a slightly negative aspersion – and here we would respectfully disagree that the cell
model data are used distinctly from the cancer data. The analyses are integrated; the cell model data are central to
the narrative and fully integrated with the cancer data:

- • Section 1 on “Diversity of indel patterns in PRRd” – here, data from cell models are the main subject matter.
 - • Section 2 on “Limitations of current indel taxonomy” – here, data from cell models are the main subject matter
and demonstrate the problems with the current taxonomy and that a change is needed. This is an experimentally-
driven exercise. It is robust and agnostic.
 - • Section 3 on “A new framework for classifying indels” – data from cell models are definitely not cast aside. The
new indel framework is proposed based on the logical argument of using flanking sequence context (amongst
other features).
 - ○ The new framework *is assessed in cancers AND*
 - ○ then in paragraphs 3-6 of the same section of the manuscript, *the experimental data are used alongside*
*cancer data to examine the technical strengths of the new classification system.*

Thus, the experimental data are fully and comprehensively utilised and integrated.

RTR.v4

I agree with the authors. The cell line data are used appropriately in demonstrating how the current signatures don't capture the variation observed in the isogenic models and the data are also used to establish that the new signature framework captures this variation. No further work is required here.

Importantly, the final classifier PRRDetect seems to offer incremental classification benefit over their previous MRRdetect algorithm and SBS signatures, especially given the rarity of polymerase dysfunction and the ease of classifying it with SBSs. It would seem like their unique cell model dataset and new indel taxonomy system would be able to more precisely diagnose the repair dysfunction (for example to a specific protein defect). Such a demonstration would elevate it much more clearly above the state of the art (including using the burden of SBS6 and 10 to diagnose polymerase deficiency).

Here, we humbly disagree with the notion that "incremental benefit" is expected because of clinical rarity of polymerase dysfunction. Rarity is not an adequate argument for dismissing a classification process.

- First, rarity in one tumor-type may not translate to rarity in other tumor-types (e.g., Mismatch repair deficiency).
- Second, 1% of a common disease like cancer, translates to many thousands of people with cancer.
- Third, most genetic diseases are extremely rare, but this does not warrant precluding them from scientific investigation or accurate classification. To date, there are no models to classify polymerase dysfunction or combined polymerase dysfunction and MMR deficiency in a single assay. PRRDetect is the only classifier that does so, and performs better than existing biomarker (i.e., TMB) and algorithms in detecting samples with MSI.

We do agree that gene specificities are a very interesting feature of mutational signatures and we acknowledge this in the discussion. However, to date, there is no clinical indication to distinguish between gene-specific signatures in MMR deficiency (MMRd). All MMRd samples regardless of their originating genotype, receive the same therapy. Also see answer to Point 19.

I agree with the authors. Although further work is required to clarify the utility of PRRdetect (see below).

Specific critiques

- The indel taxonomy still seems arbitrary, and not really justified by biological principles or data (including the RPE1 data), especially to a general genetics reader. While it is clear from their paper that COSMIC-83 is lacking, it is not convincingly shown that the proposed taxonomy is the natural successor.

If we could discuss the logic here: By extension of the reviewer's argument, the COSMIC-83 indel classification would be entirely arbitrary and not justified by biological principles/data, because:

- It is not constructed through examination of any underlying data, as plainly demonstrated in panel **a** of **Extended Data Figure 3 (Former Extended Data Figure 4)**.
- Because of limited power, the channels were arbitrarily capped at homopolymers of 6bp in lengths.

We wonder why that is considered reasonable. Yet our position, which is a systematic experimental demonstration of the inadequacies of the current indel classification system, with logical suggestions of improving it by extending the homopolymer run to 9bp, introducing flanking sequence context (a system used for substitutions), justifying it in the experimental data and showing that it works in cancer, is not considered adequate and is dismissed rather easily.

We are suggesting our taxonomy as an alternative methodology for indel classification, which could reveal more biological granularity and is able to disambiguate PRRd signatures and/or uncover new signatures, as demonstrated by our RPE1 experimental data and cancer data. We have not, at any point, pitched the new classification system as the "natural successor" to COSMIC-83. Further, we acknowledge in line 454 that **"Nevertheless, optimal classification remains an active research area, and alternative schema could unveil additional mutational processes in the future"**.

Our proposed classification expands on the current COSMIC-83 taxonomy, with key additions, considerations and modifications as follows:

- 07
08 1/ Incorporating the 5' and 3' flanking bases for 1bp indels.
09 2/ Extending the cap on homopolymer length from 6 to 9bp, to allow for better discriminatory power as we
observed that there are more indel signals at longer homopolymers (**Extended Data Figure 3a,e**), which may
offer the resolution required to tease out distinct biological phenotypes. Note that the frequency of
homopolymers in the genome decreases with increasing homopolymer length (see **Point 3** figure). There's
a trade-off between being exhaustive to account for all the homopolymer/indel lengths and conserving
power for signature extraction.
3/ Though longer repeats (up to 9bp) are captured in our classification, to improve the indel *signal-to-channel*
ratio, we grouped the homopolymer lengths into short (0-4bp), medium (6-7bp), long (8-9bp) based on how
they are distributed (**Figure 2a, Supplementary Note 1, Extended Data Figure 3**), to prevent over-sparsity.
This was an informed decision based on the distribution of indels/homopolymer tracts in the reference
genome. This was not arbitrary.
4/ We aggregated channels where indel signals were low across the pan-cancer cohort data because they
would contribute limited value (*i.e.*, no point in having empty channels) while costing statistical power for
analyses.

These are thoughtful deliberations based on the distribution of mutations (actual data is in **Extended Figure 3**) across
multiple cohorts involving most human cancer types and are not arbitrary decisions. Our approach is data-driven and
unbiased.

From a large body of published work, we know that indel formation is commonly driven by the underlying sequence
motifs. While many in the community have been introduced to COSMIC-83 for signature analysis, it is not as widely
adopted as the substitution channels because it has been problematic, and simply because it was the first, and has
been introduced by the Sanger/COSMIC team, does not mean that it should be dogma. Many have struggled and
found it challenging to interpret the COSMIC ID signature results. This is because COSMIC-ID lacks detailed sequence
context information, resulting in peculiar signature assignments, leading to uncertainties in the interpretation of
results (PMID: 38361034). This is not only apparent in our RPE1 dataset presented here, but also has been shown by
others (PMID: 35320711, 35140396) that the COSMIC-83 is not able to resolve many biological phenotypes.

It is with these biological observations as motivations and also based on additional insights gleaned from many
mechanistic studies (PMID: 30982602, 32661091, 35320711, 35140396) showing that a lot of mutational processes
and mutagens show a predilection for specific sequence contexts to form adducts/damage and cause indels (*e.g.* UV
at TT, etc), that we expanded on the current COSMIC-83 framework to include the considerations outlined above.

Therefore, we maintain that our classification system is not arbitrary. It is informed by biological observations, and
guided by the understanding of the underlying data distribution. It is also noteworthy that we, in addition to
capturing all previous COSMIC-83 signatures with confirmed aetiologies, also identified multiple new signatures in
the 7 GEL cancer types, while providing additional insights/dimensions to guide signature interpretation.

We stand by the validity, robustness, and biological value of our approach.

I agree with the authors here, although I note that the language used to describe the COSMIC-83 results could be
perceived as disparaging. I suggest the authors alter the text in the manuscript to adopt a tone of discovery rather
than competition between signature taxonomies. The authors also need to make it more explicit that the new
taxonomy is not a replacement for the old taxonomy. Rather, it is one that better captures differences in the
mutation patterns between types of post replicative repair dysfunction.

Some suggestions where the text could be reworded:

Line 29: ...prevailing indel classification framework for deriving mutational signatures falls short...

Line 138: We thus questioned the sufficiency of COSMIC-83 taxonomy....

Line 145: Surprisingly, indel signatures of Δ MSH2 and Δ MLH1 showed no similarity to the purported MMRd-
associated ID7

30 Line 152: Yet, it is widely known

31 Line 415:disregards surrounding sequence context....

RTR.v4

etc

#Point 1

1. One way to present the new taxonomy is as a modification / improvement to COSMIC-83. There seem to be two key differences with COSMIC: First, longer repeat lengths (up to 9) are captured. Second, the base context to the 5' and 3' of a mononucleotide repeat is considered. Is this true? Can these two choices be concisely justified by main text data analysis or cited biological first principles?

These are broadly correct but there is more complexity (which the reviewer has mentioned in #Point 3).

We present below an alluvial plot mapping the COSMIC-83 features to our InD-89 system, which we made it into the new Extended Figure 4 (and took out the former Extended Data Figure on strand bias plot, and instead presented strand bias results as the new Supplementary Table 3 to keep to the figure limit). We kindly seek your guidance on this.

As can be seen above, there is no direct 1-to-1 mapping of COSMIC-83 channels to InD-89 channels. Predominantly, the InD-89 taxonomy expands upon the T indels (where most indel signals are concentrated) and condenses the longer indels (>= 2bp) (where the signals are scant). Detailed mapping of the channels between the two classification systems and some examples were provided in Supplementary Table S4 (former S3).

We have inserted the following statement to fall in line with the PCAWG classification:

Line 193: Overall, compared to the COSMIC-83, the 89-channel taxonomy expands upon 1 bp T indels (where most of the signals were) into a larger array of channels, and condenses longer indels and/or genome motifs that are infrequent in the genome (where signals were scant or non-existent) into fewer indel subcategories.

There is, however, far more complexity that makes our system more reproducible and robust, which the reviewer has highlighted in #Point 3 below.

This is a good addition to the manuscript.

#Point 2

2. Along these lines there is data in the supplement, including histograms of repeat lengths which actually show a substantial fraction of indels with repeat lengths of 9 or greater (for example 1bp T indel). Does it really make sense

RTR.v4

to collapse them? In other words, why draw the line at 9+? If the authors are looking to rethink the indel
taxonomy, are they being bold enough?

This is not correct. The histogram shows indels up to repeat lengths of 9, and *not 9 or greater*. See below for the
zoomed-in example of all possible 1bp T deletions. The same applies to all homopolymer, polynucleotide repeats
(*i.e.*, repeat # <= 9)

1 bp T indel channels (81 subcategories)

*It is not possible using today's genomic data pipelines of any institution, to confidently go beyond repeat lengths of >*
*9 because of the error rates in mutation-calling algorithms. Nearly all commonly-used genomic pipelines (Broad GATK,*
*Dragen, Strelka, Platypus, Pindel, etc) have a post-hoc filter capped at repeat lengths < 10.*

The contribution of indels at very long homopolymer tract length of >9 to indel signatures could be limited given
that there is a much lower prevalence across the genome (see image above). Having said that, we agree that indels
at longer homopolymers may be important and should be investigated at a later date when indel data obtained
through newer lower-error/long read technologies become available, in quality and quantity. Thus, we have added
the following statement:

**Lines 432: The possibility of adapting the taxonomy in the future, to include features that are currently not**
**explorable due to the limitations incurred by technological error rates, could also be revealing.**

I can see where the reviewer is coming from here - the figures in the supplementary note are misleading and
suggest +9. Please fix these. The author response here highlights an important technical limitation, and it would be
good to see this mentioned explicitly (with the above text or similar), perhaps in the supplementary note.

**#Point 3**

3. Are there any other important changes vs. COSMIC-83? If so, what motivates and justifies them? The remaining
differences seem minor: These include collapsing of categories - for example the authors group all length 2 to 5

RTR.v4

microhomology deletions into a single category, while COSMIC considers them separately. Complex indels are given their own channel, and ignored in COSMIC-83. There is also a new approach to indel segmentation, which perhaps may help reproducibility across indels called using different conventions. It is not clear whether COSMIC-83 already does this kind of segmentation.

Yes, the reviewer has highlighted a key difference and is correct in saying that our approach in classifying indels through a “segmentation process” does result in a more unified, reproducible, and standardised classification. Prior to our approach, because a binary threshold-based classification schemes was used (*i.e.*, see “repeat-mediated” vs “microhomology-mediated” deletions in #Point 3 Figure, for example), there were ambiguous cases that arose that could be classified in either category. This has never been addressed previously. To deal with this ambiguity, our “segmentation” strategy explicitly identifies the smallest prefix sequence motif that displays a maximal repetitive relationship with the 3' sequence context and within the indel itself. This provides a standardized approach for relating indel sequence to the 3' sequence context. COSMIC-83 does not do that.

Examples:

Indels	InD-89	COSMIC-83
C AAATAAAATA AAATACCTAGGAAT	Del(3,):U(3):R(3,9)	5:Del:M:5
G CCC CTCAGGATGATTTCCCTCCCTGG	Del(2,8):U(1,2):R(2,4)	3:Del:M:1

In the first example, AAATAAAATA is the deletion motif, AAATA is the smallest prefix sequence motif, the unit is 5 bp, and repeated 3 times. In COSMIC-83, it is classified as long deletions (>5bp) with 5bp microhomology. However, using our strategy, it is interpreted as a loss of 2 of the 3 AAATA pentanucleotide repeats.

In the second example, CCC is the deletion motif, C is the smallest prefix sequence motif, the unit is 1 bp, and repeated 4 times. In COSMIC-83, it is classified as 3bp deletion with 1bp microhomology. However, using our strategy, it is interpreted as a loss of 3 of the 4 x C mononucleotides.

Please also see answer to #Point 1. Details of segmentation and motivations/justifications were previously described in **Supplementary Note 1**. However, we agree that we have not adequately highlighted key differences between the classification systems and have included this in the abstract (Lines 32-35):

“we developed an alternative classification system that considers the 5' and 3' sequences flanking indels and includes informative motifs (*e.g.*, longer homopolymeric tracts) while removing uninformative channels. Our new cataloguing system ensures that every indel can be unambiguously classified into one of 89 indel subcategories and is capable of disambiguating gene-specific experimentally-generated signatures.”

We have also added text in the final paragraph of the Introduction to ensure that the distinction between our methodology and that of COSMIC-83 is clearly articulated (Lines 72-77):

“Additionally, motifs known to increase mutational vulnerability that are conflated by the current system (*i.e.*, homopolymer tracts >6nt), and genome-wide prevalence of such motifs were factored into our proposition. Our approach unambiguously classifies each indel into a specific subcategory.”

Thank you for raising this.

This is a good addition.

#Point 4

4. The average reader should be able to very readily grasp important features of the new taxonomy and key differences with COSMIC 83. The schematic in Figure 2a seems important for this and also to explain the channel bar plots. However, as it stands, it seems overly complicated and main differences with COSMIC-83 are not apparent. Perhaps it could be simplified, and the full version can be moved to extended data.

RTR.v4

We wonder whether moving **Figure 2a** upper panel to the Supplement would make interpretation of the rest of figures challenging. Readers would have to flip back and forth to understand the full complement of the profiles, *e.g.*, in 2d. As **Figure 2a** serves as the basis for the signature plots in **Figure 2d** and **Figure 3**, we believe that having it as a main figure panel is essential.

One option is to contrast the two classification systems to highlight differences (see the alluvial plot shown in **Point 4**). Detailed mapping of the channels between the two classification systems and indel examples were already provided in **Supplementary Table S4**. It is quite an illustrative plot. The alluvial plot contrasting the two classification systems is now provided and referenced as **Extended Data Figure 4**.

This is a good addition.

#Point 5

5. How are insertions with microhomology classified? Namely, insertions aligning with a spacer to flanking sequence. It does not seem like they have a category here.

Insertions with microhomology are a rare feature (single digits if at all), even in HR-deficient samples, where they are alleged to be seen more frequently. *Critically, they cannot be unambiguously mapped to a single category.*

That's because templated insertions are:

- often complex (*i.e.*, they are accompanied by deletion at the junction). Majority of them would have been overlooked or mis-annotated using standard indel variant callers (PMID: 26657142).
- could also be templated from inverted repeats (both left and right flanks) and not be immediately at the junction of the indel, and unaccompanied by deletions.

Although we do not designate a category specifically for these complex events, they were segmented and classified using our unified framework into various non-overlapping "insertion" and "complex" categories. Due to the complexity of these events, they are best characterised by specialised algorithms (PMID: 32234782, 26657142).

This is another important technical limitation which should be explicitly mentioned in the manuscript or supplement (with the above text or similar).

#Point 6

6. Since the taxonomy is apparently independent of the RPE1 experiments, the authors may actually want to consider presenting the cancer genome analyses first and ending with the RPE1 data. In addition to more solid motivation for the indel taxonomy, this would require restructuring of the paper, but maybe provide an opportunity to close with some biological insights.

The taxonomy *is not independent* of the RPE1 experiments. The RPE1 experiments culminated in the evidence that shows that the current COSMIC-83 is inadequate. Once the taxonomy was designed, it was also tested on the RPE1 data to show the remarkable improvement in discerning the gene-specificity that is present in the data (**Figure 2b-d**, **Extended Data Figure 5**).

We respectfully disagree with this suggestion. Furthermore, first, it would not make sense to start with an adaptation of the COSMIC-83 system for no reason at all, and then use the experiments to back up our argument. Second, it might be seen as trying to "appear clever" *if we started by philosophising a change*. An alternative reviewer could easily ask in this scenario, "why did they think it was necessary to adapt it in the first place?".

Thus, we are concerned that restructuring would, in fact, be detrimental to the manuscript and is not the truth of how the thought process went either.

I agree with the authors that the current paper structure is suitable.

#Point 7

- There still appear to be some circularities in the classifier training and testing, including a concern about data leakage.

7. The new 89-channel system is derived from the entirety of pan-cancer data (18522 samples) which seems also to include samples used to test PRRdetect, including ICGC, TCGA, and HMF. Since these test cohorts (ICGC, TCGA, HMF) were used for feature engineering (Extended Fig 4, supplementary note) they are not independent. The feature engineering and training of signatures and classifiers should be done on a single cohort, for example GEL, leaving the other datasets for testing, including ROC / AUC analysis and calculations of sensitivity and specificity.

Note that the former **Extended Fig 4** is now revised as the new **Extended Data Figure 3**.

The reviewer has conflated several distinct pieces of work together in the argument above. There are three distinct pieces of work:

- 1) The demonstration of validity of the 89-channel system
- 2) Indel signature extraction from seven cancer types
- 3) The PRRDetect classifier development

These are different analyses. Historically, a paper would be done on (1) and (2). For example, the substitution signatures (Cell 2012a, Nature 2013) are an example of analyses similar to items (1) and (2). Later, some type of classifier might be created (often using past publicly-available data because that's what is available and we need something as a "gold standard"). For example, HRDetect (Nature Medicine 2017) – this would be similar to item (3). These are different analyses resulting in different papers. In this manuscript, we have argued for an alternative classification taxonomy (1) and shown how it would be used to find signatures (2). Item (3) is just like HRDetect, it is a separate exercise, and could easily have been a separate paper. These are simply different analyses.

1) The 89-channel system

The entirety of the pan-cancer data of 18,822 samples was *simply used to show the indel distribution across all categories from available cancer types across all three cohorts (GEL, ICGC and Hartwig)*. Because the same reviewer previously raised concern that our proposed framework was potentially "biased towards PRRD phenotype", we presented the plots in the now **Extended Data Figure 3** to show that:

A/ indel distributions across InD channels are preserved regardless of cohorts or cancer types, and are not biased towards PRRd phenotypes or skewed towards hypermutator samples (**Extended Data Figure 3e**).

B/ Additionally, the new indel classification channels have data in all channels across all cancer types, in contrast to the sparse profiles with COSMIC-83 (**Extended Data Figure 3a-d**).

§ ratio, and importantly, shows that our final channel set does not bias towards hypermutator phenotypes (and hence PRRd samples), and reinforces that our approach is data-driven.

2) Indel signature extraction

In the main text (starting from line 251), we clearly stated that indel signature extraction *was performed on the 7 selected cancer types within the GEL cohort*. An indel signature extraction has only been performed on 4,775 GEL samples, not on the whole cohort of 18,822 samples. This is unrelated to **Extended Data Figure 3**.

3) PRRDetect

In this also entirely separate section of the manuscript, we have developed PRRDetect as a multinomial classifier for polymerase dysfunction and MMRd.

TRAINING of PRRDetect

It was trained on **571 GEL** samples (a subset of the 4,775 GEL cancers), where

- i. 291 were confirmed to have driver mutations/ IHC staining.
- ii. 280 were negative set.
- iii. The training cohort of 571 went through a ~ 7:3 split for training.

These samples have confident PRRd labels. Ground truth is solid for all samples.

RTR.v4

VALIDATION of PRRDetect

PRRDetect was validated on an *independent cohort of 504 ICGC breast samples for which PRRd labels were available* (Line 360). This cohort was used specifically for validation because we are confident of the PRRd label of all samples, informed by IHC staining and careful curation of all drivers. We know that we are not missing any true PRRd samples. *These samples have confident labels. Ground truth is solid for all samples.*

Also note that during this revision of the manuscript, we have expanded on our validation cohort to include additional samples (not used for PRRDetect training) for statistical power. See #Point 14 for details.

GENERAL APPLICATION

PRRDetect was then applied on ICGC and Hartwig samples (starting line 367, $n=1,335$) to make illustrative descriptions of generalisability. For these samples, the ground truth status is not (completely) known. We only have driver information that has been published by others. *These samples do not have confident labels. Ground truth is not so solid; hence we have not used them for validation.*

We showed, however, the concordance/discordance from different prediction algorithms, and whether predicted PRRd samples have annotated driver mutations or not (**Figure 4e**). Manuscript lines 359-385. Please note, this is not the same thing as testing or training.

Additionally, all samples used (whether in training or validation, whether positive or negative, with driver and/or IHC) are all annotated in Supplementary Table S7 (former S6).

I am generally happy with the authors response here although I do believe the presentation of the GENERAL APPLICATION in the manuscript is misleading. Reporting AUC in this section implies there is some type of gold-standard labelling. While the authors acknowledge that this labelling is based solely on mutation status, I don't think they should be reporting AUC using these data alone as there will be many PRRd cases without driver mutations (as acknowledged by the authors later in the manuscript). AUC indicates some type of classification-based performance evaluation which the data doesn't really support. Rather, a more appropriate use of the data would be to report on prevalence of positive cases, concordance with mutation status, and overlap with other methods.

#Point 8

8. Related to above, it is not clear whether new ICGC breast analysis was done on samples included in the ICGC analyses shown in Extended Fig. 4.

The former **Extended Fig 4** is now revised as the new **Extended Data Figure 3**.

No. The ICGC breast samples used for PRRDetect **testing/validation** were not used in any analyses in **Extended Data Figure 3**. In fact, **Extended Data Figure 3** has nothing to do with PRRDetect. It is simply showing what the distribution of indels are across all cohorts, according to the COSMIC-83 and the new 89-channel system. See also #Point 1 – "1) The 89-channel system".

Extended Data Figure 3 is not related to PRRDetect at all. It does not bias the PRRDetect analysis because it not related to the PRRDetect analysis.

This is clear from the authors. However, given the reviewer's mix-up here, perhaps it is worth making these distinctions extra clear in the text?

#Point 9

9. It is still unclear how gold-standard labels are derived - namely what are the criteria for labeling a sample as PRRd or PRR proficient. Methods currently refer to "manual curation" for GEL samples and for the remaining samples, "true labels were established based on IHC of four MMR proteins and driver mutations". The authors should offer much more detail about these criteria and how frequently each was applied (in how many samples IHC vs. driver mutations vs. combinations was used to determine the label, including whether / how allelic status of driver mutations was assessed).

RTR.v4

Regarding the term “manual curation”

Please note the term “manual curation” for identifying drivers is not a negative thing. This means that not only was
a driver identified through bioinformatic methods, we will have also:

- - checked that the other allele was lost if it is a tumor suppressor and
- - we physically inspected the variant using IGV.

This, in our view, is being thorough and doing our due diligence. It is not and should not be viewed in a negative
way.

In any case, we have removed the words “manual curation” from paragraph starting from line 349.

True labels for each sample

Note that a full, comprehensive list of every sample, whether they were used for training or validation, whether they
had IHC or drivers is fully available in **Supplementary Table S7 (former S6)**. Note that:

- • A mutation is only considered a driver in any of the MMR repair proteins (*MLH1*, *MSH2*, *MSH6*, *PMS2*) if it
has also lost the wild-type parental allele and is thus biallelically null. Mutations in *POLE/POLD1* are only
annotated as drivers if they are previously reported mutations in functionally relevant domains. This is what
we mean by “curated” drivers - we have gone the extra mile to ensure that these are not simply passenger
mutations that have hit these genes by chance.
- • Samples were labelled with an IHC staining result if one was available. Not all samples have had IHC.

Thus, a positive PRRd label is assigned to a sample if one of the two criteria (driver or an abnormal IHC) are
met.

That we have then additionally checked to ensure that these gold-standard labels are unambiguous, must surely be
a strength, not a weakness. Samples that do not have the right gold-standard label and are not in the right category
would be detrimental to training an algorithm. Hence, we also double checked that samples in the training cohort of
571 samples were unambiguously PRR-deficient or PRR-proficient by ensuring that the relevant signatures matched
the labels, which they did.

I appreciate the authors manual curation efforts, but I agree more detail is needed regarding how the labels were
assigned to each sample. It is not clear what evidence was used to create the labels in Supplementary Table S7. If
IHC alone was used, or IHC plus mutation, mutation alone, signatures etc.

One thing I find unusual is the use of the signatures themselves to confirm negative cases. What happens if a
sample is negative for drivers and IHC but positive for PRR related signatures? Is it simply removed from the
training set? If so, this is a biased training set and this should be more explicitly stated. I would have thought MSI
could be used to assist with labelling as this is another source of information that is typically used in a clinical
setting. Nevertheless, for the training cohort, the use of signatures, although biased, is ok for creating a cleaner
training dataset. However, can the authors confirm that signatures were not used *at all* in the process of labelling
the GEL hold-out dataset (this point is also covered below)?

**#Point 10**

10. Specifically, how were true labels derived for samples / cohorts (like ICGC / HMF) where presumably IHC labels
are not available. (for example, Figure 4e). I presume it's using driver status. But then the authors refer to the false
negative rate of using strictly driver events (66%, line 365).

The reviewer was correct in saying that IHC labels were not available for ICGC/Hartwig cohorts. Hence, in line 369,
we stated that “**Based solely on available published driver information, ...**”. Where IHC is not available, we are
dependent on driver statuses reported by published ICGC or HMF studies. Note that many samples may be MMR-
deficient through promoter hypermethylation of *MLH1* and will not thus show a genetic “driver”. Hence, the point
at the end of the paragraph “**should customary targeted sequencing strategies be solely used to identify driver**
**events associated with these deficiencies, a significant proportion of cases (66%) could be missed.**”

RTR.v4

That's because targeted panels will only "see" the genetic driver mutations in ~33% of cases (17 of the 50 predicted positive cases had reported drivers) and will miss all the MMRd/polymerase dysfunctional cases that are due to promoter hypermethylation or other causes.

This paragraph, by the way, is not a training or validation point for PRRDetect. This is simply a demonstration of the biological/clinical insight that if we stuck with targeted panels of MMR/polymerase genes, we would miss a significant proportion of the cases. Given the lack of clarity, we have adapted the manuscript to clarify the assumed context: "the assumption of PRRDetect predicted results being true":

Line 383: "Should PRRDetect predictions be all true, and customary targeted sequencing strategies be solely used to identify driver events associated with these deficiencies, a significant proportion of cases (66%) could be missed."

The change on line 365 addresses the reviewer concerns. However, I am not sure I agree with the statement regarding gene panels. Gene panels offer more than just driver mutations - it is possible to determine MSI status and TMB, therefore making a statement only about drivers seems artificial.

#Point 11

11. Reading through methods, signature analysis seems to have been used to pick PRRd and non PRRd samples for GEL ("samples .. were confirmed to lack driver mutations .. and displayed no evidence of signatures associated abnormalities" (lines 881-883). Were signatures (or the lack of) part of the criteria used to determine PRR proficient samples? If so, was this done only for training or for testing? If so, this could obviously bias either training and testing or both.

Please see response to **#Point 9** above. **To define the ground truth set of PRRD and PRR-proficient samples, high-confidence driver mutations (including allelic status) and/or IHC were used.**

Having a clear ground truth set is incredibly important. Thus, to be exacting and highly conservative, we ensured that there was no ambiguity at all in the ground truth set: So, we manually checked and were sure that:

- True positives really did have drivers and the associated signatures.
- True negatives really did not have drivers or any hint of any similar signatures.

It was not a criterion. It was a simply a check. We are unclear why this is an issue. This would usually be preferable over the alternative, of not checking.

I share some of the reviewers concerns here given the lack of clarity over precisely how labels were assigned (see comment on point 9 above). For the validation set, the signatures should not be used *at all* during the labelling process as this would create a biased dataset. Any removal of mutation negative cases with positive signature levels will artificially inflate performance.

#Point 12

12. From which cohorts are the numbers in lines 350-358 derived (sensitivity, AUC). Cited figure 4 panels (c,e-f) show ICGC / Hartwig results but unclear how the true labels were defined and hence how sensitivity / AUC were computed for these cohorts. Also 4c labels these cohorts as "application" but it appears that they are being used for testing. If so, then why not include them in the ROC analysis (Figure 4d) ?

- **Figure 4c** is a diagram of the workflow. It is not a result. It is simply telling the reader what samples and numbers are in the various cohorts, and which cohort is used for which purpose.
- **Figure 4d** is the validation of PRRDetect performed on 504 independent ICGC breast samples as this is the cohort with highly curated, confident PRRD labels for us to calculate an accurate AUC. **Figure 4d** has now been revised to include more samples (see **#Point 14** below).
- **Figure 4e** is not validation – it is simply a graphical representation of PRRDetect predictions contrasted to TMB, MSIsq, MMRDetect and drivers in ICGC/Hartwig. These samples are not being used for PRRDetect training/testing or validation at all (see **#Point 7**). The numbers in the original lines 350-358 relate to the 'compare and contrast' between the different classifiers.

As this was confusing, we have made the following changes to the manuscript, starting from line 369:

“Based solely on available published driver information for PRRd status, PRRDetect could distinguish Pol-dys and mixed samples from polymerase-proficient samples with an AUC of 0.999, compared to an AUC of 0.83 if TMB (10 muts/Mb) was used. Overall, PRRDetect performed superiorly to current biomarker strategies, correctly predicting all Pol-dys, MMRd/Pol-dys samples and missing two subclonal MMRd samples. MSIsq missed six of 43 PRRDetect-predicted MMRd, two of four mixed MMRd/Pol-dys cases while displaying poor concordance for detecting pure Pol-dys cases. Unsurprisingly, PRRDetect captured all MMRDetect positive cases. However, MMRDetect failed to identify all PRRd cases as it was not designed to detect Pol-dys/mixed phenotypes and missed 7 MMRd samples. We also noted multiple instances of PRRDetect positive cases with no associated driver, which carries clinical significance: of 50 PRRDetect positive cases, 39 were MMRd (eight had an associated driver mutation), seven were Pol-dys (all had driver mutations in polymerase proofreading domains), four were predicted as mixed MMRd/Pol-dys (two had exonuclease mutations, none had MMR drivers). Should PRRDetect predictions be all true, and customary targeted sequencing strategies be solely used to identify driver events associated with these deficiencies, a significant proportion of cases (66%) could be missed.”

See response to point 7 regarding my suggestion against reporting on AUC in the non-validation cohort.

#Point 13

13. It is unclear which groupings refer to algorithm outputs rather than true labels (meaning those derived from manual curation, IHC, and / or driver analysis). This is additionally complicated by the fact that the authors have published a previous algorithm MMRdetect with overlapping outputs to PRRdetect. For example, in 4f and 4g the caption implies that some of the labels (MMRd, Poly-dis) are algorithm outputs, however in the previous panel MMRd and Poly-dis refer to the true labels. Also in Extended figure 10, among other places in the manuscript. It would help to use uniform and unambiguous terminology to make this distinction clear across the manuscript.

Figures 4e,f,g and Extended Data Figure 10 are all PRRDetect algorithm outputs. These were all specified in the figure legends, starting from line 553 and are copied here for you:

“e, **PRRDetect results** of $n=1,335$ ICGC and Hartwig cancers, ordered from the lowest to the highest prediction probability across the x axis (left to right) for polymerase-dysfunctional samples (orange), combined mismatch-repair deficient (MMRd) samples (blue), and MMRd samples (purple). Negative samples were ordered by TMB in increasing order from left to right. Results of MSIsq, MMRDetect, cancer gene driver annotation, and cancer tissue origin are labelled at the bottom tracks. Dashed rectangle highlights the extent of false positive overcalling if using TMB > 10 muts/Mb as a cutoff.

f, Concordance of calls among TMB-H (> 10 mutations/Mb), positive exposure to SBS signatures that impart hypermutation, and **PRRDetect prediction** across $n = 1,335$ ICGC and Hartwig cancers.

g, Concordance of calls among TMB-H (> 10 mutations/Mb), positive exposure to SBS signatures that impart hypermutation, and **PRRDetect prediction** across $n = 4,775$ GEL tumors.”

However, given the query and the potential lack of clarity, we additionally labelled Figure 4 panels and Extended Data Figure 10 with “PRRDetect” on the figure panels:

To make it impossible to miss, we also incorporated main text changes to clarify PRRd predicted labels in:

Line 389: “just over a tenth of 459 cases classified as TMB-High (50/459, 10.9%) had **predicted** PRR dysfunction”.
 Line 397: “Among the 1,371 TMB-H cases, nearly half the cohort (677, 49.4%) **was predicted** as having MMRd and/or polymerase dysfunction (Fig. 4g).”

These changes are good and help with clarity, although the authors should acknowledge other relevant literature showing the shortcomings of TMB (e.g. <https://pubmed.ncbi.nlm.nih.gov/33736924/> among others).

#Point 14

14. The ICGC breast cohort used for validation (Figure 4d) does not appear to have many positive (PRRd) samples, just judging by the shape of the curves. How many Are the results in Figure 4d statistically significant? In other words, is the AUC generated by PRRdetect significantly higher than the previously published methods including MMRdetect?

Indeed, the ICGC breast cohort has only 10 PRRd samples. PRRd is rare in other cancer types but we chose to use a different, independent cohort from the training set for validation, and where we have solid, confident PRRd status of the samples.

PRRd is more common in colorectal and uterine cancers. However:

- We cannot reuse the colorectal/uterine cancers from GEL cohort because as the reviewer has argued above, we shouldn't reuse the training set for validation.
- ICGC colorectal ($n=52$) and uterine cohorts ($n=44$) are very small, because WGS has not been performed much in these tumor types, and crucially, there are no MMRd samples ($n=0$) based on driver information. There are 7 colorectal and 1 uterine cancers with polymerase driver mutation. That's fewer than the validation cohort that we did use.
- The colorectal and uterine cohorts of Hartwig are also not large ($n=379$ and $n=42$, respectively).
- More critically, the ground truth labels of samples in these additional cohorts are not solid (allelic status not defined, IHC not available, and risks not being informative).

However, to increase the sample size of the validation cohort, and hence, the power of PRRDetect validation, we have now added to the $n=504$ breast cohort, an additional $n=847$ GEL cancers (see revised **Figure 4c** below) with confirmed PRRd status for 104 positive and 743 negative samples. Critically, these samples are completely independent, and never used in PRRDetect training previously.

With more samples (also more PRRd cases), we could now see clear separation of the AUCs (revised **Figure 4d** above). PRRDetect does perform significantly better than MMRDetect ($P < 2.2e-16$), MSIsseq ($P = 9.773e-05$), and TMB (10 muts/Mb) ($P < 2.2e-16$).

RTR.v4

The addition of the GEL samples here strengthens the results considerably (pending clarification of the labelling used, outlined in comments above). To further enhance the presentation of the results, I suggest the authors update the above schematic to include the number of cases that are PRRd. I also recommend the authors present an optimal point on the AUC (Youden's index or similar) and mention this in the text so the trade-off between capturing PRRd samples and making (potentially) false predictions can be easily determined.

#Point 15

15. The previously published MMRdetect algorithm seems to very comparable outputs to PRRdetect (Figure 4e). This is not a coincidence, since PRRdetect meant to capture MMR dysfunction as well as polymerase dysfunction, as well as a mixture of the two. However, polymerase dysfunction is very rare, as extended figure 10 shows, so the number of cases this will affect is small. Hence it begs the question as to what is the incremental advance of PRRdetect, especially as a clinical tool. Does it substantially improve both MMRd and polymerase deficiency detection, or just one of the two?

First, MMRDetect was only designed to detect MMRd phenotypes. By contrast, PRRDetect detects three phenotypes: MMRd, polymerase dysfunction / (pol-dys), and mixed MMRd/pol-dys phenotypes. This is already an advantage.

Second, compared to MMRDetect, PRRDetect improves on both MMRd and Pol-dys detection.

a) In the ICGC/Hartwig cohort, MMRDetect classified 34 samples as MMRd. In fact, 2 out of 34 were mixed MMRd/Pol-dys and identified as such by PRRDetect. By contrast, PRRDetect classified 39 samples as pure MMRd. Thus, MMRDetect missed the following 7 MMRd samples:

- 1) 9fc5b5c7-3973-42b4-8710-454de0cb5b50 Skin (SBS7 dominated)
- 2) HMF003589A - CNS
- 3) HMF003436A - CNS
- 4) HMF001534A - Skin
- 5) HMF002693A - Lung
- 6) HMF000572A - Colorectal
- 7) HMF000104A - Colorectal

So, PRRDetect has increased sensitivity and specificity over MMRDetect. Yes, this is a small number, but this is a limited study. If this were extended to a population level incidence of cancer, the numbers would be more substantial in clinical practice.

b) In the validation cohort, MMRDetect missed the following 4 samples:

1. PD5937a (APOBEC hypermutator)
2. GEL-2414040-11 (SBS6)
3. GEL-2495139-11 (mixed)
4. GEL-2925291-11 (mixed)

These samples were missed by MMRDetect because MMRDetect uses the cosine similarity between the SBS profile of tumor and that of MMR knockouts as one of its input parameters. In addition to having the expected MMRd-associated SBS, these samples were either subclonal for MMRd or were also dominated by other processes, such as UV exposure SBS7 or APOBEC SBS2/13 hypermutation, causing the MMRDetect-calculated probability to be penalised, resulting in false negative calls by MMRDetect.

Additionally, MMRDetect uses only the experimentally derived MMRd-SBS signatures (*i.e.*, SBS26 and SBS44), precluding the possibility of identifying samples with MMRd-SBS signatures other than SBS26/44, such as SBS15 and SBS97. PRRDetect uses SBS6, 26, 44, 15 and 97 – all SBS signatures that have been associated with MMRd, for MMRd prediction and indel signatures as well.

Critically, all MMRDetect-predicted MMRd samples are also captured by PRRDetect. The prediction results of all algorithms used, wherever possible, of all samples analysed in the manuscript, were provided in **Supplementary Table S7 (formerly S6)**.

RTRv4

These are thoughtful deliberations based on the distribution of mutations (actual data in Extended Figure 3), across multiple contexts involving most of the major cancer types as a data not arbitrary decision. Our approach is data driven and unbiased, this is already in action today. Following WGS, each genome is read for all its drivers, gene fusions, germline mutations, somatic signatures, TMB, Myriad LOH score, HRDetect score, and PRRDetect scores as a matter of course. From a large body of published work, we know that indel formation is commonly driven by the underlying sequence motifs. While many in the community have been introduced to COSMIC3 for signature analysis, it is not as widely adopted as the substitution channels because it has been problematic and simply because it was the first, and has been introduced by the Sanger/COSMIC team does not mean that it should be dogma. Many have struggled and found it challenging to interpret the COSMIC ID signature results. This is because COSMIC ID lacks detailed sequence context information, resulting in peculiar signature assignments, leading to uncertainties in the interpretation of results (PMID: 38361034). This is not only apparent in our RPE1 dataset presented here, but also has been shown by others (PMID: 35320711, 35140396) that the COSMIC3 is not able to resolve many biological phenotypes. We would like to thank the reviewer for raising this point, however, as it has led us to make some changes to the current model. It is with these biological observations as motivations and also based on additional insights gleaned from any mechanistic studies (PMID: 30982602, 32661091, 35320711, 35140396) showing that a lot of mutational processes and mutagens show a predilection for specific sequence contexts to form adducts/damage and cause indels (e.g., at TT, etc) that we expanded on the current COSMIC3 framework to include the considerations outlined above.

#Point 16

Therefore, we maintain that our classification system is not arbitrary. It is informed by biological observations, and guided by the understanding of the underlying data distribution. It is also noteworthy that we in addition to capturing all previous COSMIC3 signatures with confirmed aetiologies also identified multiple new signatures in the 7 GEL cancer types, while providing additional insights/dimensions to guide signature interpretation. SBS6 is not due to polymerase dysfunction. Perhaps the reviewer was referring to using solely SBS10a (POLE) and We stand by the validity, robustness and biological value of our approach and contrast PRRDetect's performance against all other possibilities. The results were provided in **Extended Data Table 1**, and copied below for you:

#Point 1

1. One way to present the new taxonomy is as a modification / improvement of COSMIC3. There seem to be two key differences with COSMIC: First, longer repeat lengths (up to 9) are captured. Second, the base context to the 5' and 3' of a mononucleotide repeat is considered. Is this true? Can these two choices be concisely justified by statistical data analysis or cited biological first principles?

Model	Signature	Prop	ACC	WM	MED.AC	CS std	λ std	Med Coef. Std	Median Multicl	Bes t α WM	Best λ WM	Multicl ass AUC
Sigs	Prop	0.9947	0.9965	0.433	0.000	0.000	0.461	0.9961	0.05	1E-07	0.9975
Sigs+R	Prop	0.9947	0.99127	0.433	0.000	0.000	0.461	0.9961	0.05	1E-07	0.9977
Sigs+TMB	Prop	0.3748	0.377	0.2	0.02	0	0.5	0.65	0.099	0.5	0.5
Only IND	Prop	0.9807	0.9825	0.333	0.000	0.708	0.9844	0.1	1E-07	0.9932	0.9932
Only SBS	Prop	0.965	0.9651	0	0	0.16	0.9734	1	1E-07	0.968	0.968

Shaded row shows the model selected as the final PRRDetect classifier.

Model:

- Sigs, using both SBS and InD signatures;
- Sigs+R, using SBS, InD signatures, and (total indel count/total SNV count) ratio;
- Sigs+TMB, using SBS, InD signatures, and tumour mutation burden (TMB);
- Only IND, using InD signatures only;
- Only SBS, using SBS signatures only.

Current PRRDetect model (no.2 in the table above) was chosen because of its highest test set median accuracy, AUC, and stability over other models with single parameters (i.e., SBS only, InD only).

The response has dealt with the reviewer concerns.

#Point 17

17. Similar for cell models - the study focuses on indels, but how much additional accuracy do indels provide on top of SBS6+10 for polymerase dysfunction detection in these models? Extended Figure 2g shows cosine similarity but does not address this specifically.

This question is very similar to **#Point 16**. SBS6 is not used to define polymerase dysfunction. SBS10a may be used on its own to denote polymerase epsilon dysfunction. However, like any other classifiers, with additional strands of orthogonal evidence, (e.g., another signature from another class of mutation), the ability to be more specific is higher. This is a well-known principle – multiple strands increase specificity of signal. In situations where the polymerase dysregulation is in a sample with low tumor cellularity or is in a subclone, the addition of orthogonal signatures is critical for accurate classification.

The response has dealt with the reviewer concerns.

#Point 18

18. How does the new taxonomy, signature system, and classifier perform on mutants profiled in the 2021 Nature Cancer paper? Given the overlap of the previous WGS data on knockouts (including MMR knockouts), it would seem fitting to analyze this.

This analysis was performed previously and included in the manuscript already.

Please see **Extended Data Figure 8b** and **Extended Data Figure 9** where we showed that across knockouts of overlapping genes in two different human biological systems (RPE1 and iPSC), the genotype-specific experimentally derived signatures are extremely similar. This is demonstrated plainly through hierarchical clustering in **Extended Data Fig 9**, which puts all the relevant iPSC and RPE1 genotypes together, with cancer derived InD signatures.

The response has dealt with the reviewer concerns.

#Point 19

19. Taking a step back, it would seem more interesting to build a classifier to diagnose repair dysfunctions more specifically than what the authors are doing here, rather than just 3 dysfunction categories, two of which seem quite rare (Pol-dys, MMRd/Pol-dys). Why can't the RPE1 data be used (in combination with previously published 2021) and cancer genomics data to build such a classifier? This would tie together the two currently disparate parts of the paper.

We agree that academically, it is terrifically interesting that there are such marked gene-specific signatures. Mismatch repair deficiency is not simply “MSI” or “a high TMB”.

- We show this experimentally
- We find evidence of this in the cancer datasets and
- We made this assertion in paragraph starting line 295/428, “**eight gene-specific MMRd and polymerase-dysfunction InDs. ...**”

However, clinically, gene specificities are not required for treatment. All MMR gene defects are treated equally. Whether this is the best way to treat patients or not is another matter that is beyond our ability to resolve/and the scope of the manuscript. We see the same situation with *BRCA1/BRCA2*. Currently, all the customary assays simply denote tumours as *BRCA1/2*-deficient or proficient and do not require distinguishing between *BRCA1/BRCA2* mutated cancers. Yet, they are different genes with different physiological functions, producing subtly different signatures. This is a known generic problem, and not something that we can address.

However, it is an important philosophical point. Perhaps the day will come where we need to highlight to clinicians/trialists, that there are gene-specificities and perhaps this plays a role in treatment outcomes. If the reviewer is happy with this, we are happy to make the point in the discussion which we think will elevate the argument.

RTR.v4

Line 446: "Finally, our classifier does not distinguish between the MMR-deficiency genotypes although there are clear differences between *MLH1/MSH2/MSH6* and *PMS2*, for example. Currently, there is no clinical indication to do so. However, should it become clinically important to distinguish between these genes, it shall be possible to do so."

I agree with the authors that this is an interesting suggestion but is outside the scope of the paper.

#Point 20

20. The strand bias signals are interesting, though it seems that more could be made of this. A strand bias not only reflects a difference in mutation rates but also a pyrimidine vs. purine preference for mutations arising on that strand. In other words, while the convention is to align mutational strands to pyrimidines, a leading strand bias implies either an increased pyrimidine mutation rate on the leading strand or an increased purine mutation rate on the lagging strand. Under the biological assumption that dysfunction of POLE and POLD increases mutation rates on the leading and lagging strand, respectively, their data implies that both POLE and POLD dysfunction preferentially causes T rather than A insertions. How do the authors explain this?

When strand asymmetry is observed, there are two possible reasons (covered elsewhere extensively). There may be a bias of DNA damage or there is a bias in DNA repair.

For environmental mutagens which are known to cause DNA adducts that require the activity of transcription-coupled repair, we can confidently assert whether the DNA damage occurred on a purine or a pyrimidine base, based on **transcriptional strand asymmetries** (and not replication strand asymmetries).

- For example, C>T mutations of UV light, when inspected for transcriptional strand bias, shows an excess of C>T mutations on the non-transcribed strand more than the transcribed strand. This is due to the activity of transcription-couple repair (TCR) preferentially repairing DNA damage on the transcribed strand. This also fits with the known information that UV causes covalent modifications on pyrimidines.
- By contrast, C>A mutations of tobacco smoke, when inspected for transcriptional strand bias, shows an excess of C>A mutations on the transcribed strand more than the non-transcribed strand. Because the imprint of TCR activity is the excess of mutations on the non-transcribed than the transcribed strands, we can know that the true DNA damage must be on the G (purine), and not on the C. This is in keeping with known biochemistry of guanine adduct formation caused by polycyclic aromatic hydrocarbons.

For endogenous mutagenesis caused by defective repair processes, we cannot quite make the same assertion as above. However, we have argued and shown (which we think the reviewer agrees with), that there is a bias of mainly T insertions on the leading strand for *POLE-mut* (which is known to be the major replicative polymerase of the leading strand, and the opposite is seen for *POLD1* (which preferentially replicates and proof-reads the lagging strand). Biochemical analyses have shown that B-family polymerases are particularly proficient at discriminating between correctly and incorrectly paired bases at template adenine (PMID: 21036870), *i.e.*, less accurate on replicating template thymine.

In agreement with published work (PMID: 33764464), our data suggests that the *POLE/POLD1* polymerase mutants frequently accumulate +A insertions (on the nascent strand) particularly when replicating through T-homopolymer of specific tract lengths (5-7nts). We note that we have not overtly made the point about T insertions in the manuscript. Given that the reviewer has kindly raised it, we have now included it in:

Line 119: ..., specifically T insertions at polynucleotide repeat lengths of 5-7 nts (Extended Data Fig. 3).

Line 120: This is in keeping with the hypothesized preferential activity of Pol ϵ and Pol δ in leading and lagging strand synthesis respectively^{23,24}, suggesting that *POLE/POLD1* mutants tend to accumulate 1bp A insertions on the nascent strand while replicating through 5-7 nts poly-T-tracts (PMID: 33764464, 21036870). This aligns with the proposition that polymerase ϵ and δ are more proficient at detecting incorrectly paired bases at template adenine (PMID: 21036870).

The response has dealt with the reviewer concerns.

RTR.v4

**#Point 21**

21. Since the study does not include immunotherapy analyses, it does not seem appropriate to mention
immunotherapy in the abstract. To the reviewer, it does seem likely that the modest improvements in PRRd
accuracy provided by the classifier here will improve prediction of immunotherapy outcomes.

Clinically today, the reason for identifying MMR-deficient and PRR-deficient cases is because these patients are
candidates for immunotherapies. Hence, we have stated originally in the abstract:

“Furthermore, we develop a highly specific classifier of PRRd status in tumors, PRRDetect, with potential implications
for immunotherapies”.

We have **not** said “...with superior performance in immunotherapies”, which would have required an analysis. The
use of the word “potential” communicates the tentativeness; it implies that it is possible or likely, but has not yet
been realised or confirmed.

We are happy to defer to the editor on this point but we are often encouraged to say what “impact” a manuscript
has. Perhaps we could suggest this change:

“Furthermore, we develop a highly specific classifier of PRRd status in tumors, PRRDetect, ~~with~~ given potential
~~implications~~ for immunotherapies”.

I think the original wording is fine.

**#Point 22**

22. Authors cite space considerations, but it seems that some of the main panels (such as 1f, 1d, 3b, 4a, 4e) could
be moved to extended data or substantially condensed to show key signals. Also NG allows for up to 7 figures.

I am not clear what this suggestion refers to. I would think shuffling of the figures to accommodate space concerns
should be fine if necessary.

We’re happy to discuss the path forward with the editor/reviewer.

We thank you for all your feedback and comments, which have helped us clarify issues and improve the
manuscript.

RTR.v5 (December 2024, 10)

KEY to colour coding:

Current arbitrating reviewer: in green.

Previous reviewer's comments: in black

Previous RTR from authors: in blue.

Current RTR from authors: in purple.

Please find below my comments on the previous review. Overall, I am supportive of this work as it improves our understanding of indel generating processes immensely. I only have one major point that needs addressing (other suggestions are minor). The PRRdetect performance evaluation requires important clarifications regarding the validation set labelling before I can fully assess its utility (see below). In a related point, I believe any statements implying clinical utility need to be moved from the results to the discussion, as this is more speculative at this stage. To claim clinical utility, at a minimum PRRdetect would have to be assessed against current clinical diagnostic standards on the same samples (for example, IHC +microsatellite PCR), especially as TMB has recently been shown to be inadequate. Please also remove the word clinical from the section header.

We thank the reviewer for their constructive and positive feedback on our revised manuscript. Given the extensive nature of the previous rounds of RTR, we aim to summarize and address the remaining concerns raised by the reviewer below. The **ITEMS #** listed have been cross-referenced to the original RTR for clarity.

ITEM #A

As per reviewer's suggestion, we've removed the term "Clinical" from the section header and revised it to "A signature-based classifier of PRR dysfunction" – line 338.

Specific critiques

- **The indel taxonomy still seems arbitrary, and not really justified by biological principles or data (including the RPE1 data), especially to a general genetics reader. While it is clear from their paper that COSMIC-83 is lacking, it is not convincingly shown that the proposed taxonomy is the natural successor.**

If we could discuss the logic here: By extension of the reviewer's argument, the COSMIC-83 indel classification would be entirely arbitrary and not justified by biological principles/data, because:

- It is not constructed through examination of any underlying data, as plainly demonstrated in panel a of **Extended Data Figure 3 (Former Extended Data Figure 4)**.
- Because of limited power, the channels were arbitrarily capped at homopolymers of 6bp in lengths.

We wonder why that is considered reasonable. Yet our position, which is a systematic experimental demonstration of the inadequacies of the current indel classification system, with logical suggestions of improving it by extending the homopolymer run to 9bp, introducing flanking sequence context (a system used for substitutions), justifying it in the experimental data and showing that it works in cancer, is not considered adequate and is dismissed rather easily.

We are suggesting our taxonomy as an alternative methodology for indel classification, which could reveal more biological granularity and is able to disambiguate PRRd signatures and/or uncover new signatures, as demonstrated by our RPE1 experimental data and cancer data. We have not, at any point, pitched the new classification system as the "natural successor" to COSMIC-83. Further, we acknowledge in line 454 that "**Nevertheless, optimal classification remains an active research area, and alternative schema could unveil additional mutational processes in the future**".

Our proposed classification expands on the current COSMIC-83 taxonomy, with key additions, considerations and modifications as follows:

1/ Incorporating the 5' and 3' flanking bases for 1bp indels.

2/ Extending the cap on homopolymer length from 6 to 9bp, to allow for better discriminatory power as we observed that there are more indel signals at longer homopolymers (**Extended Data Figure 3a,e**), which may offer the resolution required to tease out distinct biological phenotypes. Note that the frequency of homopolymers in the genome decreases with increasing homopolymer length (see **Point 2** figure). There's

RTR.v5 (December 2024, 10)

a trade-off between being exhaustive to account for all the homopolymer/indel lengths and conserving power for signature extraction.

3/ Though longer repeats (up to 9bp) are captured in our classification, to improve the indel *signal-to-channel* ratio, we grouped the homopolymer lengths into short (0-4bp), medium (6-7bp), long (8-9bp) based on how they are distributed (**Figure 2a, Supplementary Note 1, Extended Data Figure 3**), to prevent over-sparsity. This was an informed decision based on the distribution of indels/homopolymer tracts in the reference genome. This was not arbitrary.

4/ We aggregated channels where indel signals were low across the pan-cancer cohort data because they would contribute limited value (*i.e.*, no point in having empty channels) while costing statistical power for analyses.

These are thoughtful deliberations based on the distribution of mutations (actual data is in **Extended Figure 3**) across multiple cohorts involving most human cancer types and are not arbitrary decisions. Our approach is data-driven and unbiased.

From a large body of published work, we know that indel formation is commonly driven by the underlying sequence motifs. While many in the community have been introduced to COSMIC-83 for signature analysis, it is not as widely adopted as the substitution channels because it has been problematic, and simply because it was the first, and has been introduced by the Sanger/COSMIC team, does not mean that it should be dogma. Many have struggled and found it challenging to interpret the COSMIC ID signature results. This is because COSMIC-ID lacks detailed sequence context information, resulting in peculiar signature assignments, leading to uncertainties in the interpretation of results (PMID: 38361034). This is not only apparent in our RPE1 dataset presented here, but also has been shown by others (PMID: 35320711, 35140396) that the COSMIC-83 is not able to resolve many biological phenotypes.

It is with these biological observations as motivations and also based on additional insights gleaned from many mechanistic studies (PMID: 30982602, 32661091, 35320711, 35140396) showing that a lot of mutational processes and mutagens show a predilection for specific sequence contexts to form adducts/damage and cause indels (*e.g.* UV at TT, etc), that we expanded on the current COSMIC-83 framework to include the considerations outlined above.

Therefore, we maintain that our classification system is not arbitrary. It is informed by biological observations, and guided by the understanding of the underlying data distribution. It is also noteworthy that we, in addition to capturing all previous COSMIC-83 signatures with confirmed aetiologies, also identified multiple new signatures in the 7 GEL cancer types, while providing additional insights/dimensions to guide signature interpretation.

We stand by the validity, robustness, and biological value of our approach.

I agree with the authors here, although I note that the language used to describe the COSMIC-83 results could be perceived as disparaging. I suggest the authors alter the text in the manuscript to adopt a tone of discovery rather than competition between signature taxonomies. The authors also need to make it more explicit that the new taxonomy is not a replacement for the old taxonomy. Rather, it is one that better captures differences in the mutation patterns between types of post replicative repair dysfunction.

Some suggestions where the text could be reworded:

Line 29: ...prevailing indel classification framework for deriving mutational signatures falls short...

Line 138: We thus questioned the sufficiency of COSMIC-83 taxonomy....

Line 145: Surprisingly, indel signatures of Δ MSH2 and Δ MLH1 showed no similarity to the purported MMRd-associated ID7

Line 152: Yet, it is widely known

Line 415:disregards surrounding sequence context...

ITEM #B

Thank you. We have revised the language accordingly, while maintaining the factual accuracy of the points discussed. Below are the specific changes made:

Line 29: "prevailing indel classification framework for deriving mutational signatures falls short in" -> "...is unable to ..."

RTR.v5 (December 2024, 10)

Line 134: "We thus questioned the sufficiency of COSMIC-83 taxonomy..." -> "... investigated..."

Line 140: "Surprisingly, indel signatures of ΔMSH2 and ΔMLH1 showed no similarity to the purported MMRd-associated ID7" -> "Surprisingly, ..."

Line 148: "Yet it is widely known" -> "Yet, it is widely known that the probability of indel ..."

Line 416: "...disregards surrounding sequence context..." -> "... does not take surrounding sequence context into account ..."

#Point 2

2. Along these lines there is data in the supplement, including histograms of repeat lengths which actually show a substantial fraction of indels with repeat lengths of 9 or greater (for example 1bp T indel). Does it really make sense to collapse them? In other words, why draw the line at 9+? If the authors are looking to rethink the indel taxonomy, are they being bold enough?

This is not correct. The histogram shows indels up to repeat lengths of 9, and not 9 or greater. See below for the zoomed-in example of all possible 1bp T deletions. The same applies to all homopolymer, polynucleotide repeats (i.e., repeat # <= 9)

1 bp T indel channels (81 subcategories)

It is not possible using today's genomic data pipelines of any institution, to confidently go beyond repeat lengths of > 9 because of the error rates in mutation-calling algorithms. Nearly all commonly-used genomic pipelines (Broad GATK, Dragen, Strelka, Platypus, Pindel, etc) have a post-hoc filter capped at repeat lengths < 10.

Genome distribution of polynucleotide repeat tracts

The contribution of indels at very long homopolymer tract length of >9 to indel signatures could be limited given that there is a much lower prevalence across the genome (see image above). Having said that, we agree that indels at longer homopolymers may be important and should be investigated at a later date when indel data obtained through newer lower-error/long read technologies become available, in quality and quantity. Thus, we have added the following statement:

RTR.v5 (December 2024, 10)

Lines 432: The possibility of adapting the taxonomy in the future, to include features that are currently not explorable due to the limitations incurred by technological error rates, could also be revealing.

I can see where the reviewer is coming from here - the figures in the supplementary note are misleading and suggest +9. Please fix these. The author response here highlights an important technical limitation, and it would be good to see this mentioned explicitly (with the above text or similar), perhaps in the supplementary note.

ITEM #C

Thank you. We have revised the texts and figures accordingly to show that repeat/polynucleotide repeat lengths are only considered up to 9 instead of 9+.

Additionally, we have added to Supplementary Note 1 (line 42-45): "Within our described framework of small indels, we focus specifically on insertions, deletions, and complex indel events with lengths < 100bp; homopolymer repeat and polynucleotide repeat with units < 10 because of the high error rates in mutation-calling algorithms at long, simple repeats; and a 3' sequence context of up to 200bp in length."

#Point 5

5. How are insertions with microhomology classified? Namely, insertions aligning with a spacer to flanking sequence. It does not seem like they have a category here.

Insertions with microhomology are a rare feature (single digits if at all), even in HR-deficient samples, where they are alleged to be seen more frequently. *Critically, they cannot be unambiguously mapped to a single category.* That's because templated insertions are:

- often complex (*i.e.*, they are accompanied by deletion at the junction). Majority of them would have been overlooked or mis-annotated using standard indel variant callers (PMID: 26657142).
- could also be templated from inverted repeats (both left and right flanks) and not be immediately at the junction of the indel, and unaccompanied by deletions.

Although we do not designate a category specifically for these complex events, they were segmented and classified using our unified framework into various non-overlapping "insertion" and "complex" categories. Due to the complexity of these events, they are best characterised by specialised algorithms (PMID: 32234782, 26657142).

This is another important technical limitation which should be explicitly mentioned in the manuscript or supplement (with the above text or similar).

ITEM #D

Thank you. Added to Supplementary Note 1, starting from line 48:

"Insertions with microhomology are segmented and classified using our unified framework into various non-overlapping "insertion" and "complex" channels for we currently do not have a designated category for these complex events. This is because templated insertions are rare, and often complex (*i.e.*, they are accompanied by deletion at the junction). Majority of them would have been overlooked or mis-annotated using standard indel variant callers³. Furthermore, they could also be templated from inverted repeats (both left and right flanks) and not be immediately at the indel junction, and unaccompanied by deletions. Due to the complexity of these events, they are best characterized by specialized algorithms^{3,4}.

#Point 7

- There still appear to be some circularities in the classifier training and testing, including a concern about data leakage.

7. The new 89-channel system is derived from the entirety of pan-cancer data (18522 samples) which seems also to include samples used to test PRRdetect, including ICGC, TCGA, and HMF. Since these test cohorts (ICGC, TCGA, HMF) were used for feature engineering (Extended Fig 4, supplementary note) they are not independent. The

RTR.v5 (December 2024, 10)

feature engineering and training of signatures and classifiers should be done on a single cohort, for example GEL, leaving the other datasets for testing, including ROC / AUC analysis and calculations of sensitivity and specificity.

Note that the former **Extended Fig 4** is now revised as the new **Extended Data Figure 3**.

The reviewer has conflated several distinct pieces of work together in the argument above. There are three distinct pieces of work:

- 1) The demonstration of validity of the 89-channel system
- 2) Indel signature extraction from seven cancer types
- 3) The PRRDetect classifier development

These are different analyses. Historically, a paper would be done on (1) and (2). For example, the substitution signatures (Cell 2012a, Nature 2013) are an example of analyses similar to items (1) and (2). Later, some type of classifier might be created (often using past publicly-available data because that's what is available and we need something as a "gold standard"). For example, HRDetect (Nature Medicine 2017) – this would be similar to item (3). These are different analyses resulting in different papers. In this manuscript, we have argued for an alternative classification taxonomy (1) and shown how it would be used to find signatures (2). Item (3) is just like HRDetect, it is a separate exercise, and could easily have been a separate paper. These are simply different analyses.

1) The 89-channel system

The entirety of the pan-cancer data of 18,822 samples was *simply used to show the indel distribution across all categories from available cancer types across all three cohorts (GEL, ICGC and Hartwig)*. Because the same reviewer previously raised concern that our proposed framework was potentially "biased towards PRRD phenotype", we presented the plots in the now **Extended Data Figure 3** to show that:

A/ indel distributions across InD channels are preserved regardless of cohorts or cancer types, and are not biased towards PRRd phenotypes or skewed towards hypermutator samples (Extended Data Figure 3e).

B/ Additionally, the new indel classification channels have data in all channels across all cancer types, in contrast to the sparse profiles with COSMIC-83 (**Extended Data Figure 3a-d**).

§ ratio, and importantly, shows that our final channel set does not bias towards hypermutator phenotypes (and hence PRRd samples), and reinforces that our approach is data-driven.

2) Indel signature extraction

In the main text (starting from line 251), we clearly stated that indel signature extraction *was performed on the 7 selected cancer types within the GEL cohort*. An indel signature extraction has only been performed on 4,775 GEL samples, not on the whole cohort of 18,822 samples. This is unrelated to **Extended Data Figure 3**.

3) PRRDetect

In this also entirely separate section of the manuscript, we have developed PRRDetect as a multinomial classifier for polymerase dysfunction and MMRd.

TRAINING of PRRDetect

It was trained on **571 GEL** samples (a subset of the 4,775 GEL cancers), where

- i. 291 were confirmed to have driver mutations/ IHC staining.
- ii. 280 were negative set.
- iii. The training cohort of 571 went through a ~ 7:3 split for training.

These samples have confident PRRd labels. Ground truth is solid for all samples.

VALIDATION of PRRDetect

PRRDetect was *validated on an independent cohort of 504 ICGC breast samples for which PRRd labels were available (Line 360)*. This cohort was used specifically for validation because we are confident of the PRRd label of

RTR.v5 (December 2024, 10)

all samples, informed by IHC staining and careful curation of all drivers. We know that we are not missing any true PRRd samples. *These samples have confident labels. Ground truth is solid for all samples.*

Also note that during this revision of the manuscript, we have expanded on our validation cohort to include additional samples (not used for PPRDetect training) for statistical power. See #Point 14 for details.

GENERAL APPLICATION

PPRDetect was then applied on ICGC and Hartwig samples (starting line 367, $n=1,335$) to make illustrative descriptions of generalisability. For these samples, the ground truth status is not (completely) known. We only have driver information that has been published by others. *These samples do not have confident labels. Ground truth is not so solid; hence we have not used them for validation.*

We showed, however, the concordance/discordance from different prediction algorithms, and whether predicted PRRd samples have annotated driver mutations or not (**Figure 4e**). Manuscript lines 359-385. Please note, this is not the same thing as testing or training.

Additionally, all samples used (whether in training or validation, whether positive or negative, with driver and/or IHC) are all annotated in Supplementary Table S7 (former S6).

I am generally happy with the authors response here although I do believe the presentation of the GENERAL APPLICATION in the manuscript is misleading. Reporting AUC in this section implies there is some type of gold-standard labelling. While the authors acknowledge that this labelling is based solely on mutation status, I don't think they should be reporting AUC using these data alone as there will be many PRRd cases without driver mutations (as acknowledged by the authors later in the manuscript). AUC indicates some type of classification-based performance evaluation which the data doesn't really support. Rather, a more appropriate use of the data would be to report on prevalence of positive cases, concordance with mutation status, and overlap with other methods.

ITEM #E

Thank you. The reviewer raised a fair point. We have removed the AUC statements from the aforementioned paragraph. Instead, we now only report the concordance and discordance between different algorithms. The prediction results of all algorithms are also available in Supplementary Table S7. Changes in the manuscript begin from line 368:

Original:

~~“Next, to survey the prevalence of PRRd across alternative cancer cohorts, we applied PPRDetect on seven cancer types commonly enriched with hypermutator samples from ICGC³⁴ and Hartwig³⁵ (Fig. 4c, e-f, Extended Data Fig. 10a). Based solely on available published driver information for PRRd status, PPRDetect could distinguish Pol-dys and mixed samples from polymerase-proficient samples with an AUC of 0.999, compared to an AUC of 0.83 if TMB (10 muts/Mb) was used. Overall, PPRDetect performed superiorly to current biomarker strategies, correctly predicting all Pol-dys, MMRd/Pol-dys samples and missing two subclonal MMRd samples. MSIseq missed six of 43 PPRDetect-predicted MMRd, two of four mixed MMRd/Pol-dys cases while displaying poor concordance for detecting pure Pol-dys cases.”~~

Revised:

“Next, to survey the prevalence of PRRd across alternative cancer cohorts, we applied PPRDetect on seven cancer types commonly enriched with hypermutator samples from ICGC³⁴ and Hartwig³⁵ (Fig. 4c, e-f, Extended Data Fig. 10a). PPRDetect predicted 3.7% (50 of 1,335) samples to be PRR-dysfunctional, correctly identifying all Pol-dys, MMRd/Pol-dys samples and missing two subclonal MMRd samples (based on available published driver information for PRRd status). MSIseq missed six of 43 PPRDetect-predicted MMRd, two of four mixed MMRd/Pol-dys cases while displaying poor concordance for detecting pure Pol-dys cases (*i.e.*, missed all seven cases).”

Additionally, we have also added to Extended Data Fig. 10a, Venn diagrams showing the overlaps between different predictors:

#Point 8

8. Related to above, it is not clear whether new ICGC breast analysis was done on samples included in the ICGC analyses shown in Extended Fig. 4.

The former **Extended Fig 4** is now revised as the new **Extended Data Figure 3**.

No. The ICGC breast samples used for PRRDetect testing/validation were not used in any analyses in **Extended Data Figure 3**. In fact, **Extended Data Figure 3** has nothing to do with PRRDetect. It is simply showing what the distribution of indels are across all cohorts, according to the COSMIC-83 and the new 89-channel system. See also **Point 7** – “1) The 89-channel system”.

Extended Data Figure 3 is not related to PRRDetect at all. It does not bias the PRRDetect analysis because it not related to the PRRDetect analysis.

This is clear from the authors. However, given the reviewer’s mix-up here, perhaps it is worth making these distinctions extra clear in the text?

Thank you. The Figure Legend reads:

“**Indel frequency distribution across pan-cancer datasets and the consolidation of 476 full indel channels of the redefined indel taxonomy into a refined 89-channel framework.** a, Aggregated COSMIC-83 indel profile of ICGC samples, with (*left*) and without (*right*) hypermutator samples. b, Aggregated 89-channel indel profile of ICGC samples, with (*left*) and without (*right*) hypermutator samples. c, Percentage of total indels per channel for ICGC cohort, with COSMIC-83 (*left*) or 89-channel (*right*) format. d, Percentage of total indels per channel for Hartwig cohort, with COSMIC-83 (*left*) or 89-channel (*right*) format. e, Aggregated 476-full-channel indel profiles of ICGC, Hartwig and GEL cohorts. f, Final, consolidated 89-channel indel profiles of aggregated samples from GEL cohort, with (*left*) and without (*right*) hypermutator samples.”

#Point 9

9. It is still unclear how gold-standard labels are derived - namely what are the criteria for labeling a sample as PRRd or PRR proficient. Methods currently refer to "manual curation" for GEL samples and for the remaining samples, "true labels were established based on IHC of four MMR proteins and driver mutations". The authors should offer much more detail about these criteria and how frequently each was applied (in how many samples IHC vs. driver mutations vs. combinations was used to determine the label, including whether / how allelic status of driver mutations was assessed).

Regarding the term “manual curation”

Please note the term “manual curation” for identifying drivers is not a negative thing. This means that not only was a driver identified through bioinformatic methods, we will have also:

- checked that the other allele was lost if it is a tumor suppressor and
- we physically inspected the variant using IGV.

This, in our view, is being thorough and doing our due diligence. It is not and should not be viewed in a negative way.

RTR.v5 (December 2024, 10)

In any case, we have removed the words “manual curation” from paragraph starting from line 349.

True labels for each sample

Note that a full, comprehensive list of every sample, whether they were used for training or validation, whether they had IHC or drivers is fully available in **Supplementary Table S7 (former S6)**. Note that:

- A mutation is only considered a driver in any of the MMR repair proteins (*MLH1*, *MSH2*, *MSH6*, *PMS2*) if it has also lost the wild-type parental allele and is thus biallelically null. Mutations in *POLE/POLD1* are only annotated as drivers if they are previously reported mutations in functionally relevant domains. This is what we mean by “curated” drivers - we have gone the extra mile to ensure that these are not simply passenger mutations that have hit these genes by chance.
- Samples were labelled with an IHC staining result if one was available. Not all samples have had IHC.

Thus, a positive PRRd label is assigned to a sample if one of the two criteria (driver or an abnormal IHC) are met.

That we have then additionally checked to ensure that these gold-standard labels are unambiguous, must surely be a strength, not a weakness. Samples that do not have the right gold-standard label and are not in the right category would be detrimental to training an algorithm. Hence, we also double checked that samples in the training cohort of 571 samples were unambiguously PRR-deficient or PRR-proficient by ensuring that the relevant signatures matched the labels, which they did.

I appreciate the authors manual curation efforts, but I agree more detail is needed regarding how the labels were assigned to each sample. It is not clear what evidence was used to create the labels in Supplementary Table S7. If IHC alone was used, or IHC plus mutation, mutation alone, signatures etc. One thing I find unusual is the use of the signatures themselves to confirm negative cases. What happens if a sample is negative for drivers and IHC but positive for PRR related signatures? Is it simply removed from the training set? If so, this is a biased training set and this should be more explicitly stated. I would have thought MSI could be used to assist with labelling as this is another source of information that is typically used in a clinical setting. Nevertheless, for the training cohort, the use of signatures, although biased, is ok for creating a cleaner training dataset. However, can the authors confirm that signatures were not used at all in the process of labelling the GEL hold-out dataset (this point is also covered below)?

ITEM #F

Thank you for your comments. We can confirm that the signatures were **not** used to label PRRd-positive or -negative samples during the construction of PRRDetect. Previously, we meant that signatures were simply used as an additional *post-hoc* check on the accuracy of the available IHC and driver information, but this was not helping and causing confusion, and has thus been removed.

To provide further clarity, if the reviewer refers to **Supplementary Table S7**, and filters the table using columns “Training_set”, “Test_set”, “IHC” and “Driver,” they will be able to identify which samples were included in the training versus the testing sets, as well as which samples exhibit abnormal IHC, IHC combined with driver mutations, or driver mutations alone. Here, we also summarise for the reviewer:

Of the 571 samples used in the construction of PRRDetect, 291 samples have positive PRRD labels; 33/291 have both an abnormal IHC and driver mutations, 258/291 have only the drivers; none have only abnormal IHC but no driver mutations.

To avoid further confusion and to maximise clarity, we have removed the wordings “~~displayed no evidence of signatures associated with these abnormalities~~” from line 354, and in Methods, line 931 “~~displayed no evidence of signatures associated with these abnormalities~~”. Instead, we confirm here and, in the main text and Methods: “negative samples displayed no evidence of MSI associated with these abnormalities⁴³”.

We have also included MSI status (from MSIsseq) for all 571 samples used in PRRDetect construction, and for 1,351 samples used in PRRDetect validation in Supplementary Table S7.

**#Point 10**

10. Specifically, how were true labels derived for samples / cohorts (like ICGC / HMF) where presumably IHC labels
are not available. (for example, Figure 4e). I presume it's using driver status. But then the authors refer to the false
negative rate of using strictly driver events (66%, line 365).

The reviewer was correct in saying that IHC labels were not available for ICGC/Hartwig cohorts. Hence, in line 369,
we stated that “Based solely on available published driver information, ...”. Where IHC is not available, we are
dependent on driver statuses reported by published ICGC or HMF studies. Note that many samples may be MMR-
deficient through promoter hypermethylation of *MLH1* and will not thus show a genetic “driver”. Hence, the point
at the end of the paragraph “should customary targeted sequencing strategies be solely used to identify driver
events associated with these deficiencies, a significant proportion of cases (66%) could be missed.”

That’s because targeted panels will only “see” the genetic driver mutations in ~33% of cases (17 of the 50 predicted
positive cases had reported drivers) and will miss all the MMRd/polymerase dysfunctional cases that are due to
promoter hypermethylation or other causes.

This paragraph, by the way, is not a training or validation point for PRRDetect. This is simply a demonstration of the
biological/clinical insight that if we stuck with targeted panels of MMR/polymerase genes, we would miss a
significant proportion of the cases. Given the lack of clarity, we have adapted the manuscript to clarify the assumed
context: “the assumption of PRRDetect predicted results being true”:

Line 383: “Should PRRDetect predictions be all true, and customary targeted sequencing strategies be solely used to
identify driver events associated with these deficiencies, a significant proportion of cases (66%) could be missed.”

The change on line 365 addresses the reviewer concerns. However, I am not sure I agree with the statement
regarding gene panels. Gene panels offer more than just driver mutations - it is possible to determine MSI status
and TMB, therefore making a statement only about drivers seems artificial.

Thank you. If we could dissect the statement carefully: “Should ..., and customary targeted sequencing strategies be
solely used to identify driver events associated with these deficiencies”, we did not make a statement regarding gene
panels in general. The sentence as we had written it refers to a scenario where if one only uses driver mutations
from panel sequencing to inform decision, we would miss these cases.

However, *per* the reviewer’s suggestion, we are happy to revise the wording and have since removed “customary
targeted sequencing strategies” from the sentence, which now reads, “Should PRRDetect predictions be all true, and
sequencing approaches were focused exclusively on identifying driver events associated with these deficiencies, a
significant proportion of cases (66%) could be missed.” – Line 383.

**#Point 11**

11. Reading through methods, signature analysis seems to have been used to pick PRRd and non PRRd samples for
GEL (“samples .. were confirmed to lack driver mutations .. and displayed no evidence of signatures associated
abnormalities” (lines 881-883). Were signatures (or the lack of) part of the criteria used to determine PRR
proficient samples? If so, was this done only for training or for testing? If so, this could obviously bias either training
and testing or both.

Please see response to **#Point 9** above. **To define the ground truth set of PRRD and PRR-proficient samples, high-**
**confidence driver mutations (including allelic status) and/or IHC were used.**

Having a clear ground truth set is incredibly important. Thus, to be exacting and highly conservative, we ensured
that there was no ambiguity at all in the ground truth set: So, we manually checked and were sure that:

- True positives really did have drivers and the associated signatures.
 - True negatives really did not have drivers or any hint of any similar signatures.

It was not a criterion. It was a simply a check. We are unclear why this is an issue. This would usually be preferable over the alternative, of not checking.

I share some of the reviewers concerns here given the lack of clarity over precisely how labels were assigned (see comment on point 9 above). For the validation set, the signatures should not be used at all during the labelling process as this would create a biased dataset. Any removal of mutation negative cases with positive signature levels will artificially inflate performance.

See Point #9 above (ITEM #F). To reinforce, the signatures were not used at all during the labelling process.

#Point 13

13. It is unclear which groupings refer to algorithm outputs rather than true labels (meaning those derived from manual curation, IHC, and / or driver analysis). This is additionally complicated by the fact that the authors have published a previous algorithm MMRdetect with overlapping outputs to PRRdetect. For example, in 4f and 4g the caption implies that some of the labels (MMRd, Poly-dis) are algorithm outputs, however in the previous panel MMRd and Poly-dis refer to the true labels. Also in Extended figure 10, among other places in the manuscript. It would help to use uniform and unambiguous terminology to make this distinction clear across the manuscript.

Figures 4e,f,g and Extended Data Figure 10 are all PRRDetect algorithm outputs. These were all specified in the figure legends, starting from line 553 and are copied here for you:

e, PRRDetect results of n=1,335 ICGC and Hartwig cancers, ordered from the lowest to the highest prediction probability across the x axis (left to right) for polymerase-dysfunctional samples (orange), combined mismatch-repair deficient (MMRd) samples (blue), and MMRd samples (purple). Negative samples were ordered by TMB in increasing order from left to right. Results of MSIsq, MMRDetect, cancer gene driver annotation, and cancer tissue origin are labelled at the bottom tracks. Dashed rectangle highlights the extent of false positive overcalling if using TMB > 10 muts/Mb as a cutoff.

f, Concordance of calls among TMB-H (> 10 mutations/Mb), positive exposure to SBS signatures that impart hypermutation, and PRRDetect prediction across n = 1,335 ICGC and Hartwig cancers.

g, Concordance of calls among TMB-H (> 10 mutations/Mb), positive exposure to SBS signatures that impart hypermutation, and PRRDetect prediction across n = 4,775 GEL tumors."

However, given the query and the potential lack of clarity, we additionally labelled Figure 4 panels and Extended Data Figure 10 with "PRRDetect" on the figure panels:

To make it impossible to miss, we also incorporated main text changes to clarify PRRd predicted labels in:

Line 389: "just over a tenth of 459 cases classified as TMB-High (50/459, 10.9%) had predicted PRR dysfunction".

Line 397: "Among the 1,371 TMB-H cases, nearly half the cohort (677, 49.4%) was predicted as having MMRd and/or polymerase dysfunction (Fig. 4g)."

RTR.v5 (December 2024, 10)

These changes are good and help with clarity, although the authors should acknowledge other relevant literature showing the shortcomings of TMB (e.g. <https://pubmed.ncbi.nlm.nih.gov/33736924/> among others).

Thank you. Cited in line 396 and 442. Also added PMID: 31145420. Reference 50, 51.

#Point 14

14. The ICGC breast cohort used for validation (Figure 4d) does not appear to have many positive (PRRd) samples, just judging by the shape of the curves. How many Are the results in Figure 4d statistically significant? In other words, is the AUC generated by PRRdetect significantly higher than the previously published methods including MMRdetect?

Indeed, the ICGC breast cohort has only 10 PRRd samples. PRRd is rare in other cancer types but we chose to use a different, independent cohort from the training set for validation, and where we have solid, confident PRRd status of the samples.

PRRd is more common in colorectal and uterine cancers. However:

- We cannot reuse the colorectal/uterine cancers from GEL cohort because as the reviewer has argued above, we shouldn't reuse the training set for validation.
- ICGC colorectal (n=52) and uterine cohorts (n=44) are very small, because WGS has not been performed much in these tumor types, and crucially, there are no MMRd samples (n=0) based on driver information. There are 7 colorectal and 1 uterine cancers with polymerase driver mutation. That's fewer than the validation cohort that we did use.
- The colorectal and uterine cohorts of Hartwig are also not large (n=379 and n=42, respectively).
- More critically, the ground truth labels of samples in these additional cohorts are not solid (allelic status not defined, IHC not available, and risks not being informative).

However, to increase the sample size of the validation cohort, and hence, the power of PRRDetect validation, we have now added to the n=504 breast cohort, an additional n=847 GEL cancers (see revised Figure 4c below) with confirmed PRRd status for 104 positive and 743 negative samples. Critically, these samples are completely independent, and never used in PRRDetect training previously.

With more samples (also more PRRd cases), we could now see clear separation of the AUCs (revised Figure 4d above). PRRDetect does perform significantly better than MMRDetect (P<2.2e-16), MSIsseq (P=9.773e-05), and TMB (10 muts/Mb) (P<2.2e-16).

The addition of the GEL samples here strengthens the results considerably (pending clarification of the labelling used, outlined in comments above). To further enhance the presentation of the results, I suggest the authors update the above schematic to include the number of cases that are PRRd. I also recommend the authors present an optimal point on the AUC (Youden's index or similar) and mention this in the text so the trade-off between capturing PRRd samples and making (potentially) false predictions can be easily determined.

RTR.v5 (December 2024, 10)

Thank you for the kind suggestion. We have added the number of +ve cases in the validation cohort to the plot ($n=1,351$; 144 +ve). We feel that it may be more easily understandable for general readerships to interpret an AUC score. However, for a more comprehensive comparisons of the algorithms, we also now reported the AUPRC in the main text, starting from line 360: "Then, in an independent validation cohort of 504 ICGC breast cancers^{44,45} and 847 GEL cancers, for which the true labels of PRRd were known, PRRDetect achieved an AUROC (area under the ROC curve) of 1 and an AUPRC (precision recall curve) of 0.99 at distinguishing PRR-dysfunctional from PRR-proficient samples..."

Additionally, we also added Extended Data Table 2, which we referenced in the main text (line 366), to refer interested readers to the detailed performance metrics of these algorithms. Extended Data Table 2 is also reproduced below for the reviewer:

Method	Youden's J Index	Youden's J Index - Specificity	Youden's J Index - Sensitivity	AUC	Sensitivity	Specificity	F1	AUPRC
PRRDetect	0.9766768	0.9854487	0.9912281	1	1	0.9737	0.9988	0.992703
MSIseq	0.877193	1	0.877193	0.94	1	0.8772	0.9944	0.903938
MMRDetect	0.725418	0.8043654	0.9210526	0.87	0.7284	0.9386	0.8401	0.301567
TMB	0.7632927	0.8334681	0.9298246	0.88	0.8335	0.9298	0.906	0.326697

Additional Reviewers' Comments:

Reviewer #2 (Remarks to the Author):

Please see attached pdf for my comments on the previous review and rebuttal (marked in green). Thank you. Please see above.

Reviewer #2 (Remarks on code availability):

The PRRdetect package installs correctly but I was unable to fully test the code without example data. I suggest example data and a vignette be provided.

Thank you. Example data and further instructions are now provided on the repo: <https://github.com/Nik-Zainal-Group/PRRDetect>

There was no documentation provided on how to use functions in the <https://github.com/xqzou/indelsig.tools.lib> repo to reproduce the results found in the paper. This needs to be added.

Thank you. Example data and further instructions are now provided on the repo: <https://github.com/Nik-Zainal-Group/indelsig.tools.lib>.

Links to the browsable data on the <https://signal.mutationalsignatures.com> website are broken.

We have tested the link. It should all work once the reviewer's logged in.

Go to <https://signal.mutationalsignatures.com/login>

On this page login with the following credentials

- Username: *reviewer*
- Password: *Repaint-Grieving6-Patchwork*

If successful you will see this in the top bar:

While in prerelease mode, links to indel data in the paper will work.